# REVISITING PREFIX-TUNING: STATISTICAL BENEFITS OF REPARAMETERIZATION AMONG PROMPTS

**Minh Le**[*1]**, Chau Nguyen**[*1]**, Huy Nguyen**[*2]**, Quyen Tran**[1]**, Trung Le**[3]**, Nhat Ho**[2]

[1] Qualcomm AI Research[†]    [2] The University of Texas at Austin    [3] Monash University

## ABSTRACT

Prompt-based techniques, such as prompt-tuning and prefix-tuning, have gained prominence for their efficiency in fine-tuning large pre-trained models. Despite their widespread adoption, the theoretical foundations of these methods remain limited. For instance, in prefix-tuning, we observe that a key factor in achieving performance parity with full fine-tuning lies in the reparameterization strategy. However, the theoretical principles underpinning the effectiveness of this approach have yet to be thoroughly examined. Our study demonstrates that reparameterization is not merely an engineering trick but is grounded in deep theoretical foundations. Specifically, we show that the reparameterization strategy implicitly encodes a shared structure between prefix key and value vectors. Building on recent insights into the connection between prefix-tuning and mixture of experts models, we further illustrate that this shared structure significantly improves sample efficiency in parameter estimation compared to non-shared alternatives. The effectiveness of prefix-tuning across diverse tasks is empirically confirmed to be enhanced by the shared structure, through extensive experiments in both visual and language domains. Additionally, we uncover similar structural benefits in prompt-tuning, offering new perspectives on its success. Our findings provide theoretical and empirical contributions, advancing the understanding of prompt-based methods and their underlying mechanisms.

## 1 INTRODUCTION

The rapid growth in data availability, along with advances in computational power and training algorithms, has driven the development of numerous foundational models that achieve impressive results across a wide range of tasks (Kaplan et al., 2020; Rae et al., 2021; Dosovitskiy et al., 2021). Leveraging these models' strong generalization abilities, fine-tuning them for downstream tasks has become a widely adopted and successful approach (Iofinova et al., 2022). However, full fine-tuning involves updating all model parameters, demanding storage for separate models per task, which becomes computationally and memory-intensive, especially with models containing billions of parameters (Dosovitskiy et al., 2021; Dehghani et al., 2023; Lialin et al., 2023).

To address these limitations, parameter-efficient fine-tuning (PEFT) has emerged as a promising alternative (Hu et al., 2021; Lian et al., 2022; Xin et al., 2024). By updating only a small subset of parameters, PEFT can achieve performance comparable to, or even surpassing, that of full fine-tuning while significantly reducing computational and memory overhead (Houlsby et al., 2019; Jia et al., 2022). Among these, prompting (Lester et al., 2021; Li & Liang, 2021; Jia et al., 2022) is gaining momentum as a promising solution by updating task-specific tokens while keeping the pre-trained transformer model frozen. Specifically, Lester et al. (2021) introduced trainable continuous embeddings, or continuous prompts, which are appended to the original sequence of input word embeddings, with only these prompts being updated during training. Extending this idea, prefix-tuning (Li & Liang, 2021) optimizes not just the input embeddings but also the inputs to every attention layer within the transformer model, appending them to the key and value vectors.

To ensure stability during optimization, prefix-tuning employs a reparameterization strategy (Li & Liang, 2021; Liu et al., 2021; Han et al., 2024), where prefix vectors are reparameterized rather than

---

[*]Equal Contribution    [†]Qualcomm Vietnam Company Limited

being optimized directly. After training, only the prefix vectors are retained for inference. However, the theoretical justification for this approach remains largely unexplored. Key questions, such as why reparameterization is necessary and what theoretical principles support its effectiveness, have not been comprehensively addressed. In investigating these questions, we argue that reparameterization is not merely an engineering trick but is supported by deep theoretical foundations. Our findings suggest that the reparameterization trick implicitly encodes a *shared structure* between the prefix key and value vectors. Through extensive experiments, we demonstrate that this shared structure plays a pivotal role in enabling prefix-tuning to achieve competitive performance.

Recent work by Le et al. (2024) has revealed that self-attention (Vaswani, 2017) functions as a specialized mixture of experts (MoE) architecture (Jacobs et al., 1991; Jordan & Jacobs, 1994). Within this framework, prefix-tuning serves as a mechanism for introducing new experts into these models. Building on this connection, we provide a detailed analysis of reparameterization from the perspective of expert estimation. We show that the shared structure enhances sample efficiency in prompt estimation compared to cases where the structure is not shared.

**Contribution.** The contributions of this paper can be summarized as follows: **(i)** We uncover that the reparameterization trick in prefix-tuning, often regarded as an engineering technique, is grounded in solid theoretical principles. Specifically, we show that reparameterization induces a shared structure between the prefix key and value vectors, which is crucial in enabling prefix-tuning to achieve competitive performance. **(ii)** Through comprehensive experiments in both visual and linguistic domains, we empirically demonstrate that this shared structure significantly enhances the effectiveness of prefix-tuning, highlighting its importance across diverse tasks. **(iii)** Via the connection between prefix-tuning and mixtures of experts, we provide theoretical justifications for these empirical observations, showing that the shared structure leads to faster convergence rates compared to non-shared alternatives. **(iv)** Furthermore, we observe analogous patterns of shared structure in prompt-tuning. Our insights not only explain the role of common practices in prefix-tuning implementation but also offer a partial exploration of the mechanisms underlying the effectiveness of prompt-tuning.

**Organization.** The rest of the paper is structured as follows. In Section 2, we provide an overview of prompt-based techniques and their connection to the mixture of experts framework. Section 3 introduces the shared structure, which is inspired by the reparameterization strategy. In Section 4, we present theoretical convergence rates for scenarios involving shared structures, demonstrating improved sample efficiency compared to non-shared cases. Section 5 details our empirical evaluations on visual and language tasks. Finally, in Section 6, we discuss the limitations and suggest future directions. Full proofs and experimental details are provided in the appendices.

**Notation.** Firstly, let we denote $[n] = \{1, 2, \ldots, n\}$ for any $n \in \mathbb{N}$. Next, for any vector $u \in \mathbb{R}^d$, we use $u = (u^{(1)}, u^{(2)}, \ldots, u^{(d)})$ and $u = (u_1, u_2, \ldots, u_d)$ interchangeably. Given any $\alpha := (\alpha_1, \alpha_2, \ldots, \alpha_d) \in \mathbb{N}^d$, let $u^\alpha = u_1^{\alpha_1} u_2^{\alpha_2} \ldots u_d^{\alpha_d}$, $|u| := u_1 + u_2 + \ldots + u_d$ and $\alpha! := \alpha_1! \alpha_2! \ldots \alpha_d!$, while $\|u\|$ stands for its 2-norm value. Additionally, let $|S|$ denote its cardinality for any set $S$. Lastly, for any two positive sequences $(a_n)_{n \geq 1}$ and $(b_n)_{n \geq 1}$, we write $a_n = \mathcal{O}(b_n)$ or $a_n \lesssim b_n$ if $a_n \leq C b_n$ for all $n \in \mathbb{N}$, where $C > 0$ is some universal constant. The notation $a_n = \mathcal{O}_P(b_n)$ indicates that $a_n / b_n$ is stochastically bounded.

## 2 BACKGROUND

We begin by reviewing the background of prompt-based fine-tuning techniques. Following this, we describe the concept of mixture of experts models and examine how prefix-tuning can be interpreted within the context of MoE models. A detailed discussion of related work is provided in Appendix D.

### 2.1 PROMPT-BASED APPROACHES

The Transformer (Vaswani, 2017; Dosovitskiy et al., 2021) architecture comprises multiple multi-head self-attention (MSA) layers. To illustrate the function of a single MSA layer, consider an input sequence of embeddings $[\boldsymbol{x}_1, \ldots, \boldsymbol{x}_N]^\top \in \mathbb{R}^{N \times d}$, where $N$ is the sequence length and $d$ is the embedding dimension. The MSA layer processes this sequence as follows:

$$\text{MSA}(\boldsymbol{X}_Q, \boldsymbol{X}_K, \boldsymbol{X}_V) := \text{Concat}(\boldsymbol{h}_1, \ldots, \boldsymbol{h}_m) W^O \in \mathbb{R}^{N \times d}, \tag{1}$$

$$\boldsymbol{h}_i := \text{Attention}(\boldsymbol{X}_Q W_i^Q, \boldsymbol{X}_K W_i^K, \boldsymbol{X}_V W_i^V), \; i \in [m], \tag{2}$$

where $\boldsymbol{X}_Q = \boldsymbol{X}_K = \boldsymbol{X}_V = [\boldsymbol{x}_1, \ldots, \boldsymbol{x}_N]^\top$ are the query, key, and value matrices, respectively. Here $m$ is the number of heads, and $W^O \in \mathbb{R}^{md_v \times d}$ is the projection matrix. Each attention head $\boldsymbol{h}_i$

is parameterized by $W_i^Q \in \mathbb{R}^{d \times d_k}, W_i^K \in \mathbb{R}^{d \times d_k}$, and $W_i^V \in \mathbb{R}^{d \times d_v}$ with $d_k = d_v = \frac{d}{m}$. Building on this, fine-tuning techniques such as prompt-tuning (Lester et al., 2021) and prefix-tuning (Li & Liang, 2021) have emerged as efficient methods for adapting pre-trained transformer-based models to downstream tasks. These methods introduce prompt parameters $\boldsymbol{P} \in \mathbb{R}^{N_p \times d}$, which are used to modify the input embeddings fed into MSA layers, where $N_p$ denotes the prompt length.

**Prompt-tuning** involves prepending prompt vectors to the input embeddings, which is equivalent to concatenating the same prompt parameters $\boldsymbol{P}$ to $\boldsymbol{X}_Q, \boldsymbol{X}_K$, and $\boldsymbol{X}_V$:

$$f_{\text{prompt}}^{\text{Pro}-\text{T}}(\boldsymbol{X}_Q, \boldsymbol{X}_K, \boldsymbol{X}_V; \boldsymbol{P}) := \text{MSA}\left(\begin{bmatrix} \boldsymbol{P} \\ \boldsymbol{X}_Q \end{bmatrix}, \begin{bmatrix} \boldsymbol{P} \\ \boldsymbol{X}_K \end{bmatrix}, \begin{bmatrix} \boldsymbol{P} \\ \boldsymbol{X}_V \end{bmatrix}\right) = \text{Concat}(\hat{\boldsymbol{h}}_1, ..., \hat{\boldsymbol{h}}_m)W^O, \quad (3)$$

resulting in an output in $\mathbb{R}^{(N+N_p) \times d}$ with increased dimensions.

**Prefix-tuning** decomposes $\boldsymbol{P}$ into $\boldsymbol{P}_K \in \mathbb{R}^{\frac{N_p}{2} \times d}$ and $\boldsymbol{P}_V \in \mathbb{R}^{\frac{N_p}{2} \times d}$, which are then appended to $\boldsymbol{X}_K$ and $\boldsymbol{X}_V$, respectively:

$$f_{\text{prompt}}^{\text{Pre}-\text{T}}(\boldsymbol{X}_Q, \boldsymbol{X}_K, \boldsymbol{X}_V; \boldsymbol{P}) := \text{MSA}\left(\boldsymbol{X}_Q, \begin{bmatrix} \boldsymbol{P}_K \\ \boldsymbol{X}_K \end{bmatrix}, \begin{bmatrix} \boldsymbol{P}_V \\ \boldsymbol{X}_V \end{bmatrix}\right) = \text{Concat}(\tilde{\boldsymbol{h}}_1, ..., \tilde{\boldsymbol{h}}_m)W^O. \quad (4)$$

In contrast to prompt-tuning, prefix-tuning preserves the output sequence length, keeping it identical to the input sequence length and enabling flexible adaptation across the network.

## 2.2 MIXTURE OF EXPERTS MEETS PREFIX-TUNING

An MoE model consists of $N'$ expert networks, $f_i : \mathbb{R}^d \to \mathbb{R}^{d_v}$ for $i \in [N']$, and a gating function $G : \mathbb{R}^d \to \mathbb{R}^{N'}$ that allocates contributions of each expert based on the input $\boldsymbol{x}$. The gating mechanism uses learned score functions, $s_i : \mathbb{R}^d \to \mathbb{R}$, associated with each expert, resulting in:

$$\hat{\mathbf{y}} = \sum_{i=1}^{N'} G(\boldsymbol{x})_i \cdot f_i(\boldsymbol{x}) = \sum_{i=1}^{N'} \frac{\exp\left(s_i(\boldsymbol{x})\right)}{\sum_{j=1}^{N'} \exp\left(s_j(\boldsymbol{x})\right)} \cdot f_i(\boldsymbol{x}), \quad (5)$$

where $G(\boldsymbol{x}) = \text{softmax}(s_1(\boldsymbol{x}), \ldots, s_{N'}(\boldsymbol{x}))$. Building on this formulation, recent work by Le et al. (2024) demonstrates that each attention head within the MSA layer can be interpreted as a specialized architecture composed of multiple MoE models. The study further suggests that prefix-tuning serves as a mechanism for introducing new experts into these MoE models, facilitating their adaptation to downstream tasks. Specifically, from equation (4), consider the output of the $l$-th head $\tilde{\boldsymbol{h}}_l = [\tilde{\boldsymbol{h}}_{l,1}, \ldots, \tilde{\boldsymbol{h}}_{l,N}]^\top \in \mathbb{R}^{N \times d_v}$. Let $\boldsymbol{X} = \begin{bmatrix} \boldsymbol{x}_1^\top, \ldots, \boldsymbol{x}_N^\top \end{bmatrix}^\top \in \mathbb{R}^{Nd}$ represent the concatenated input embeddings, and let $\boldsymbol{P}_K = \begin{bmatrix} \boldsymbol{p}_1^K, \ldots, \boldsymbol{p}_L^K \end{bmatrix}^\top \in \mathbb{R}^{L \times d}, \boldsymbol{P}_V = \begin{bmatrix} \boldsymbol{p}_1^V, \ldots, \boldsymbol{p}_L^V \end{bmatrix}^\top \in \mathbb{R}^{L \times d}$, where $L = \frac{N_p}{2}$. We define $N$ pre-trained experts $f_j : \mathbb{R}^{Nd} \to \mathbb{R}^{d_v}$ encoded in the MSA layer, along with $L$ prefix experts $f_{N+j'} : \mathbb{R}^{Nd} \to \mathbb{R}^{d_v}$ introduced via the prompt as follows:

$$f_j(\boldsymbol{X}) := W_l^{V \top} E_j \boldsymbol{X} = W_l^{V \top} \boldsymbol{x}_j, \quad f_{N+j'}(\boldsymbol{X}) := W_l^{V \top} \boldsymbol{p}_{j'}^V,$$

for $j \in [N]$ and $j' \in [L]$, where the matrix $E_j \in \mathbb{R}^{d \times Nd}$ is such that $E_j \boldsymbol{X} := \boldsymbol{x}_j$. Next, we introduce $N \times (N + L)$ score functions, $s_{i,j} : \mathbb{R}^{Nd} \to \mathbb{R}$, associated with these experts:

$$s_{i,j}(\boldsymbol{X}) := \frac{\boldsymbol{X}^\top E_i^\top W_l^Q W_l^{K \top} E_j \boldsymbol{X}}{\sqrt{d_v}}, \quad s_{i,N+j'}(\boldsymbol{X}) := \frac{\boldsymbol{X}^\top E_i^\top W_l^Q W_l^{K \top} \boldsymbol{p}_{j'}^K}{\sqrt{d_v}},$$

for $i \in [N], j \in [N]$ and $j' \in [L]$. Consequently, each output vector $\tilde{\boldsymbol{h}}_{l,i}$ can be formulated as the result of an MoE model, utilizing the experts and score functions defined above:

$$\tilde{\boldsymbol{h}}_{l,i} = \sum_{j=1}^{N} \frac{\exp(s_{i,j}(\boldsymbol{X}))}{\sum_{k=1}^{N} \exp(s_{i,k}(\boldsymbol{X})) + \sum_{k'=1}^{L} \exp(s_{i,N+k'}(\boldsymbol{X}))} f_j(\boldsymbol{X})$$

$$+ \sum_{j'=1}^{L} \frac{\exp(s_{i,N+j'}(\boldsymbol{X}))}{\sum_{k=1}^{N} \exp(s_{i,k}(\boldsymbol{X})) + \sum_{k'=1}^{L} \exp(s_{i,N+k'}(\boldsymbol{X}))} f_{N+j'}(\boldsymbol{X}). \quad (6)$$

Notably, only $\boldsymbol{P}_K$ and $\boldsymbol{P}_V$ are learnable, meaning that only the prefix experts $f_{N+j'}$ and their corresponding score functions $s_{i,N+j'}$ are trained. These new experts work in conjunction with the pre-trained ones embedded in the original model, enabling efficient adaptation to downstream tasks.

## 3 MOTIVATION: REPARAMETERIZATION STRATEGY

In this section, we first introduce the concept of shared structure, derived from the reparameterization technique. We then explain how this structure is integrated into the formulation of prompt-tuning.

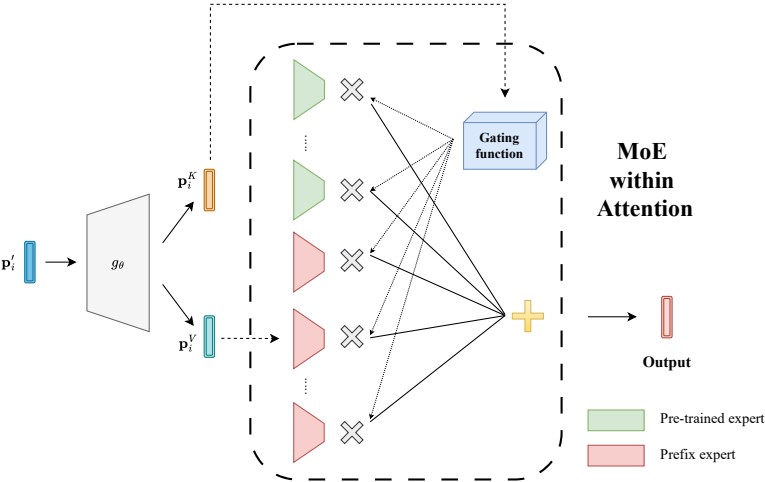

Figure 1: Reparameterization defines both the prefix key $\boldsymbol{p}_i^K$ and value $\boldsymbol{p}_i^V$ as functions of shared parameters $\boldsymbol{p}_i'$, transformed by $g_\theta$. This introduces parameter sharing between the score functions and expert parameters in the MoE framework in attention. The gating function computes expert weights based on score functions, and the MoE output is a weighted average of all expert outputs.

In equation (4), instead of directly updating the prompt parameters $\boldsymbol{P}_K$ and $\boldsymbol{P}_V$, which can lead to unstable optimization and a slight drop in performance, Li & Liang (2021) proposed reparameterizing the matrix $[\boldsymbol{P}_K, \boldsymbol{P}_V] \in \mathbb{R}^{L \times 2d}$ using a smaller matrix $\boldsymbol{P}' = [\boldsymbol{p}_1', \ldots, \boldsymbol{p}_L']^\top \in \mathbb{R}^{L \times d}$, which is then composed with a feedforward neural network $g_\theta : \mathbb{R}^d \to \mathbb{R}^{2d}$,

$$\left[\boldsymbol{p}_i^K, \boldsymbol{p}_i^V\right] = g_\theta(\boldsymbol{p}_i'), \tag{7}$$

for $i = 1, \ldots, L$, where $L = \frac{N_p}{2}$. After training, the reparameterization can be discarded, and only the final prompt parameters, $\boldsymbol{P}_K$ and $\boldsymbol{P}_V$, need to be stored. We observe that the reparameterization strategy implicitly encodes a shared structure between the prefix key and prefix value vectors. This relationship can be made explicit by reformulating equation (7) as follows:

$$\boldsymbol{p}_i^K = \sigma_1(\boldsymbol{p}_i'), \ \boldsymbol{p}_i^V = \sigma_2(\boldsymbol{p}_i'), \tag{8}$$

where $\sigma_1 : \mathbb{R}^d \to \mathbb{R}^d$ and $\sigma_2 : \mathbb{R}^d \to \mathbb{R}^d$ are two functions derived from $g_\theta$. Both the prefix key $\boldsymbol{p}_i^K$ and prefix value $\boldsymbol{p}_i^V$ are functions of the same underlying parameters $\boldsymbol{p}_i'$ but modulated by distinct transformations $\sigma_1$ and $\sigma_2$. We refer to this as the *shared structure* among the prompt parameters.

As discussed in Section 2.2, drawing from the relationship between prefix tuning and MoE models, the prefix key and value can be viewed as corresponding to the score functions and expert parameters, respectively. This suggests that the shared structure introduces a form of parameter sharing between the score functions and expert parameters within the MoE framework in attention, as illustrated in Figure 1. In Section 4, we show that this sharing strategy enhances sample efficiency from the perspective of the parameter estimation problem, compared to models without such shared structure.

**Shared structure in prompt-tuning.** Prompt-tuning, by attaching prompt parameters to the key, query, and value matrices, refines pre-trained MoE models by integrating additional experts, similar to prefix-tuning, and also allows the incorporation of new MoE models. Detailed proof is provided in Appendix A. While prompt-tuning can integrate new MoE models, our study focuses on pre-trained MoE models within each attention head as a preliminary exploration of the underlying mechanism.

As shown in equation (3), prompt-tuning employs a single prompt parameter $\boldsymbol{P}$ for both key and value vectors. We find that this strategy also introduces a shared structure, similar to the pattern

described in Section 3. Specifically, the prefix key and prefix value vectors are now expressed as:

$$\boldsymbol{P}_K = \sigma_1(\boldsymbol{P}) = \boldsymbol{P}, \; \boldsymbol{P}_V = \sigma_2(\boldsymbol{P}) = \boldsymbol{P}, \tag{9}$$

where $\sigma_1$ and $\sigma_2$ are identity functions. Consequently, prompt-tuning encodes a shared structure between key and value vectors, leading to parameter sharing between the score functions and expert parameters in pre-trained MoE models. As discussed further in Section 4, this parameter-sharing mechanism promotes faster convergence in parameter estimation, offering theoretical justifications for using the same prompt parameters for both key and value vectors. We posit that these insights contribute to a partial explanation of the efficiency and effectiveness of prompt-tuning, which applies the same prompt parameters to the key, query, and value matrices.

## 4 THEORETICAL ANALYSIS FOR PROMPT LEARNING IN PREFIX-TUNING

The interpretation of prefix-tuning via mixtures of experts in equation (6) provides a natural way to understand prompt learning in prefix-tuning via the convergence analysis of prompt estimation in these MoE models. Moreover, as shown in equation (6), each MoE model in each attention head follows a similar structure. Thus, to simplify the presentation of our analysis, we focus only on the first head, namely, $l = 1$ in equation (6), and the first row of the attention in this head, namely, $i = 1$ in equation (6). In particular, we consider a regression framework for MoE models as follows.

**Setting.** We assume that $(\boldsymbol{X}_1, Y_1), (\boldsymbol{X}_2, Y_2), \ldots, (\boldsymbol{X}_n, Y_n) \in \mathbb{R}^d \times \mathbb{R}$ are i.i.d. samples of size $n$ generated from the model:

$$Y_i = f_{G_*}(\boldsymbol{X}_i) + \varepsilon_i, \quad i = 1, 2, \ldots, n, \tag{10}$$

where $\varepsilon_1, \ldots, \varepsilon_n$ are independent Gaussian noise variables such that $\mathbb{E}[\varepsilon_i | \boldsymbol{X}_i] = 0$ and $\mathrm{Var}(\varepsilon_i | \boldsymbol{X}_i) = \nu^2$ for all $1 \leq i \leq n$. Additionally, we assume that $\boldsymbol{X}_1, \boldsymbol{X}_2, \ldots, \boldsymbol{X}_n$ are i.i.d. samples from some probability distribution $\mu$. The regression function $f_{G_*}(\cdot)$ in equation (10) then takes the form of a prefix MoE model with $N$ pre-trained experts and $L$ unknown experts,

$$f_{G_*}(\boldsymbol{X}) := \sum_{j=1}^{N} \frac{\exp(\boldsymbol{X}^\top A_j^0 \boldsymbol{X} + a_j^0)}{D_f(\boldsymbol{X})} \cdot h(\boldsymbol{X}, \eta_j^0) + \sum_{j'=1}^{L} \frac{\exp((B\boldsymbol{p}_{*,j'}^K)^\top \boldsymbol{X} + b_{*,j'})}{D_f(\boldsymbol{X})} \cdot C\boldsymbol{p}_{*,j'}^V, \tag{11}$$

where $D_f(\boldsymbol{X}) := \sum_{k=1}^{N} \exp(\boldsymbol{X}^\top A_k^0 \boldsymbol{X} + a_k^0) + \sum_{j'=1}^{L} \exp((B\boldsymbol{p}_{*,j'}^K)^\top \boldsymbol{X} + b_{*,j'})$, while $G_* := \sum_{j'=1}^{L} \exp(b_{*,j'}) \delta_{(\boldsymbol{p}_{*,j'}^K, \boldsymbol{p}_{*,j'}^V)}$ denotes a *mixing measure,* i.e., a weighted sum of Dirac measures $\delta$, associated with unknown parameters $(b_{*,j'}, \boldsymbol{p}_{*,j'}^K, \boldsymbol{p}_{*,j'}^V)_{j'=1}^L$ in the parameter space $\Theta \subset \mathbb{R} \times \mathbb{R}^d \times \mathbb{R}^d$. At the same time, the values of the matrix $A_j^0$, the expert parameter $\eta_j^0$, and the bias parameter $a_j^0$ are known for all $1 \leq j \leq N$. Additionally, the matrices $B \in \mathbb{R}^{d \times d}$ and $C \in \mathbb{R}^{1 \times d}$ are given and they play the role of pre-trained projection matrices in the context of prefix-tuning in equation (6).

In the sequel, we will investigate the convergence behavior of estimation for the unknown prompt parameters. Our main objective is to show that the convergence rates of the prompts will be accelerated when they share the structure, that is, they can be reparametrized as $\boldsymbol{p}^K = \sigma_1(\boldsymbol{p})$ and $\boldsymbol{p}^V = \sigma_2(\boldsymbol{p})$, for some functions $\sigma_1$ and $\sigma_2$, as motivated in Section 3. To this end, we will conduct the convergence analysis of prompt estimation when there are non-shared and shared structures among the ground-truth prompts in Section 4.1 and Section 4.2, respectively. Then, we compare the convergence rates in these scenarios to highlight the sample efficiency of the latter method.

### 4.1 WITHOUT REPARAMETRIZATION (NONSHARED STRUCTURES) AMONG PROMPTS

In this section, we first investigate the scenario when the prompt parameters do not share the inner structure, where we need to learn the prompts $\boldsymbol{p}_{*,j'}^K$ and $\boldsymbol{p}_{*,j'}^V$ in equation (11) separately. To estimate those unknown prompts or, equivalently, the ground-truth mixing measure $G_*$, we use the least square method (van de Geer, 2000). In particular, we take into account the estimator

$$\widehat{G}_n := \underset{G \in \mathcal{G}_{L'}(\Theta)}{\arg\min} \sum_{i=1}^{n} \left( Y_i - f_G(\boldsymbol{X}_i) \right)^2, \tag{12}$$

where we denote $\mathcal{G}_{L'}(\Theta) := \{G = \sum_{i=1}^{\ell} \exp(b_i) \delta_{(\boldsymbol{p}_i^K, \boldsymbol{p}_i^V)} : 1 \leq \ell \leq L', (b_i, \boldsymbol{p}_i^K, \boldsymbol{p}_i^V) \in \Theta\}$ as the set of all mixing measures with at most $L'$ atoms. In practice, since the true number of experts $L$ is

typically unknown, we assume that the number of fitted experts $L'$ is sufficiently large, i.e., $L' > L$. In order to characterize the convergence rate of prompt estimation, it is necessary to construct a loss function among prompt parameters. To this end, we propose using a loss function based on the concept of Voronoi cells (Manole & Ho, 2022), which we refer to as the Voronoi loss function.

**Voronoi loss.** For a mixing measure $G$ with $L \leq L'$ atoms, we distribute its atoms to the following Voronoi cells $\mathcal{V}_j \equiv \mathcal{V}_j(G)$, for $j \in [L]$, generated by the atoms of $G_*$:

$$\mathcal{V}_j := \{i \in [L'] : \|(\boldsymbol{p}_i^K, \boldsymbol{p}_i^V) - (\boldsymbol{p}_{*,j}^K, \boldsymbol{p}_{*,j}^V)\| \leq \|(\boldsymbol{p}_i^K, \boldsymbol{p}_i^V) - (\boldsymbol{p}_{*,\ell}^K, \boldsymbol{p}_{*,\ell}^V)\|, \forall \ell \neq j\}. \tag{13}$$

Then, the Voronoi loss function of interest is defined as

$$\mathcal{D}_{1,r}(G, G_*) := \sum_{j'=1}^{L} \Big| \sum_{i \in \mathcal{V}_{j'}} \exp(b_i) - \exp(b_{*,j'}) \Big| + \sum_{j'=1}^{L} \sum_{i \in \mathcal{V}_{j'}} \exp(b_i) \Big[ \|\Delta \boldsymbol{p}_{ij'}^K\|^r + \|\Delta \boldsymbol{p}_{ij'}^V\|^r \Big],$$

for $r \in \mathbb{N}$, where we denote $\Delta \boldsymbol{p}_{ij'}^K := \boldsymbol{p}_i^K - \boldsymbol{p}_{*,j'}^K$ and $\Delta \boldsymbol{p}_{ij'}^V := \boldsymbol{p}_i^V - \boldsymbol{p}_{*,j'}^V$. Given this loss function, we are now ready to capture the convergence behavior of prompts in the following theorem.

**Theorem 4.1.** *The following bound of estimating $G_*$ holds for any $r \in \mathbb{N}$:*

$$\sup_{G \in \mathcal{G}_{L'}(\Theta) \backslash \mathcal{G}_{L-1}(\Theta)} \mathbb{E}_{f_G}[\mathcal{D}_{1,r}(\widehat{G}_n, G)] \gtrsim n^{-1/2}, \tag{14}$$

*where $\mathbb{E}_{f_G}$ indicates the expectation taken w.r.t the product measure with $f_G^n$.*

Proof of Theorem 4.1 is in Appendix B.1. The bound in equation (14) together with the formulation of the loss $\mathcal{D}_{1,r}$ implies that the convergence rates of estimations for both the prompts $\boldsymbol{p}_{*,j'}^K$ and $\boldsymbol{p}_{*,j'}^V$ are slower than $\mathcal{O}(n^{-1/2r})$ for any $r \in \mathbb{N}$ and, therefore, could be as significantly slow as $\mathcal{O}(1/\log(n))$. This observation indicates that the performance of prompt learning will be negatively affected when there are no shared structures among the prompt parameters.

## 4.2 WITH REPARAMETRIZATION (SHARED STRUCTURES) AMONG PROMPTS

In this section, we consider the scenario when the prompts share their structures with each other. In particular, we reparameterize the prompts as $\boldsymbol{p}^K = \sigma_1(\boldsymbol{p})$ and $\boldsymbol{p}^V = \sigma_2(\boldsymbol{p})$ where $\boldsymbol{p} \in \mathbb{R}^{d'}$, the functions $\sigma_1, \sigma_2 : \mathbb{R}^{d'} \to \mathbb{R}^d$, and the dimension $d' \geq 1$ is given. That parametrization indicates that the prompts will share the input of the functions $\sigma_1$ and $\sigma_2$.

To theoretically demonstrate the benefits of reparametrization among prompts in prompt learning, we specifically take into account the following two settings of the functions $\sigma_1$ and $\sigma_2$:

**(i)** *Simple linear setting*: $\sigma_1(\boldsymbol{p}) = \boldsymbol{p}$ and $\sigma_2(\boldsymbol{p}) = \boldsymbol{p}$ for any $\boldsymbol{p} \in \mathbb{R}^d$;

**(ii)** *One-layer neural network setting*: $\sigma_1(\boldsymbol{p}) = \bar{\sigma}_1(W_1\boldsymbol{p})$ and $\sigma_2(\boldsymbol{p}) = \bar{\sigma}_2(W_2\boldsymbol{p})$ for any $\boldsymbol{p} \in \mathbb{R}^{d'}$ where $W_1 \in \mathbb{R}^{d \times d'}$ and $W_2 \in \mathbb{R}^{d \times d'}$ are learnable weights.

Here, $\bar{\sigma}_1$ and $\bar{\sigma}_2$ are two given real-valued activation functions. Furthermore, for any vector $x = (x^{(1)}, \ldots, x^{(d)}) \in \mathbb{R}^d$, we denote $\bar{\sigma}_i(x) = (\bar{\sigma}_i(x^{(1)}), \ldots, \bar{\sigma}_i(x^{(d)}))$ for any $1 \leq i \leq 2$, that is, the functions $\bar{\sigma}_1$ and $\bar{\sigma}_2$ are applied to each element of the vector $x$.

### 4.2.1 SIMPLE LINEAR SETTING

We begin our analysis with the simple linear setting under which $\boldsymbol{p}^K = \sigma_1(\boldsymbol{p}) = \boldsymbol{p}$ and $\boldsymbol{p}^V = \sigma_2(\boldsymbol{p}) = \boldsymbol{p}$ for any $\boldsymbol{p} \in \mathbb{R}^d$. This setting is motivated by prompt-tuning strategy as being discussed in Section 3. Then, the ground-truth regression function in equation (11) turns into

$$f_{\bar{G}_*}(\boldsymbol{X}) := \frac{\sum_{j=1}^{N} \exp(\boldsymbol{X}^\top A_j^0 \boldsymbol{X} + a_j^0) h(\boldsymbol{X}, \eta_j^0) + \sum_{j'=1}^{L} \exp((B\boldsymbol{p}_{*,j'})^\top \boldsymbol{X} + b_{*,j'}) \cdot C\boldsymbol{p}_{*,j'}}{D_f(\boldsymbol{X})},$$

where $\bar{G}_* = \sum_{j'=1}^{L} \exp(b_{*,j'}) \delta_{\boldsymbol{p}_{*,j'}}$ is a mixing measure with unknown parameters $(b_{*,j'}, \boldsymbol{p}_{*,j'})_{j'=1}^{L}$ belonging to the parameter space $\Omega \subset \mathbb{R} \times \mathbb{R}^{d'}$. To ensure the identifiability of estimating prompts in the simple linear setting, we assume that $B\boldsymbol{p}_{*,1}, \ldots, B\boldsymbol{p}_{*,L}$ are pairwise different. Similar to the

nonshared structure setting of prompts in Section 4.1, we also employ the least square method to estimate the unknown parameters or, equivalently the mixing measure $\bar{G}_*$. In particular, the least square estimator of interest is given by:

$$\bar{G}_n := \underset{\bar{G} \in \bar{\mathcal{G}}_{L'}(\Omega)}{\arg\min} \sum_{i=1}^{n} \left(Y_i - f_{\bar{G}}(\boldsymbol{X}_i)\right)^2, \tag{15}$$

where $\bar{\mathcal{G}}_{L'}(\Omega) := \{\bar{G} = \sum_{i=1}^{\ell} \exp(b_i)\delta_{\boldsymbol{p}_i} : 1 \leq \ell \leq L', (b_i, \boldsymbol{p}_i) \in \Omega\}$ is the set of all mixing measures with at most $L'$ atoms, where $L' > L$, and parameters belonging to the space $\Omega$. Then, we need to build a new Voronoi loss function to capture the convergence rate of prompt estimation.

**Voronoi loss.** The Voronoi loss tailored to the simple linear setting of prompts is defined as

$$\mathcal{D}_2(\bar{G}, \bar{G}_*) := \sum_{j'=1}^{L} \left| \sum_{i \in \mathcal{V}_{j'}} \exp(b_i) - \exp(b_{*,j'}) \right| + \sum_{j' \in [L]: |\mathcal{V}_{j'}|=1} \sum_{i \in \mathcal{V}_{j'}} \exp(b_i)\|\Delta\boldsymbol{p}_{ij'}\|$$
$$+ \sum_{j' \in [L]: |\mathcal{V}_{j'}|>1} \sum_{i \in \mathcal{V}_{j'}} \exp(b_i)\|\Delta\boldsymbol{p}_{ij'}\|^2,$$

where we denote $\Delta\boldsymbol{p}_{ij'} := \boldsymbol{p}_i - \boldsymbol{p}_{*,j'}$ for any $i, j'$. Equipped with this loss function, we wrap up the simple linear setting of prompts by providing the convergence rate of prompt estimation in Theorem 4.2 whose proof is deferred to Appendix B.2.

**Theorem 4.2.** *Given the least square estimator $\bar{G}_n$ defined in equation (15), we have that*

$$\mathcal{D}_2(\bar{G}_n, \bar{G}_*) = \mathcal{O}_P(\sqrt{\log(n)/n}).$$

It follows from the bound in Theorem 4.2 and the formulation of the loss $\mathcal{D}_2$ that for prompts $\boldsymbol{p}_{*,j'}$ whose Voronoi cells have exactly one element, that is $|\mathcal{V}_{j'}| = 1$, the rate for estimating them is of order $\mathcal{O}_P(\sqrt{\log(n)/n})$, which is parametric on the sample size $n$. On the other hand, the estimation rate for those whose Voronoi cells have more than one element, that is $|\mathcal{V}_{j'}| > 1$, is slightly slower, standing at the order of $\mathcal{O}_P(\sqrt[4]{\log(n)/n})$. In both cases, it is clear that these prompt estimation rates are substantially faster than those in Theorem 4.1, which could be as slow as $\mathcal{O}(1/\log(n))$. Therefore, we can claim that reparameterizing the prompts as $\boldsymbol{p}^K = \boldsymbol{p}^V = \boldsymbol{p}$ helps enhance the sample efficiency of the prompt learning process, thereby leading to a superior performance to the scenario when there are no shared structures among prompts in Section 4.1.

### 4.2.2 ONE-LAYER NEURAL NETWORK SETTING

We now move to the setting where the prompts are reparameterized as one-layer neural networks, that is, $\boldsymbol{p}^K = \sigma_1(\boldsymbol{p}) = \bar{\sigma}_1(W_1\boldsymbol{p})$ and $\boldsymbol{p}^V = \sigma_2(\boldsymbol{p}) = \bar{\sigma}_2(W_2\boldsymbol{p})$ in which $W_1 \in \mathbb{R}^{d \times d'}, W_2 \in \mathbb{R}^{d \times d'}$ are learnable weight matrices and $\bar{\sigma}_1, \bar{\sigma}_2$ are two given real-valued element-wise activation functions. Our goal is to demonstrate that the reparametrization among prompts still yields sample efficiency benefits beyond the simple linear setting in Section 4.2.1. Different from the simple linear setting, the true regression function under the one-layer neural network setting takes the form:

$$f_{\widetilde{G}_*}(\boldsymbol{X}) := \sum_{j=1}^{N} \frac{\exp(\boldsymbol{X}^\top A_j^0 \boldsymbol{X} + a_j^0)}{D_f(\boldsymbol{X})} \cdot h(\boldsymbol{X}, \eta_j^0)$$
$$+ \sum_{j'=1}^{L} \frac{\exp((B\bar{\sigma}_1(W_{*,1}\boldsymbol{p}_{*,j'}))^\top \boldsymbol{X} + b_{*,j'})}{D_f(\boldsymbol{X})} \cdot C\bar{\sigma}_2(W_{*,2}\boldsymbol{p}_{*,j'}),$$

where the true mixing measure is of the form $\widetilde{G}_* := \sum_{j'=1}^{L} \exp(b_{*,j'})\delta_{(W_{*,1}\boldsymbol{p}_{*,j'}, W_{*,2}\boldsymbol{p}_{*,j'})}$, that is, a weighted sum of Dirac measures associated with unknown parameters $(b_{*,j'}, W_{*,1}\boldsymbol{p}_{*,j'}, W_{*,2}\boldsymbol{p}_{*,j'})_{j'=1}^{L}$ in the parameter space $\Xi \subset \mathbb{R} \times \mathbb{R}^d \times \mathbb{R}^d$. To guarantee the identifiability of prompt estimation in the one-layer neural network setting, we assume that $B\bar{\sigma}_1(W_{*,1}\boldsymbol{p}_{*,1}), \ldots, B\bar{\sigma}_1(W_{*,1}\boldsymbol{p}_{*,L})$ are pairwise different. In order to estimate these unknown parameters, we utilize the least square estimator, which is given by:

$$\widetilde{G}_n := \underset{\widetilde{G} \in \widetilde{\mathcal{G}}_{L'}(\Xi)}{\arg\min} \sum_{i=1}^{n} \left(Y_i - f_{\widetilde{G}}(\boldsymbol{X}_i)\right)^2, \tag{16}$$

where $\widetilde{\mathcal{G}}_{L'}(\Xi) := \{\widetilde{G} = \sum_{i=1}^{\ell} \exp(b_i)\delta_{(W_1\boldsymbol{p}_i, W_2\boldsymbol{p}_i)} : 1 \leq \ell \leq L', \ (b_i, W_1\boldsymbol{p}_i, W_2\boldsymbol{p}_i) \in \Xi\}$ as the set of mixing measures with at most $L'$ atoms, where $L' > L$, and with parameters in the space $\Xi$.

**Voronoi loss.** In alignment with the regression function change, it is necessary to construct an appropriate Voronoi loss function for the analysis of this setting, which is given by:

$$
\begin{aligned}
\mathcal{D}_3(\widetilde{G}, \widetilde{G}_*) := \sum_{j'=1}^{L} \Big| \sum_{i \in \mathcal{V}_{j'}} \exp(b_i) - \exp(b_{*,j'}) \Big| \\
+ \sum_{j' \in [L]: |\mathcal{V}_{j'}|=1} \sum_{i \in \mathcal{V}_{j'}} \exp(b_i)(\|W_1\boldsymbol{p}_i - W_{*,1}\boldsymbol{p}_{*,j'}\| + \|W_2\boldsymbol{p}_i - W_{*,2}\boldsymbol{p}_{*,j'}\|) \\
+ \sum_{j' \in [L]: |\mathcal{V}_{j'}|>1} \sum_{i \in \mathcal{V}_{j'}} \exp(b_i)(\|W_1\boldsymbol{p}_i - W_{*,1}\boldsymbol{p}_{*,j'}\|^2 + \|W_2\boldsymbol{p}_i - W_{*,2}\boldsymbol{p}_{*,j'}\|^2).
\end{aligned}
$$

Subsequently, since the prompt reparametrization under this setting involves the activation functions $\bar{\sigma}_1$ and $\bar{\sigma}_2$, let us introduce two standard assumptions on these two functions prior to presenting the convergence analysis of prompt estimation.

**Assumptions.** The two activation functions $\bar{\sigma}_1$ and $\bar{\sigma}_2$ are given such that the followings holds:

*(A.1) (Uniform Lipschitz) Let $F(\boldsymbol{X}; W_1\boldsymbol{p}, W_2\boldsymbol{p}) := \exp((B\bar{\sigma}_1(W_1\boldsymbol{p}))^\top \boldsymbol{X})C\bar{\sigma}_2(W_2\boldsymbol{p})$. Then, for any $r \in \{1, 2\}$, we have*

$$
\sum_{|\alpha|=r} \left| \left( \frac{\partial^{|\alpha|} F}{\partial(W_1\boldsymbol{p})^{\alpha_1}\partial(W_2\boldsymbol{p})^{\alpha_2}}(\boldsymbol{X}; W_1\boldsymbol{p}, W_2\boldsymbol{p}) - \frac{\partial^{|\alpha|} F}{\partial(W_1\boldsymbol{p})^{\alpha_1}\partial(W_2\boldsymbol{p})^{\alpha_2}}(\boldsymbol{X}; W_1\boldsymbol{p}', W_2\boldsymbol{p}') \right)\gamma^\alpha \right|
$$
$$
\leq C\|(W_1\boldsymbol{p}, W_2\boldsymbol{p}) - (W_1\boldsymbol{p}', W_2\boldsymbol{p}')\|^\zeta \|\gamma\|^r,
$$

*for any vector $\gamma \in \mathbb{R}^{2d}$ and for some positive constants $\zeta$ and $C$ which are independent of $\boldsymbol{X}$ and $(W_1\boldsymbol{p}, W_2\boldsymbol{p}), (W_1\boldsymbol{p}', W_2\boldsymbol{p}')$. Here, $\alpha = (\alpha_1, \alpha_2) \in \mathbb{N}^{2d}$ where $\alpha_1, \alpha_2 \in \mathbb{N}^d$.*

*(A.2) (Non-zero derivatives) $\frac{\partial^2 \bar{\sigma}_2}{\partial(W_2\boldsymbol{p})^{(u)}\partial(W_2\boldsymbol{p})^{(u)}}(W_{*,2}\boldsymbol{p}_{*,j'}) \neq 0$, for all $u \in [d]$ and $j' \in [L]$.*

**Example.** We can validate that $\bar{\sigma}_1(W_1\boldsymbol{p}) = \tanh(W_1\boldsymbol{p})$ and $\bar{\sigma}_2(W_2\boldsymbol{p}) = \tanh(W_2\boldsymbol{p})$, where the function $\tanh$ is applied element-wise, meet both the assumptions (A.1) and (A.2). By contrast, if $\bar{\sigma}_2$ is a linear function, e.g. $\bar{\sigma}_2(W_2\boldsymbol{p}) = W_2\boldsymbol{p}$, then the assumption (A.2) is violated.

**Theorem 4.3.** *Assume that the given activation functions $\bar{\sigma}_1$ and $\bar{\sigma}_2$ satisfy both the above assumptions (A.1) and (A.2), then it follows that*

$$
\mathcal{D}_3(\widetilde{G}_n, \widetilde{G}_*) = \mathcal{O}_P(\sqrt{\log(n)/n}).
$$

Proof of Theorem 4.3 is in Appendix B.3. This theorem indicates that the rates for estimating $W_{*,1}\boldsymbol{p}_{*,i}, W_{*,2}\boldsymbol{p}_{*,i}$ are of orders $\mathcal{O}_P(\sqrt{\log(n)/n})$ and $\mathcal{O}_P(\sqrt[4]{\log(n)/n})$ if $|\mathcal{V}_{j'}| = 1$ and $|\mathcal{V}_{j'}| > 1$, respectively. Furthermore, let $\widetilde{W}_{n,1}\widetilde{\boldsymbol{p}}_{n,i}$ and $\widetilde{W}_{n,2}\widetilde{\boldsymbol{p}}_{n,i}$ be estimators of $W_{*,1}\boldsymbol{p}_{*,j'}$ and $W_{*,2}\boldsymbol{p}_{*,j'}$, respectively. Since the activation functions $\bar{\sigma}_1$ and $\bar{\sigma}_1$ are Lipschitz continuous, that is,

$$
\|\bar{\sigma}_\ell(\widetilde{W}_{n,\ell}\widetilde{\boldsymbol{p}}_{n,i}) - \bar{\sigma}_\ell(W_{*,\ell}\boldsymbol{p}_{*,j'})\| \lesssim \|\widetilde{W}_{n,\ell}\widetilde{\boldsymbol{p}}_{n,i} - W_{*,\ell}\boldsymbol{p}_{*,j'}\|, \quad \text{for any } \ell \in \{1, 2\}
$$

we deduce that the prompts $\boldsymbol{p}_{*,j'}^K = \bar{\sigma}_1(W_{*,1}\boldsymbol{p}_{*,j'})$ and $\boldsymbol{p}_{*,j'}^V = \bar{\sigma}_1(W_{*,2}\boldsymbol{p}_{*,j'})$ admit the same estimation rates as those of $W_{*,1}\boldsymbol{p}_{*,i}$ and $W_{*,2}\boldsymbol{p}_{*,i}$. Note that these rates are significantly faster than those in Theorem 4.1 where the prompts does not share their inner structures, which could be as slow as $\mathcal{O}(1/\log(n))$. This observation together with that from Theorem 4.2 demonstrate that the reparametrization among prompts under both the simple linear setting and the one-layer neural network setting helps improve the sample efficiency of prompt learning considerably.

## 5 EXPERIMENTS

### 5.1 EXPERIMENTAL SETUP

In our experiments on visual and language tasks, we follow the settings of Jia et al. (2022) and Li & Liang (2021), respectively. Please refer to Appendix E for further details.

Table 1: Comparison of prefix-tuning with and without reparameterization on FGVC and VTAB-1K benchmarks. We report the average accuracy over five independent runs. Best results among all methods except Finetune are **bolded**.

| Method | FGVC | | | | | | VTAB-1K | | |
|---|---|---|---|---|---|---|---|---|---|
| | Mean Acc | CUB-200-2011 | NABirds | Oxford Flowers | Stanford Dogs | Stanford Cars | Natural | Specialized | Structured |
| Finetune | 88.54 | 87.3 | 82.7 | 98.8 | 89.4 | 84.5 | 75.88 | 83.36 | 47.64 |
| Deep-share$_{\text{SHALLOW}}$ | 84.36 | 87.2 | 81.5 | **98.6** | 91.1 | 63.4 | 75.79 | 79.48 | 38.53 |
| No-share$_{\text{SHALLOW}}$ | 80.38 | 85.1 | 77.8 | 97.9 | 86.4 | 54.7 | 69.00 | 77.20 | 29.65 |
| Deep-share$_{\text{DEEP}}$ | **88.28** | **87.8** | **84.5** | 98.2 | **91.6** | **79.3** | **77.06** | **82.28** | **52.00** |
| No-share$_{\text{DEEP}}$ | 82.32 | 85.9 | 79.0 | 97.9 | 86.3 | 62.5 | 70.29 | 80.20 | 37.69 |

Table 2: Comparison of prefix-tuning with and without reparameterization on language datasets including E2E, WebNLG, and XSUM. Best results among all methods except Finetune are **bolded**.

| Method | E2E | | | | | WebNLG | | | | | | | | | XSUM | | |
|---|---|---|---|---|---|---|---|---|---|---|---|---|---|---|---|---|---|
| | BLEU | NIST | MET | R-L | CIDEr | BLEU | | | MET | | | TER ↓ | | | R-1 | R-2 | R-L |
| | | | | | | S | U | A | S | U | A | S | U | A | | | |
| Finetune | 68.2 | 8.62 | 46.2 | 71.0 | 2.47 | 64.2 | 27.7 | 46.5 | 0.45 | 0.30 | 0.38 | 0.33 | 0.76 | 0.53 | 45.14 | 22.27 | 37.25 |
| Deep-share | 69.9 | 8.78 | 46.3 | 71.5 | 2.45 | 63.9 | 44.3 | 54.5 | 0.45 | 0.36 | 0.41 | 0.34 | 0.52 | 0.42 | 42.62 | 19.66 | 34.36 |
| No-share | 68.0 | 8.61 | 45.8 | 71.0 | 2.41 | 61.1 | 42.8 | 53.5 | 0.43 | 0.35 | 0.40 | 0.36 | 0.49 | 0.42 | 36.86 | 15.16 | 29.89 |

**Datasets and metrics.** For visual tasks, we use the FGVC and VTAB-1K (Zhai et al., 2019) benchmarks. FGVC includes five Fine-Grained Visual Classification datasets: CUB-200-2011 (Wah et al., 2011), NABirds (Van Horn et al., 2015), Oxford Flowers (Nilsback & Zisserman, 2008), Stanford Dogs (Khosla et al., 2011), and Stanford Cars (Gebru et al., 2017). VTAB-1K comprises 19 visual tasks in three categories: Natural (standard camera images), Specialized (specialized equipment images), and Structured (tasks requiring structural reasoning like 3D depth prediction). We report accuracy on the test set. For language tasks, we assess performance in table-to-text generation and summarization. We evaluate table-to-text generation with E2E (Novikova et al., 2017) and WebNLG (Gardent et al., 2017) datasets, using BLEU (Papineni et al., 2002), NIST (Belz & Reiter, 2006), METEOR (Banerjee & Lavie, 2004), ROUGE-L (Lin, 2004), CIDEr (Vedantam et al., 2015), and TER (Snover et al., 2005). Summarization is assessed with the XSUM dataset (Narayan et al., 2018) using ROUGE-1, ROUGE-2, and ROUGE-L. Table 3 summarizes the metrics for each dataset.

**Baselines.** To assess the effectiveness of the shared structure, we evaluate prefix-tuning under the following configurations: *Deep-share*: uses prefix-tuning with the reparameterization trick; *No-share*: applies prefix-tuning without reparameterization, with prefix key and value vectors as independent parameters; *Simple-share*: similar to *Deep-share*, but with $\sigma_1$ and $\sigma_2$ as the identity function (see Section 3). Additionally, following Jia et al. (2022), we explore two variants: SHALLOW, where prompts attach only to the first layer, and DEEP, where prompts are attached to all layers. Unless otherwise specified, references to prefix-tuning denote the DEEP variant. We also compare prefix-tuning with several fine-tuning techniques: *Finetune*: updates all backbone model parameters; *Partial-k*: fine-tunes only the last $k$ layers of the backbone while freezing the others; *Adapter* (Houlsby et al., 2019; Lin et al., 2020): inserts new MLP modules with residual connections into the Transformer layers; *VPT* (Jia et al., 2022): designed for visual tasks, integrates learnable prompts into the input space of Transformer layers, following prompt-tuning approach.

**Pre-trained backbones.** We use the Vision Transformer (ViT-B/16) (Dosovitskiy et al., 2021), pre-trained on ImageNet-21K (Deng et al., 2009), for visual tasks. For table-to-text, we utilize GPT2$_{\text{MEDIUM}}$ (Radford et al., 2019), with linearized input tables. For summarization, we employ BART$_{\text{LARGE}}$ (Lewis, 2019), truncating source articles to 512 BPE tokens.

## 5.2 MAIN RESULTS

Tables 1 and 2 present the performance of prefix-tuning with and without reparameterization. Detailed per-task results for VTAB-1K are provided in Appendix F.

**Prefix-tuning with reparameterization can achieve competitive performance with full fine-tuning.** As shown in Table 1, although prefix-tuning has not been widely explored for visual tasks, our results indicate that Deep-share$_{\text{DEEP}}$ performs comparably to full fine-tuning, surpassing it in 2 out of 4 problem classes (13 out of 24 tasks). For instance, prefix-tuning achieved 91.6% accuracy

on Stanford Dogs, surpassing full fine-tuning by 2.2%, and 52% accuracy on VTAB-1K Structured, exceeding fine-tuning by 4.36%. While it underperformed on more challenging tasks like Stanford Cars, Deep-share$_{\text{DEEP}}$ still achieved a comparable average accuracy (88.28% vs. 88.54%). Similar trends are observed for language tasks, as shown in Table 2. On E2E, prefix-tuning outperformed fine-tuning across most metrics, though it slightly lagged in the XSUM summarization task.

**Reparameterization plays a crucial role in enhancing the effectiveness of prefix-tuning.** It can be observed that the performance significantly declines when the reparameterization strategy is omitted. As shown in Table 1, Deep-share outperforms No-share by a substantial margin across both variants, DEEP and SHALLOW. For instance, on Stanford Cars, Deep-share$_{\text{DEEP}}$ exceeds No-share$_{\text{DEEP}}$ by 16.8%. This trend is consistent across the majority of datasets (22 out of 24 tasks), underscoring the effectiveness of reparameterization in improving prefix-tuning performance. This empirical finding aligns with our theoretical results presented in Section 4, which demonstrate that reparameterization significantly enhances sample efficiency in parameter estimation. These trends persist across both visual and language tasks. In Table 2, Deep-share surpasses No-share on most metrics across three datasets. For example, in summarization tasks, Deep-share outperforms No-share on all metrics by a considerable margin. This illustrates the critical role of reparameterization in enabling prefix-tuning to achieve competitive performance.

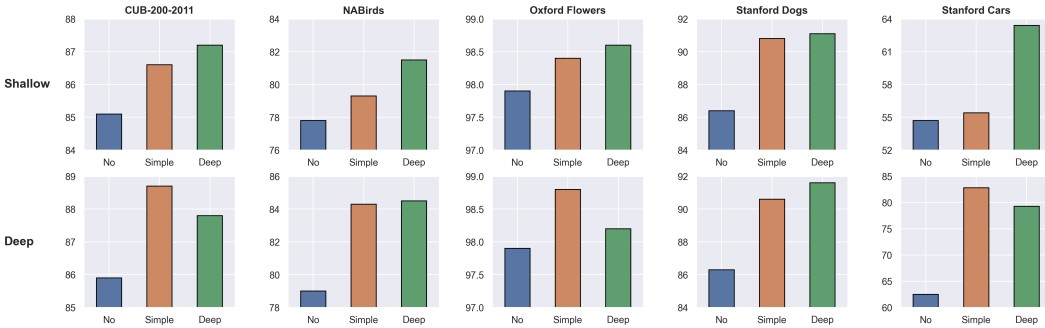

Figure 2: Comparison of prefix-tuning across three configurations: Deep-share, Simple-share, and No-share, referred to as Deep, Simple, and No, respectively, on FGVC benchmarks.

**The shared structure significantly improves prefix-tuning performance.** To further assess the impact of the shared structure, we compare prefix-tuning under the Simple-share configuration, where $\sigma_1$ and $\sigma_2$ are identity functions. As discussed in Section 4.2, our theoretical analysis suggests that both Deep-share and Simple-share substantially outperform the No-share baseline. These findings are consistent with our empirical results, as shown in Figure 2. Across all FGVC datasets, both Simple-share and Deep-share consistently yield significantly better performance than No-share. This consistent improvement demonstrates the empirical effectiveness of shared structures in enhancing prefix-tuning performance. For further experimental results, see Appendix F.

## 6 DISCUSSION AND CONCLUSION

In this paper, we offer theoretical insights into the reparameterization strategy employed in prefix-tuning, which is often regarded as an engineering technique. We demonstrate that reparameterization induces a shared structure between the prefix key and value vectors, which significantly enhances sample efficiency during prompt estimation. Beyond the theoretical analysis, we empirically validate the advantages of this shared structure through experiments across both vision and language tasks. However, the current reparameterization implementation, which relies on an MLP to generate prefix vectors during training, introduces a potential memory overhead. Future work could focus on optimizing this implementation to reduce such overhead. Additionally, while our focus is on prefix-tuning, we propose that the benefits of the shared structure may extend to other parameter-efficient fine-tuning techniques, such as LoRA. We also identify similar patterns of shared structure in prompt-tuning, offering a preliminary investigation into the underlying mechanisms contributing to its effectiveness. However, our study is limited to pre-trained MoE models in the context of prompt-tuning, serving as an initial exploration. Future research could explore the influence of newly introduced MoE models and the interactions between these models.

## REPRODUCIBILITY STATEMENT

In order to facilitate the reproduction of our empirical results, we provide detailed descriptions of the experimental setup in Section 5.1 and Appendix E. All datasets used in this study are publicly available, enabling full replication of our experiments.

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

# Supplement to "Revisiting Prefix-tuning: Statistical Benefits of Reparametrization among Prompts"

In this supplementary material, we begin by exploring the relationship between prompt-tuning and mixture of experts in Appendix A. Following this, we provide detailed proofs for the theoretical results discussed in Section 4. Additionally, we present an in-depth discussion of related work in Appendix D. Appendix E offers further implementation details for the experiments outlined in Section 5. Finally, Appendix F includes additional experimental results.

## A  PROMPT-TUNING AND MIXTURE OF EXPERTS

We demonstrate that applying prompt-tuning not only fine-tunes pre-trained MoE models by incorporating new experts but also facilitates the introduction of entirely new MoE models within the attention mechanism. Specifically, similar to Section 2.2, we consider the $l$-th head within the MSA layer. Let $\boldsymbol{P} = \left[\boldsymbol{p}_1, \ldots, \boldsymbol{p}_{N_p}\right]^\top \in \mathbb{R}^{N_p \times d}$. We define new experts $f_{N+j} : \mathbb{R}^{Nd} \to \mathbb{R}^{d_v}$ along with their corresponding new score functions $s_{i,N+j} : \mathbb{R}^{Nd} \to \mathbb{R}$ for pre-trained MoE models as follows:

$$f_{N+j}(\boldsymbol{X}) := W_l^{V^\top} \boldsymbol{p}_j, \quad s_{i,N+j}(\boldsymbol{X}) := \frac{\boldsymbol{X}^\top E_i^\top W_l^Q W_l^{K^\top} \boldsymbol{p}_j}{\sqrt{d_v}} = \frac{\boldsymbol{x}_i^\top W_l^Q W_l^{K^\top} \boldsymbol{p}_j}{\sqrt{d_v}} \tag{17}$$

for $i \in [N]$ and $j \in [N_p]$. For $N_p$ new MoE models, we define the score functions $s_{N+i,\,j} : \mathbb{R}^{Nd} \to \mathbb{R}$ associated with pre-trained experts as:

$$s_{N+i,\,j}(\boldsymbol{X}) := \frac{\boldsymbol{p}_i^\top W_l^Q W_l^{K^\top} E_j \boldsymbol{X}}{\sqrt{d_v}} = \frac{\boldsymbol{p}_i^\top W_l^Q W_l^{K^\top} \boldsymbol{x}_j}{\sqrt{d_v}}, \tag{18}$$

for $i \in [N_p]$ and $j \in [N]$. The score functions $s_{N+i,N+j} : \mathbb{R}^{Nd} \to \mathbb{R}$ for new experts within new MoE models are defined as:

$$s_{N+i,N+j}(\boldsymbol{X}) := \frac{\boldsymbol{p}_i^\top W_l^Q W_l^{K^\top} \boldsymbol{p}_j}{\sqrt{d_v}}, \tag{19}$$

for $i \in [N_p]$ and $j \in [N_p]$. Then from equation (3), the output of the $l$-th head can be expressed as:

$$\hat{\boldsymbol{h}}_l = \text{Attention}\left(\begin{bmatrix} \boldsymbol{P} \\ \boldsymbol{X}_Q \end{bmatrix}, \begin{bmatrix} \boldsymbol{P} \\ \boldsymbol{X}_K \end{bmatrix}, \begin{bmatrix} \boldsymbol{P} \\ \boldsymbol{X}_V \end{bmatrix}\right) = \left[\hat{\boldsymbol{h}}_{l,1}, \ldots, \hat{\boldsymbol{h}}_{l,N+N_p}\right]^\top \in \mathbb{R}^{(N+N_p) \times d_v}, \tag{20}$$

$$\hat{\boldsymbol{h}}_{l,i} = \sum_{j=1}^{N} \frac{\exp(s_{i,j}(\boldsymbol{X}))}{\sum_{k=1}^{N} \exp(s_{i,k}(\boldsymbol{X})) + \sum_{k'=1}^{N_p} \exp(s_{i,N+k'}(\boldsymbol{X}))} f_j(\boldsymbol{X})$$

$$+ \sum_{j'=1}^{N_p} \frac{\exp(s_{i,N+j'}(\boldsymbol{X}))}{\sum_{k=1}^{N} \exp(s_{i,k}(\boldsymbol{X})) + \sum_{k'=1}^{N_p} \exp(s_{i,N+k'}(\boldsymbol{X}))} f_{N+j'}(\boldsymbol{X}), \tag{21}$$

for $i \in [N + N_p]$. Prompt-tuning extends pre-trained MoE models by incorporating $N_p$ additional experts $f_{N+j}$, which are defined by the prompt vectors $\boldsymbol{p}_j$. Additionally, prompt-tuning introduces new MoE models, $\hat{\boldsymbol{h}}_{l,N+1}, \ldots, \hat{\boldsymbol{h}}_{l,N+N_p}$, that utilize linear and scalar score functions.

## B  PROOFS

### B.1  PROOF OF THEOREM 4.1

The proof is divided into two step as follows:

**Step 1.** To begin with, we demonstrate that the following limit holds true for any $r \geq 1$:

$$\lim_{\varepsilon \to 0} \inf_{G \in \mathcal{G}_{L'}(\Theta) : \mathcal{D}_{1,r}(G,G_*) \leq \varepsilon} \frac{\|f_G - f_{G_*}\|_{L^2(\mu)}}{\mathcal{D}_{1,r}(G, G_*)} = 0. \tag{22}$$

Note that it is sufficient to construct a mixing measure sequence $(G_n)_{n \geq 1}$ that satisfies both $\mathcal{D}_{1,r}(G_n, G_*) \to 0$ and $\|f_{G_n} - f_{G_*}\|_{L^2(\mu)}/\mathcal{D}_{1,r}(G_n, G_*) \to 0$, as $n \to \infty$.

For that purpose, we take into account the sequence $G_n = \sum_{i=1}^{L+1} \exp(b_{n,i}) \delta_{(\boldsymbol{p}_{n,i}^K, \boldsymbol{p}_{n,i}^V)}$, where

- $\exp(b_{n,1}) = \exp(b_{n,2}) = \frac{1}{2}\exp(b_{*,1}) + \frac{1}{2n^{r+1}}$ and $\exp(b_{n,i}) = \exp(b_{n,i-1})$ for any $3 \leq i \leq L + 1$;
- $\boldsymbol{p}_{n,1}^K = \boldsymbol{p}_{n,2}^K = \boldsymbol{p}_{*,1}^K$ and $\boldsymbol{p}_{n,i}^K = \boldsymbol{p}_{n,i-1}^K$ for any $3 \leq i \leq L + 1$;
- $\boldsymbol{p}_{n,1}^V = \boldsymbol{p}_{*,1}^V + \frac{1}{n}(1, 0, \ldots, 0)$, $\boldsymbol{p}_{n,2}^V = \boldsymbol{p}_{*,1}^V - \frac{1}{n}(1, 0, \ldots, 0)$ and $\boldsymbol{p}_{n,i}^V = \boldsymbol{p}_{*,i-1}^V$ for any $3 \leq i \leq L + 1$.

Then, we can compute the loss function $\mathcal{D}_{1,r}(G_n, G_*)$ as

$$\mathcal{D}_{1,r}(G_n, G_*) = \frac{1}{n^{r+1}} + \left[\exp(b_{*,1}) + \frac{1}{n^{r+1}}\right] \cdot \frac{1}{n^r} = \mathcal{O}(n^{-r}). \tag{23}$$

It can be seen that $\mathcal{D}_{1,r}(G_n, G_*) \to 0$ as $n \to \infty$.

Subsequently, we illustrate that $\|f_{G_n} - f_{G_*}\|_{L^2(\mu)}/\mathcal{D}_{1,r}(G_n, G_*) \to 0$. In particular, let us consider the quantity

$$Q_n(\boldsymbol{X}) := \left[\sum_{i'=1}^N \exp(\boldsymbol{X}^\top A_{i'}^0 \boldsymbol{X} + a_{i'}^0) + \sum_{j'=1}^L \exp((B\boldsymbol{p}_{*,j'}^K)^\top \boldsymbol{X} + b_{*,j'})\right] \cdot [f_{G_n}(\boldsymbol{X}) - f_{G_*}(\boldsymbol{X})],$$

which can be decomposed as follows:

$$Q_n(\boldsymbol{X}) = \sum_{j=1}^L \sum_{i \in \mathcal{V}_j} \exp(b_{n,i}) \left[\exp((B\boldsymbol{p}_{n,i}^K)^\top \boldsymbol{X})C\boldsymbol{p}_{n,i}^V - \exp((B\boldsymbol{p}_{*,j}^K)^\top \boldsymbol{X})C\boldsymbol{p}_{*,j}^V\right]$$

$$- \sum_{j=1}^L \sum_{i \in \mathcal{V}_j} \exp(b_{n,i}) \left[\exp((B\boldsymbol{p}_{n,i}^K)^\top \boldsymbol{X})f_{G_n}(\boldsymbol{X}) - \exp((B\boldsymbol{p}_{*,j}^K)^\top \boldsymbol{X})f_{G_n}(\boldsymbol{X})\right]$$

$$+ \sum_{j=1}^L \left(\sum_{i \in \mathcal{V}_j} \exp(b_{n,i}) - \exp(b_{*,j})\right) \left[\exp((B\boldsymbol{p}_{*,j}^K)^\top \boldsymbol{X})C\boldsymbol{p}_{*,j}^V - \exp((B\boldsymbol{p}_{*,j}^K)^\top \boldsymbol{X})f_{G_n}(\boldsymbol{X})\right]$$

$$:= A_n(\boldsymbol{X}) - B_n(\boldsymbol{X}) + C_n(\boldsymbol{X}).$$

It follows from the choices of $\boldsymbol{p}_{n,i}^K, \boldsymbol{p}_{n,i}^V$ and $b_{n,i}$ that

$$A_n(\boldsymbol{X}) = \sum_{i=1}^2 \frac{1}{2}\left[\exp(b_{*,1}) + \frac{1}{n^{r+1}}\right]\exp((B\boldsymbol{p}_{*,1}^K)^\top \boldsymbol{X})C(\boldsymbol{p}_{n,i}^V - \boldsymbol{p}_{*,1}^V)]$$

$$= \frac{1}{2}\left[\exp(b_{*,1}) + \frac{1}{n^{r+1}}\right]\exp((\boldsymbol{p}_{*,1}^K)^\top \boldsymbol{X})C[(\boldsymbol{p}_{n,1}^V - \boldsymbol{p}_{*,1}^V) + (\boldsymbol{p}_{n,2}^V - \boldsymbol{p}_{*,1}^V)]$$

$$= 0.$$

Moreover, we can also verify that $B_n(\boldsymbol{X}) = 0$, and $C_n(\boldsymbol{X}) = \mathcal{O}(n^{-(r+1)})$. Thus, we deduce that $Q_n(\boldsymbol{X})/\mathcal{D}_{1,r}(G_n, G_*) \to 0$ as $n \to \infty$ for almost every $\boldsymbol{X}$.

As the term $\left[\sum_{i'=1}^N \exp(\boldsymbol{X}^\top A_{i'}^0 \boldsymbol{X} + a_{i'}^0) + \sum_{j'=1}^L \exp((\boldsymbol{p}_{*,j'}^K)^\top \boldsymbol{X} + b_{*,j'})\right]$ is bounded, we have $[f_{G_n}(\boldsymbol{X}) - f_{G_*}(\boldsymbol{X})]/\mathcal{D}_{1,r}(G_n, G_*) \to 0$ for almost every $\boldsymbol{X}$. This limit suggests that

$$\|f_{G_n} - f_{G_*}\|_{L^2(\mu)}/\mathcal{D}_{1,r}(G_n, G_*) \to 0$$

as $n \to \infty$. Thus, we obtain the claim in equation (22).

**Step 2.** We will establish the desired result in this step, that is,

$$\inf_{\overline{G}_n \in \mathcal{G}_{L'}(\Theta)} \sup_{G \in \mathcal{G}_{L'}(\Theta) \backslash \mathcal{G}_{L-1}(\Theta)} \mathbb{E}_{f_G}[\mathcal{D}_{1,r}(\overline{G}_n, G)] \gtrsim n^{-1/2}. \tag{24}$$

Since the noise variables $\epsilon_i$ follow from the Gaussian distribution, we get that $Y_i|\boldsymbol{X}_i \sim \mathcal{N}(f_{G_*}(\boldsymbol{X}_i), \sigma^2)$ for all $i \in [n]$. Additionally, for sufficiently small $\varepsilon > 0$ and a fixed constant $C_1 > 0$ which we will select later, we can find a mixing measure $G_*' \in \mathcal{G}_{L'}(\Theta)$ such that $\mathcal{D}_{1,r}(G_*', G_*) = 2\varepsilon$ and $\|f_{G_*'} - f_{G_*}\|_{L^2(\mu)} \leq C_1\varepsilon$ thanks to the result in equation (22). According

to the Le Cam's lemma (Yu, 1997), as the Voronoi loss function $\mathcal{D}_{1,r}$ satisfies the weak triangle inequality, it follows that

$$
\begin{aligned}
\inf_{\overline{G}_n \in \mathcal{G}_{L'}(\Theta)} \sup_{G \in \mathcal{G}_{L'}(\Theta) \setminus \mathcal{G}_{L-1}(\Theta)} & \mathbb{E}_{f_G}[\mathcal{D}_{1,r}(\overline{G}_n, G)] \\
& \gtrsim \frac{\mathcal{D}_{1,r}(G'_*, G_*)}{8} \exp(-n\mathbb{E}_{\boldsymbol{X} \sim \mu}[\mathrm{KL}(\mathcal{N}(f_{G'_*}(\boldsymbol{X}), \sigma^2), \mathcal{N}(f_{G_*}(\boldsymbol{X}), \sigma^2))]) \\
& \gtrsim \varepsilon \cdot \exp(-n\|f_{G'_*} - f_{G_*}\|_{L^2(\mu)}^2) \\
& \gtrsim \varepsilon \cdot \exp(-C_1 n\varepsilon^2),
\end{aligned}
\tag{25}
$$

where the second inequality follows from the equality

$$
\mathrm{KL}(\mathcal{N}(f_{G'_*}(\boldsymbol{X}), \sigma^2), \mathcal{N}(f_{G_*}(\boldsymbol{X}), \sigma^2)) = \frac{(f_{G'_*}(\boldsymbol{X}) - f_{G_*}(\boldsymbol{X}))^2}{2\sigma^2}.
$$

Let $\varepsilon = n^{-1/2}$, then we get that $\varepsilon \cdot \exp(-C_1 n\varepsilon^2) = n^{-1/2} \exp(-C_1)$. Consequently, we achieve the desired minimax lower bound in equation (24).

## B.2 PROOF OF THEOREM 4.2

The proof of Theorem 4.2 consists of two parts. In the first part in Section B.2.1, we prove the parametric convergence rate $\mathcal{O}_P(\sqrt{\log(n)/n})$ of the estimated regression function $f_{\bar{G}_n}$ to the true regression function $f_{\bar{G}_*}$. In the second part in Section B.2.2, we establish the lower bound $\|f_{\bar{G}} - f_{\bar{G}_*}\|_{L^2(\mu)} \geq C' \mathcal{D}_2(\bar{G}, \bar{G}_*)$ for any $\bar{G} \in \bar{\mathcal{G}}_{L'}(\Omega)$ for some universal constant $C'$. This lower bound directly translates to the convergence rate $\mathcal{O}_P(\sqrt{\log(n)/n})$ of the least-square estimator $\bar{G}_n$ to the true mixing measure $\bar{G}_*$.

### B.2.1 CONVERGENCE RATE OF DENSITY ESTIMATION

**Proposition B.1.** *The convergence rate of the model estimation $f_{\bar{G}_n}(\cdot)$ to the true model $f_{\bar{G}_*}(\cdot)$ under the $L^2(\mu)$ norm is parametric on the sample size, that is,*

$$
\|f_{\bar{G}_n} - f_{\bar{G}_*}\|_{L^2(\mu)} = \mathcal{O}_P(\sqrt{\log(n)/n}).
\tag{26}
$$

Proof of Proposition B.1 is in Appendix C.1.

### B.2.2 FROM DENSITY ESTIMATION TO EXPERT ESTIMATION

Given the parametric convergence rate of the estimated regression function $f_{\bar{G}_n}$ to the true regression function $f_{\bar{G}_*}$ in Proposition B.1, to obtain the conclusion of Theorem 4.2, it is sufficient to demonstrate that $\|f_{\bar{G}} - f_{\bar{G}_*}\|_{L^2(\mu)} \geq C' \mathcal{D}_2(\bar{G}, \bar{G}_*)$ for any $\bar{G} \in \bar{\mathcal{G}}_{L'}(\Omega)$ for some universal constant $C'$. It is equivalent to demonstrate the following inequality:

$$
\inf_{\bar{G} \in \bar{\mathcal{G}}_{L'}(\Omega)} \|f_{\bar{G}} - f_{\bar{G}_*}\|_{L^2(\mu)} / \mathcal{D}_2(G, \bar{G}_*) > 0.
$$

We divide the proof of the above inequality into local and global parts.

**Local part:** We will demonstrate that

$$
\lim_{\varepsilon \to 0} \inf_{\bar{G} \in \bar{\mathcal{G}}_{L'}(\Omega): \mathcal{D}_2(G, \bar{G}_*) \leq \varepsilon} \|f_{\bar{G}} - f_{\bar{G}_*}\|_{L^2(\mu)} / \mathcal{D}_2(\bar{G}, \bar{G}_*) > 0
$$

Assume by contrary that the above claim does not hold. Then, there exists a sequence of mixing measures $\bar{G}_n := \sum_{j'=1}^{L'} \exp(b_{n,j'}) \delta_{\boldsymbol{p}_{n,j'}}$ in $\bar{\mathcal{G}}_{L'}(\Omega)$ such that as $n \to \infty$, we have

$$
\begin{cases}
\mathcal{D}_{2n} := \mathcal{D}_2(\bar{G}_n, \bar{G}_*) \to 0, \\
\|f_{\bar{G}_n} - f_{\bar{G}_*}\|_{L^2(\mu)} / \mathcal{D}_{2n} \to 0.
\end{cases}
$$

Denote $\mathcal{V}_j^n := \mathcal{V}_j(\bar{G}_n)$ as a Voronoi cell of $\bar{G}_n$ generated by the $j$-th components of $\bar{G}_*$. Since our arguments are asymptotic, we may assume that those Voronoi cells do not depend on the sample size, i.e., $\mathcal{V}_j = \mathcal{V}_j^n$. Thus, the Voronoi loss $\mathcal{D}_{2n}$ can be represented as

$$\mathcal{D}_{2n} := \sum_{j'=1}^{L} \Big| \sum_{i \in \mathcal{V}_{j'}} \exp(b_{n,i}) - \exp(b_{*,j'}) \Big| + \sum_{j' \in [L]:|\mathcal{V}_{j'}|=1} \sum_{i \in \mathcal{V}_{j'}} \exp(b_{n,i}) \|\Delta \boldsymbol{p}_{n,ij'}\|$$
$$+ \sum_{j' \in [L]:|\mathcal{V}_{j'}|>1} \sum_{i \in \mathcal{V}_{j'}} \exp(b_{n,i}) \|\Delta \boldsymbol{p}_{n,ij'}\|^2,$$

where $\Delta \boldsymbol{p}_{n,ij'} = \boldsymbol{p}_{n,i} - \boldsymbol{p}_{*,j'}$ for all $i \in \mathcal{V}_{j'}$.

Additionally, since $\mathcal{D}_{2n} \to 0$, we have $\sum_{i \in \mathcal{V}_j} \exp(b_{n,i}) \to \exp(b_{*,j})$ and $\boldsymbol{p}_{n,i} \to \boldsymbol{p}_{*,j}$ for any $i \in \mathcal{V}_j, j \in [L]$. Now, we divide the proof of the local part into three sub-steps as follows.

**Step 1 - Taylor expansion.** In this step, we would like to decompose the quantity

$$Q_n(\boldsymbol{X}) := \Big[ \sum_{j=1}^{N} \exp(\boldsymbol{X}^\top A_j^0 \boldsymbol{X} + a_j^0) + \sum_{j'=1}^{L} \exp((B\boldsymbol{p}_{*,j'})^\top \boldsymbol{X} + b_{*,j'}) \Big] \cdot [f_{\bar{G}_n}(\boldsymbol{X}) - f_{\bar{G}_*}(\boldsymbol{X})],$$

as follows:

$$Q_n(\boldsymbol{X}) = \sum_{j=1}^{L} \sum_{i \in \mathcal{V}_j} \exp(b_{n,i}) \Big[ \exp((B\boldsymbol{p}_{n,i})^\top \boldsymbol{X}) C\boldsymbol{p}_{n,i} - \exp((B\boldsymbol{p}_{*,j})^\top \boldsymbol{X}) C\boldsymbol{p}_{*,j} \Big]$$
$$- \sum_{j=1}^{L} \sum_{i \in \mathcal{V}_j} \exp(b_{n,i}) \Big[ \exp((B\boldsymbol{p}_{n,i})^\top \boldsymbol{X}) - \exp((B\boldsymbol{p}_{*,j})^\top \boldsymbol{X}) \Big] f_{\bar{G}_n}(\boldsymbol{X})$$
$$+ \sum_{j=1}^{L} \Big( \sum_{i \in \mathcal{V}_j} \exp(b_{n,i}) - \exp(b_{*,j}) \Big) \exp((B\boldsymbol{p}_{*,j})^\top \boldsymbol{X}) \Big[ C\boldsymbol{p}_{*,j} - f_{\bar{G}_n}(\boldsymbol{X}) \Big]$$
$$:= \bar{A}_n(\boldsymbol{X}) - \bar{B}_n(\boldsymbol{X}) + \bar{C}_n(\boldsymbol{X}). \tag{27}$$

**Decomposition of $\bar{A}_n(\boldsymbol{X})$.** To ease the ensuing presentation, we denote $E(\boldsymbol{X}; \boldsymbol{p}) := \exp((B\boldsymbol{p})^\top \boldsymbol{X})$ and $H(\boldsymbol{p}) = C\boldsymbol{p}$, and $F(\boldsymbol{X}; \boldsymbol{p}) = E(\boldsymbol{X}; \boldsymbol{p}) H(\boldsymbol{p})$. Since each Voronoi cell $\mathcal{V}_j$ possibly has more than one element, we continue to decompose $\bar{A}_n$ as follows:

$$\bar{A}_n(\boldsymbol{X}) = \sum_{j:|\mathcal{V}_j|=1} \sum_{i \in \mathcal{V}_j} \exp(b_{n,i}) \Big[ F(\boldsymbol{X}; \boldsymbol{p}_{n,i}) - F(\boldsymbol{X}; \boldsymbol{p}_{*,j}) \Big]$$
$$+ \sum_{j:|\mathcal{V}_j|>1} \sum_{i \in \mathcal{V}_j} \exp(b_{n,i}) \Big[ F(\boldsymbol{X}; \boldsymbol{p}_{n,i}) - F(\boldsymbol{X}; \boldsymbol{p}_{*,j}) \Big]$$
$$:= \bar{A}_{n,1}(\boldsymbol{X}) + \bar{A}_{n,2}(\boldsymbol{X}).$$

By means of the first-order Taylor expansion, we have

$$E(\boldsymbol{X}; \boldsymbol{p}_{n,i}) = E(\boldsymbol{X}; \boldsymbol{p}_{*,j}) + \sum_{|\alpha|=1} (\Delta \boldsymbol{p}_{n,ij})^\alpha \frac{\partial^{|\alpha|} E}{\partial \boldsymbol{p}^\alpha}(\boldsymbol{X}; \boldsymbol{p}_{*,j}) + R_{ij,1}(\boldsymbol{X}),$$

$$H(\boldsymbol{p}_{n,i}) = H(\boldsymbol{p}_{*,j}) + \sum_{|\alpha|=1} (\Delta \boldsymbol{p}_{n,ij})^\alpha \frac{\partial^{|\alpha|} H}{\partial \boldsymbol{p}^\alpha}(\boldsymbol{p}_{*,j}) + R_{ij,2},$$

for any $i \in \mathcal{V}_j$ and $j$ such that $|\mathcal{V}_j| = 1$. Here, $R_{ij,1}(\boldsymbol{X})$ and $R_{ij,2}$ are Taylor remainders. Putting the above results together leads to

$$
\begin{aligned}
\bar{A}_{n,1}(\boldsymbol{X}) &= \sum_{j:|\mathcal{V}_j|=1} \sum_{i \in \mathcal{V}_j} \frac{\exp(b_{n,i})}{\alpha!} \sum_{|\alpha|=1} \left\{ (\Delta\boldsymbol{p}_{n,ij})^\alpha \frac{\partial^{|\alpha|} E}{\partial \boldsymbol{p}^\alpha}(\boldsymbol{X}; \boldsymbol{p}_{*,j}) H(\boldsymbol{p}_{*,j}) \right. \\
&\quad \left. + (\Delta\boldsymbol{p}_{n,ij})^\alpha \frac{\partial^{|\alpha|} H}{\partial \boldsymbol{p}^\alpha}(\boldsymbol{p}_{*,j}) E(\boldsymbol{X}; \boldsymbol{p}_{*,j}) \right\} + \bar{R}_{n,1}(\boldsymbol{X}) \\
&= \sum_{j:|\mathcal{V}_j|=1} \sum_{|\alpha|=1} \left\{ M_{n,j,\alpha} \frac{\partial^{|\alpha|} E}{\partial \boldsymbol{p}^\alpha}(\boldsymbol{X}; \boldsymbol{p}_{*,j}) H(\boldsymbol{p}_{*,j}) \right. \\
&\quad \left. + M_{n,j,\alpha} \frac{\partial^{|\alpha|} H}{\partial \boldsymbol{p}^\alpha}(\boldsymbol{p}_{*,j}) E(\boldsymbol{X}; \boldsymbol{p}_{*,j}) \right\} + \bar{R}_{n,1}(\boldsymbol{X})
\end{aligned}
$$

where the function $\bar{R}_{n,1}(\boldsymbol{X})$ satisfies $\bar{R}_{n,1}(\boldsymbol{X})/\mathcal{D}_{2n} \to 0$ when $n \to \infty$. Furthermore, the formulations of $M_{n,j,\alpha}$ are given by:

$$
M_{n,j,\alpha} = \sum_{i \in \mathcal{V}_j} \frac{\exp(b_{n,i})}{\alpha!} (\Delta\boldsymbol{p}_{n,ij})^\alpha,
$$

for any $|\alpha| = 1$.

Moving to the term $\bar{A}_{n,2}(\boldsymbol{X})$, by applying the second-order Taylor expansions to $E(\boldsymbol{X}; \boldsymbol{p}_{n,i})$ around $E(\boldsymbol{X}; \boldsymbol{p}_{*,j})$ and $H(\boldsymbol{p}_{n,i})$ around $H(\boldsymbol{p}_{*,j})$ for any $i \in \mathcal{V}_j$ and $j$ such that $|\mathcal{V}_j| > 1$, we get that

$$
\begin{aligned}
\bar{A}_{n,2}(\boldsymbol{X}) &= \sum_{j:|\mathcal{V}_j|>1} \sum_{1 \le |\alpha| \le 2} \left\{ M_{n,j,\alpha} \frac{\partial^{|\alpha|} E}{\partial \boldsymbol{p}^\alpha}(\boldsymbol{X}; \boldsymbol{p}_{*,j}) H(\boldsymbol{p}_{*,j}) \right. \\
&\quad \left. + M_{n,j,\alpha} \frac{\partial^{|\alpha|} H}{\partial \boldsymbol{p}^\alpha}(\boldsymbol{p}_{*,j}) E(\boldsymbol{X}; \boldsymbol{p}_{*,j}) \right\} \\
&\quad + \sum_{|\alpha|=1,|\beta|=1} M_{n,j,\alpha,\beta} \frac{\partial^{|\alpha|} E}{\partial \boldsymbol{p}^\alpha}(\boldsymbol{X}; \boldsymbol{p}_{*,j}) \frac{\partial^{|\beta|} H}{\partial \boldsymbol{p}^\beta}(\boldsymbol{p}_{*,j}) + \bar{R}_{n,2}(\boldsymbol{X})
\end{aligned}
$$

where the function $\bar{R}_{n,2}(\boldsymbol{X})$ satisfies $\bar{R}_{n,2}(\boldsymbol{X})/\mathcal{D}_{2n} \to 0$ when $n \to \infty$. Furthermore, we define

$$
M_{n,j,\alpha} = \sum_{i \in \mathcal{V}_j} \frac{\exp(b_{n,i})}{\alpha!} (\Delta\boldsymbol{p}_{n,ij})^\alpha,
$$

for any $|\alpha| = 2$ and

$$
M_{n,j,\alpha,\beta} = \sum_{i \in \mathcal{V}_j} \frac{\exp(b_{n,i})}{\alpha!\beta!} (\Delta\boldsymbol{p}_{n,ij})^{\alpha+\beta},
$$

for any $|\alpha| = |\beta| = 1$. Direct calculation leads to the following formulations of the partial derivatives of $E(\boldsymbol{X}; \boldsymbol{p})$ and $H(\boldsymbol{p})$:

$$
\frac{\partial E}{\partial \boldsymbol{p}^{(u)}}(\boldsymbol{X}; \boldsymbol{p}) = \exp((B\boldsymbol{p})^\top \boldsymbol{X})(B1_u)^\top \boldsymbol{X},
$$

$$
\frac{\partial^2 E}{\partial \boldsymbol{p}^{(u)} \partial \boldsymbol{p}^{(v)}}(\boldsymbol{X}; W_1\boldsymbol{p}) = \exp((B\boldsymbol{p})^\top \boldsymbol{X})\boldsymbol{X}^\top (B1_u)(B1_v)^\top \boldsymbol{X},
$$

$$
\frac{\partial H}{\partial \boldsymbol{p}^{(u)}}(\boldsymbol{p}) = C1_u,
$$

$$
\frac{\partial^2 H}{\partial \boldsymbol{p}^{(u)} \partial \boldsymbol{p}^{(v)}}(W_2\boldsymbol{p}) = 0.
$$

Here, we denote $1_u$ is the vector that its $u$-th element is 1 while its other elements are 0 for any $1 \le u \le d$. Given the above formulations, we can rewrite $\bar{A}_{n,1}(\boldsymbol{X})$ and $\bar{A}_{n,2}(\boldsymbol{X})$ as follows:

$$\bar{A}_{n,1}(\boldsymbol{X}) = \sum_{j:|\mathcal{V}_j|=1} \exp((B\boldsymbol{p}_{*,j})^\top \boldsymbol{X})\big[L_{1,n}(\boldsymbol{p}_{*,j}) + L_{2,n}(\boldsymbol{p}_{*,j})^\top B^\top \boldsymbol{X}\big) + \bar{R}_{n,1}(\boldsymbol{X}),$$

$$\bar{A}_{n,2}(\boldsymbol{X}) = \sum_{j:|\mathcal{V}_j|>1} \exp((B\sigma_1(\boldsymbol{p}_{*,j}))^\top \boldsymbol{X})\big[\bar{L}_{1,n}(\boldsymbol{p}_{*,j}) + \bar{L}_{2,n}(\boldsymbol{p}_{*,j})^\top B^\top \boldsymbol{X}$$
$$+ (B^\top \boldsymbol{X})^\top \bar{L}_{3,n}(\boldsymbol{p}_{*,j})B^\top \boldsymbol{X}\big] + \bar{R}_{n,2}(\boldsymbol{X}),$$

where the formulations of the functions $L_{1,n}$, $L_{2,n}$, $\bar{L}_{1,n}$, $\bar{L}_{2,n}$, and $\bar{L}_{3,n}$ are given by:

$$L_{1,n}(\boldsymbol{p}) = \sum_{u=1}^d M_{n,j,1_u} C 1_u,$$

$$L_{2,n}(\boldsymbol{p}) = \sum_{u=1}^d M_{n,j,1_u} 1_u C \boldsymbol{p},$$

$$\bar{L}_{1,n}(\boldsymbol{p}) = \sum_{u=1}^d M_{n,j,1_u} C 1_u,$$

$$\bar{L}_{2,n}(\boldsymbol{p}) = \sum_{u=1}^d M_{n,j,1_u} 1_u C \boldsymbol{p} + \sum_{1\le u,v\le d} M_{n,j,1_v,1_u} C 1_u 1_v$$

$$\bar{L}_{3,n}(\boldsymbol{p}) = \sum_{1\le u,v\le d} M_{n,j,1_{uv}} 1_u 1_v^\top C \boldsymbol{p}.$$

Here, $1_{uv}$ is the matrix that its $(u,v)$-th element is 1 while its other elements are 0 for any $1 \le u,v \le d$.

**Decomposition of $\bar{B}_n(\boldsymbol{X})$.** We can rewrite $\bar{B}_n(\boldsymbol{X})$ as follows:

$$\bar{B}_n(\boldsymbol{X}) = \sum_{j:|\mathcal{V}_j|=1} \sum_{i\in\mathcal{V}_j} \exp(b_{n,i})\Big[E(\boldsymbol{X};\boldsymbol{p}_{n,i}) - E(\boldsymbol{X};\boldsymbol{p}_{*,j})\Big] f_{\bar{G}_n}(\boldsymbol{X})$$
$$+ \sum_{j:|\mathcal{V}_j|>1} \sum_{i\in\mathcal{V}_j} \exp(b_{n,i})\Big[E(\boldsymbol{X};\boldsymbol{p}_{n,i}) - E(\boldsymbol{X};\boldsymbol{p}_{*,j})\Big] f_{\bar{G}_n}(\boldsymbol{X})$$
$$:= \bar{B}_{n,1}(\boldsymbol{X}) + \bar{B}_{n,2}(\boldsymbol{X})$$

By applying the first-order and second-order Taylor expansion, we get

$$\bar{B}_{n,1}(\boldsymbol{X}) = \sum_{j:|\mathcal{V}_j|=1} \sum_{|\alpha|=1} M_{n,j,\alpha} \frac{\partial^{|\alpha|} E}{\partial \boldsymbol{p}^\alpha}(\boldsymbol{X};\boldsymbol{p}_{*,j}) f_{\bar{G}_n}(\boldsymbol{X}) + R_{n,3}(\boldsymbol{X})$$

$$\bar{B}_{n,2}(\boldsymbol{X}) = \sum_{j:|\mathcal{V}_j|=1} \sum_{1\le|\alpha|\le2} M_{n,j,\alpha} \frac{\partial^{|\alpha|} E}{\partial \boldsymbol{p}^\alpha}(\boldsymbol{X};\boldsymbol{p}_{*,j}) f_{\bar{G}_n}(\boldsymbol{X}) + R_{n,4}(\boldsymbol{X})$$

where $R_{n,3}(\boldsymbol{X}), R_{n,4}(\boldsymbol{X})$ is a Taylor remainder such that $R_{n,3}(\boldsymbol{X})/\mathcal{D}_{2n} \to 0$, $R_{n,4}(\boldsymbol{X})/\mathcal{D}_{2n} \to 0$ when $n \to \infty$. Therefore, we can express the functions $\bar{B}_{n,1}(\boldsymbol{X})$ and $\bar{B}_{n,2}(\boldsymbol{X})$ as follows:

$$\bar{B}_{n,1}(\boldsymbol{X}) = \sum_{j:|\mathcal{V}_j|=1} \exp((B\boldsymbol{p}_{*,j})^\top \boldsymbol{X}) N_{1,n}(\boldsymbol{p}_{*,j})^\top \boldsymbol{X} f_{\bar{G}_n}(\boldsymbol{X}) + R_{n,3}(\boldsymbol{X}),$$

$$\bar{B}_{n,2}(\boldsymbol{X}) = \sum_{j:|\mathcal{V}_j|>1} \exp((B\boldsymbol{p}_{*,j})^\top \boldsymbol{X})\big[\bar{N}_{1,n}(\boldsymbol{p}_{*,j})^\top B^\top \boldsymbol{X}$$
$$+ (B^\top \boldsymbol{X})^\top \bar{N}_{2,n}(\boldsymbol{p}_{*,j})(B^\top \boldsymbol{X})\big] f_{\bar{G}_n}(\boldsymbol{X}) + R_{n,4}(\boldsymbol{X}),$$

where the formulations of the functions $N_{1,n}$, $\bar{N}_{1,n}$, and $\bar{N}_{2,n}$ are given by:

$$N_{1,n}(\boldsymbol{p}) = \sum_{u=1}^{d} M_{n,j,1_u} 1_u,$$

$$\bar{N}_{1,n}(\boldsymbol{p}) = \sum_{u=1}^{d} M_{n,j,1_u} 1_u,$$

$$\bar{N}_{2,n}(\boldsymbol{p}) = \sum_{1 \leq u,v \leq d} M_{n,j,1_{uv}} 1_u 1_v^{\top}.$$

Plugging the above expressions into equation (27), we can represent $Q_n(\boldsymbol{X})$ as folows:

$$
\begin{aligned}
Q_n(\boldsymbol{X}) =& \sum_{j:|\mathcal{V}_j|=1} \exp((B\boldsymbol{p}_{*,j})^{\top}\boldsymbol{X})\big[L_{1,n}(\boldsymbol{p}_{*,j}) + L_{2,n}(\boldsymbol{p}_{*,j})^{\top}B^{\top}\boldsymbol{X}\big] \\
&+ \sum_{j:|\mathcal{V}_j|>1} \exp((B\boldsymbol{p}_{*,j})^{\top}\boldsymbol{X})\big[\bar{L}_{1,n}(\boldsymbol{p}_{*,j}) + \bar{L}_{2,n}(\boldsymbol{p}_{*,j})^{\top}B^{\top}\boldsymbol{X} + (B^{\top}\boldsymbol{X})^{\top}\bar{L}_{3,n}(\boldsymbol{p}_{*,j})B^{\top}\boldsymbol{X}\big] \\
&- \sum_{j:|\mathcal{V}_j|=1} \exp((B\boldsymbol{p}_{*,j})^{\top}\boldsymbol{X})N_{1,n}(\boldsymbol{p}_{*,j})^{\top}\boldsymbol{X}f_{\bar{G}_n}(\boldsymbol{X}) \\
&- \sum_{j:|\mathcal{V}_j|>1} \exp((B\boldsymbol{p}_{*,j})^{\top}\boldsymbol{X})\big[\bar{N}_{1,n}(\boldsymbol{p}_{*,j})^{\top}B^{\top}\boldsymbol{X} + (B^{\top}\boldsymbol{X})^{\top}\bar{N}_{2,n}(\boldsymbol{p}_{*,j})B^{\top}\boldsymbol{X}\big]f_{\bar{G}_n}(\boldsymbol{X}) \\
&- \sum_{j=1}^{L} M_{n,j,0_d} \exp((B\boldsymbol{p}_{*,j})^{\top}\boldsymbol{X})f_{G_n}(\boldsymbol{X}) + \sum_{j=1}^{L} M_{n,j,0_d} \exp((B\boldsymbol{p}_{*,j})^{\top}\boldsymbol{X})C\boldsymbol{p}_{*,j} \\
&+ \bar{R}_{n,1}(\boldsymbol{X}) + \bar{R}_{n,2}(\boldsymbol{X}) - R_{n,3}(\boldsymbol{X}) - R_{n,4}(\boldsymbol{X}) \\
=& \sum_{j:|\mathcal{V}_j|=1} \exp((B\boldsymbol{p}_{*,j})^{\top}\boldsymbol{X})\big[L'_{1,n}(\boldsymbol{p}_{*,j}) + L_{2,n}(\boldsymbol{p}_{*,j})^{\top}B^{\top}\boldsymbol{X}\big) \\
&+ \sum_{j:|\mathcal{V}_j|>1} \exp((B\boldsymbol{p}_{*,j})^{\top}\boldsymbol{X})\big[\bar{L}'_{1,n}(\boldsymbol{p}_{*,j}) + \bar{L}_{2,n}(\boldsymbol{p}_{*,j})^{\top}B^{\top}\boldsymbol{X} + (B^{\top}\boldsymbol{X})^{\top}\bar{L}_{3,n}(\boldsymbol{p}_{*,j})B^{\top}\boldsymbol{X}\big] \\
&- \sum_{j:|\mathcal{V}_j|=1} \exp((B\boldsymbol{p}_{*,j})^{\top}\boldsymbol{X})\big[M_{n,j,0_d} + N_{1,n}(\boldsymbol{p}_{*,j})^{\top}B^{\top}\boldsymbol{X}\big]f_{\bar{G}_n}(\boldsymbol{X}) \\
&- \sum_{j:|\mathcal{V}_j|>1} \exp((B\boldsymbol{p}_{*,j})^{\top}\boldsymbol{X})\big[M_{n,j,0_d} + \bar{N}_{1,n}(\boldsymbol{p}_{*,j})^{\top}B^{\top}\boldsymbol{X} + (B^{\top}\boldsymbol{X})^{\top}\bar{N}_{2,n}(\boldsymbol{p}_{*,j})B^{\top}\boldsymbol{X}\big]f_{\bar{G}_n}(\boldsymbol{X}) \\
&+ \bar{R}_{n,1}(\boldsymbol{X}) + \bar{R}_{n,2}(\boldsymbol{X}) - R_{n,3}(\boldsymbol{X}) - R_{n,4}(\boldsymbol{X}) \quad\quad (28)
\end{aligned}
$$

where $M_{n,j,0_d} = \sum_{i\in\mathcal{V}_j} \exp(b_{n,i}) - \exp(b_{*,j})$ for any $j \in [L]$, $L'_{1,n}(\boldsymbol{p}_{*,j}) = L_{1,n}(\boldsymbol{p}_{*,j}) + M_{n,j,0_d}C\boldsymbol{p}_{*,j}$, and $\bar{L}'_{1,n}(\boldsymbol{p}_{*,j}) = \bar{L}_{1,n}(\boldsymbol{p}_{*,j}) + M_{n,j,0_d}C\boldsymbol{p}_{*,j}$.

**Step 2 - Non-vanishing coefficients.** From equation (35), we can represent $Q_n(\boldsymbol{X})/\mathcal{D}_{2n}$ as a linear combination of the independent functions $\exp((B\boldsymbol{p}_{*,j})^{\top}\boldsymbol{X})$, $(B^{\top}\boldsymbol{X})^{(u)}\exp((B\boldsymbol{p}_{*,j})^{\top}\boldsymbol{X})$, $(B^{\top}\boldsymbol{X})^{(u)}(B^{\top}\boldsymbol{X})^{(v)}\exp((B\boldsymbol{p}_{*,j})^{\top}\boldsymbol{X})$, $\exp((B\boldsymbol{p}_{*,j})^{\top}\boldsymbol{X})f_{\bar{G}_n}(\boldsymbol{X})$, $(B^{\top}\boldsymbol{X})^{(u)}\exp((B\boldsymbol{p}_{*,j})^{\top}\boldsymbol{X})f_{\bar{G}_n}(\boldsymbol{X})$, and $(B^{\top}\boldsymbol{X})^{(u)}(B^{\top}\boldsymbol{X})^{(v)}\exp((B\boldsymbol{p}_{*,j})^{\top}\boldsymbol{X})f_{\bar{G}_n}(\boldsymbol{X})$ for any $1 \leq j \leq L$ and $1 \leq u,v \leq d$.

Assume that all the coefficients of these linear independent functions in the formulation of $Q_n(\boldsymbol{X})/\mathcal{D}_{2n}$ go to 0 as $n \to \infty$. It follows that $L_{1,n}(\boldsymbol{p}_{*,j})/\mathcal{D}_{2n}$, $L_{2,n}(\boldsymbol{p}_{*,j})^{(u)}/\mathcal{D}_{2n}$, $\bar{L}_{1,n}(\boldsymbol{p}_{*,j})/\mathcal{D}_{2n}$, $\bar{L}_{2,n}(\boldsymbol{p}_{*,j})^{(u)}/\mathcal{D}_{2n}$, $\bar{L}_{3,n}(\boldsymbol{p}_{*,j})^{(uv)}/\mathcal{D}_{2n}$, $N_{1,n}(\boldsymbol{p}_{*,j})/\mathcal{D}_{2n}$, $\bar{N}_{1,n}((\boldsymbol{p}_{*,j})^{(u)}/\mathcal{D}_{2n}$, $\bar{N}_{2,n}(\boldsymbol{p}_{*,j})^{(uv)}/\mathcal{D}_{2n}$, and $M_{n,j,0_d}/\mathcal{D}_{2n}$ approach 0 as $n \to \infty$ for any $1 \leq u,v \leq d$ and $1 \leq j \leq L$.

Then, as $M_{n,j,0_d}/\mathcal{D}_{2n} \to 0$, it indicates that

$$\frac{|M_{n,j,0_d}|}{\mathcal{D}_{2n}} = \frac{|\sum_{i\in\mathcal{V}_j} \exp(b_{n,i}) - \exp(b_{*,j})|}{\mathcal{D}_{2n}} \to 0,$$

for any $1 \leq j \leq L$. By summing these limits up when varying the index $j$ from 1 to $L$, we obtain that

$$\frac{\sum_{j=1}^{L} |\sum_{i \in \mathcal{V}_j} \exp(b_{n,i}) - \exp(b_{*,j})|}{\mathcal{D}_{2n}} \to 0. \tag{29}$$

Now, we consider indices $j \in [L]$ such that its corresponding Voronoi cell has only one element, i.e. $|\mathcal{V}_j| = 1$. As $L_{2,n}(\boldsymbol{p}_{*,j})^{(u)}/\mathcal{D}_{2n} \to 0$, it indicates that $M_{n,j,1_u}/\mathcal{D}_{2n} \to 0$. It indicates that

$$\frac{\sum_{u=1}^{d} \exp(b_{n,i})|M_{n,j,1_u}|}{\mathcal{D}_{2n}} = \frac{\sum_{i \in \mathcal{V}_j} \exp(b_{n,i})\|\Delta p_{n,ij}\|}{\mathcal{D}_{2n}} \to 0.$$

Putting the above results together, we find that

$$\frac{\sum_{j:|\mathcal{V}_j|=1} \sum_{i \in \mathcal{V}_j} \exp(b_{n,i})\|\Delta p_{n,ij}\|}{\mathcal{D}_{2n}} \to 0. \tag{30}$$

Moving to indices $j \in [L]$ such that $|\mathcal{V}_j| > 1$, as $\bar{L}_{3,n}(\boldsymbol{p}_{*,j})^{(uu)}/\mathcal{D}_{2n} \to 0$, we obtain that

$$\frac{\sum_{u=1}^{d} \exp(b_{n,i})\bar{L}_{3,n}(\boldsymbol{p}_{*,j})^{(uu)}}{\mathcal{D}_{2n}} = \frac{\sum_{i \in \mathcal{V}_j} \exp(b_{n,i})\|\Delta \boldsymbol{p}_{n,ij}\|^2}{\mathcal{D}_{2n}} \to 0.$$

Therefore, we find that

$$\frac{\sum_{j:|\mathcal{V}_j|>1} \sum_{i \in \mathcal{V}_j} \exp(b_{n,i})\|\Delta p_{n,ij}\|^2}{\mathcal{D}_{2n}} \to 0.$$

Collecting all the above results, we obtain that

$$1 = \frac{\mathcal{D}_{2n}}{\mathcal{D}_{2n}} \to 0$$

as $n \to \infty$, which is a contradiction.

As a consequence, not all of the coefficients of the linear independent functions in the formulations of $Q_n(\boldsymbol{X})/\mathcal{D}_{2n}$ go to 0 as $n \to \infty$.

**Step 3 - Application of Fatou's lemma.** In particular, let denote $m_n$ as the maximum of the absolute values of $L'_{1,n}(\boldsymbol{p}_{*,j})/\mathcal{D}_{2n}$, $L_{2,n}(\boldsymbol{p}_{*,j})^{(u)}/\mathcal{D}_{2n}$, $\bar{L}'_{1,n}(\boldsymbol{p}_{*,j})/\mathcal{D}_{2n}$, $\bar{L}_{2,n}(\boldsymbol{p}_{*,j})^{(u)}/\mathcal{D}_{2n}$, $\bar{L}_{3,n}(\boldsymbol{p}_{*,j})^{(uv)}/\mathcal{D}_{2n}$, $N_{1,n}(\boldsymbol{p}_{*,j})/\mathcal{D}_{2n}$, $\bar{N}_{1,n}((\boldsymbol{p}_{*,j})^{(u)}/\mathcal{D}_{2n}$, $\bar{N}_{2,n}(\boldsymbol{p}_{*,j})^{(uv)}/\mathcal{D}_{2n}$, and $M_{n,j,0_d}/\mathcal{D}_{2n}$ for all $1 \leq u,v \leq d$. From the result of Step 2, it follows that $1/m_n \not\to \infty$ as $n \to \infty$.

Recall that $\|f_{\bar{G}_n} - f_{\bar{G}_*}\|_{L^2(\mu)}/\mathcal{D}_{2n} \to 0$ as $n \to \infty$, which indicates that $\|f_{\bar{G}_n} - f_{\bar{G}_*}\|_{L^2(\mu)}/(m_n \mathcal{D}_{2n}) \to 0$. By applying Fatou's lemma, we get that

$$0 = \lim_{n \to \infty} \frac{\|f_{\bar{G}_n} - f_{\bar{G}_*}\|_{L^2(\mu)}}{m_n \mathcal{D}_{2n}} \geq \int \liminf_{n \to \infty} \frac{|f_{\bar{G}_n}(\boldsymbol{X}) - f_{\bar{G}_*}(\boldsymbol{X})|}{m_n \mathcal{D}_{2n}} d\mu(\boldsymbol{X}) \geq 0.$$

It indicates that $\liminf_{n \to \infty} \frac{|f_{\bar{G}_n}(\boldsymbol{X}) - f_{\bar{G}_*}(\boldsymbol{X})|}{m_n \mathcal{D}_{2n}} = 0$ for almost surely $\boldsymbol{X}$. As $n \to \infty$, we denote

$$\frac{L'_{1,n}(\boldsymbol{p}_{*,j})}{m_n \mathcal{D}_{2n}} \to \alpha_j, \quad \frac{L_{2,n}(\boldsymbol{p}_{*,j})}{m_n \mathcal{D}_{2n}} \to \beta_j,$$

$$\frac{\bar{L}'_{1,n}(\boldsymbol{p}_{*,j})}{m_n \mathcal{D}_{2n}} \to \bar{\alpha}_j, \quad \frac{\bar{L}_{2,n}(\boldsymbol{p}_{*,j})}{m_n \mathcal{D}_{2n}} \to \bar{\beta}_j, \quad \frac{\bar{L}_{3,n}(\boldsymbol{p}_{*,j})}{m_n \mathcal{D}_{2n}} \to \bar{\gamma}_j,$$

$$\frac{M_{n,j,0_d}}{\mathcal{D}_{2n}} \to \tilde{\alpha}_j, \quad \frac{N_{1,n}(\boldsymbol{p}_{*,j})}{m_n \mathcal{D}_{2n}} \to \tilde{\beta}_j,$$

$$\frac{\bar{N}_{1,n}(\boldsymbol{p}_{*,j})}{m_n \mathcal{D}_{2n}} \to \widehat{\beta}_j, \quad \frac{\bar{N}_{2,n}(\boldsymbol{p}_{*,j})}{m_n \mathcal{D}_{2n}} \to \widehat{\gamma}_j$$

for any $1 \leq j \leq L$. Here, from the definition of $m_n$, at least one coefficient among $\{\alpha_j, \beta_j, \tilde{\alpha}_j, \tilde{\beta}_j\}_{j:|\mathcal{V}_j|=1}$, $\{\bar{\alpha}_j, \bar{\beta}_j, \bar{\gamma}_j, \tilde{\alpha}_j, \hat{\beta}_j, \hat{\gamma}_j\}_{j:|\mathcal{V}_j|>1}$ is different from 0. Then, the equation $\liminf_{n \to \infty} \frac{|f_{\bar{G}_n}(\boldsymbol{X}) - f_{\bar{G}_*}(\boldsymbol{X})|}{m_n \mathcal{D}_{2n}} = 0$ leads to

$$
\sum_{j:|\mathcal{V}_j|=1} \exp((B\boldsymbol{p}_{*,j})^\top \boldsymbol{X})(\alpha_j + \beta_j^\top (B^\top \boldsymbol{X}))
$$
$$
+ \sum_{j:|\mathcal{V}_j|>1} \exp((B\boldsymbol{p}_{*,j})^\top \boldsymbol{X}) \big[\bar{\alpha}_j + \bar{\beta}_j^\top (B^\top \boldsymbol{X}) + (B^\top \boldsymbol{X})^\top \bar{\gamma}_j (B^\top \boldsymbol{X})\big]
$$
$$
- \sum_{j:|\mathcal{V}_j|=1} \exp((B\boldsymbol{p}_{*,j})^\top \boldsymbol{X})(\tilde{\alpha}_j + \tilde{\beta}_j^\top (B^\top \boldsymbol{X})) f_{\bar{G}_*}(\boldsymbol{X})
$$
$$
- \sum_{j:|\mathcal{V}_j|>1} \exp((B\boldsymbol{p}_{*,j})^\top \boldsymbol{X}) \big[\tilde{\alpha}_j + \hat{\beta}_j^\top (B^\top \boldsymbol{X}) + (B^\top \boldsymbol{X})^\top \hat{\gamma}_j B^\top \boldsymbol{X}\big] f_{\bar{G}_*}(\boldsymbol{X}) = 0
$$

for almost surely $\boldsymbol{X}$. By denoting $\boldsymbol{Z} = B^\top \boldsymbol{X}$, this equation also holds for almost surely $\boldsymbol{Z}$. However, the new equation implies that all the coefficients $\{\alpha_j, \beta_j, \tilde{\alpha}_j, \tilde{\beta}_j\}_{j:|\mathcal{V}_j|=1}$, $\{\bar{\alpha}_j, \bar{\beta}_j, \bar{\gamma}_j, \tilde{\alpha}_j, \hat{\beta}_j, \hat{\gamma}_j\}_{j:|\mathcal{V}_j|>1}$ are 0, which is a contradiction.

It indicates that we indeed have the conclusion of the local part, namely,

$$
\lim_{\varepsilon \to 0} \inf_{\bar{G} \in \bar{\mathcal{G}}_{L'}(\Omega): \mathcal{D}_2(\bar{G}, \bar{G}_*) \leq \varepsilon} \|f_{\bar{G}} - f_{\bar{G}_*}\|_{L^2(\mu)} / \mathcal{D}_2(\bar{G}, \bar{G}_*) > 0.
$$

**Global part:** From local part, there exists a positive constant $\varepsilon'$ such that

$$
\inf_{\bar{G} \in \bar{\mathcal{G}}_{L'}(\Omega): \mathcal{D}_2(\bar{G}, \bar{G}_*) \leq \varepsilon'} \|f_{\bar{G}} - f_{\bar{G}_*}\|_{L^2(\mu)} / \mathcal{D}_2(\bar{G}, \bar{G}_*) > 0.
$$

Therefore, it is sufficient to prove that

$$
\inf_{\bar{G} \in \bar{\mathcal{G}}_{L'}(\Omega): \mathcal{D}_2(\bar{G}, \bar{G}_*) > \varepsilon'} \|f_{\bar{G}} - f_{\bar{G}_*}\|_{L^2(\mu)} / \mathcal{D}_2(G, \bar{G}_*) > 0.
$$

Assume by contrary, then we can find a sequence of mixing measures $\bar{G}'_n := \sum_{j'=1}^{L'} \exp(b_{n,j'}) \delta_{\boldsymbol{p}_{n,j'}}$ in $\bar{\mathcal{G}}_{L'}(\Omega)$ such that as $n \to \infty$, we have

$$
\begin{cases}
\mathcal{D}_2(\bar{G}'_n, \bar{G}_*) > \varepsilon' \\
\|f_{\bar{G}'_n} - f_{\bar{G}_*}\|_{L^2(\mu)} / \mathcal{D}_2(\bar{G}'_n, \bar{G}_*) \to 0,
\end{cases}
$$

which indicates that $\|f_{\bar{G}'_n} - f_{\bar{G}_*}\|_{L^2(\mu)} \to 0$ as $n \to \infty$.

Recall that $\Omega$ is a compact set. Therefore, there exists a mixing measure $\bar{G}'$ in $\bar{\mathcal{G}}_{L'}(\Omega)$ such that one of $\bar{G}'_n$'s subsequences converges to $\bar{G}'$. Since $\mathcal{D}_2(\bar{G}'_n, \bar{G}_*) > \varepsilon'$, we deduce that $\mathcal{D}_2(\bar{G}', \bar{G}_*) > \varepsilon'$. By invoking the Fatou's lemma, we have that

$$
0 = \lim_{n \to \infty} \|f_{\bar{G}'_n} - f_{\bar{G}_*}\|_{L^2(\mu)} \geq \int \liminf_{n \to \infty} \left|f_{\bar{G}'_n} - f_{\bar{G}_*}\right|^2 d\mu(\boldsymbol{X}).
$$

Thus, we have $f_{\bar{G}'} = f_{\bar{G}_*}$ for $\mu-$almost surely $\boldsymbol{X}$. From the identifiability property (cf. the end of this proof), we deduce that $\bar{G}' \equiv \bar{G}_*$. It follows that $\mathcal{D}_2(\bar{G}', \bar{G}_*) = 0$, contradicting the fact that $\mathcal{D}_2(\bar{G}', \bar{G}_*) > \varepsilon' > 0$.
Hence, the proof of the global part is completed.

**Identifiability property.** We now prove the identifiability of shared strutures among prompts. In particular, we will show that if $f_{\bar{G}}(\boldsymbol{X}) = f_{\bar{G}_*}(\boldsymbol{X})$ for almost every $\boldsymbol{X}$, then it follows that $\bar{G} \equiv \bar{G}_*$.

For any $\bar{G} \in \bar{\mathcal{G}}_{L'}(\Omega)$, let us denote

$$
\mathrm{softmax}_{\bar{G}}(u) = \frac{\exp(u)}{\sum_{k=1}^N \exp(\boldsymbol{X}^\top A_k^0 \boldsymbol{X} + a_k^0) + \sum_{j'=1}^{L'} \exp((B\boldsymbol{p}_{j'})^\top \boldsymbol{X} + b_{j'})},
$$
$$
\mathrm{softmax}_{\bar{G}_*}(u_*) = \frac{\exp(u_*)}{\sum_{k=1}^N \exp(\boldsymbol{X}^\top A_k^0 \boldsymbol{X} + a_k^0) + \sum_{j'=1}^{L} \exp((B\boldsymbol{p}_{*,j'})^\top \boldsymbol{X} + b_{*,j'})},
$$

where

$$u \in \{\boldsymbol{X}^\top A_j^0 \boldsymbol{X} + a_j^0; (B\boldsymbol{p}_{j'})^\top \boldsymbol{X} + b_{j'} : j \in [N], j' \in [L']\},$$
$$u_* \in \{\boldsymbol{X}^\top A_j^0 \boldsymbol{X} + a_j^0; (B\boldsymbol{p}_{*,j'})^\top \boldsymbol{X} + b_{*,j'} : j \in [N], j' \in [L]\}.$$

Since $f_{\bar{G}}(\boldsymbol{X}) = f_{\bar{G}_*}(\boldsymbol{X})$ for almost every $\boldsymbol{X}$, we have

$$\sum_{j=1}^N \mathrm{softmax}_{\bar{G}}(\boldsymbol{X}^\top A_j^0 \boldsymbol{X} + a_j^0))h(\boldsymbol{X}, \eta_j^0) + \sum_{j'=1}^{L'} \mathrm{softmax}_{\bar{G}}((B\boldsymbol{p}_{j'})^\top \boldsymbol{X} + b_{j'})C\boldsymbol{p}_{j'}$$
$$= \sum_{j=1}^N \mathrm{softmax}_{\bar{G}_*}(\boldsymbol{X}^\top A_j^0 \boldsymbol{X} + a_j^0))h(\boldsymbol{X}, \eta_j^0) + \sum_{j'=1}^{L} \mathrm{softmax}_{\bar{G}_*}((B\boldsymbol{p}_{*,j'})^\top \boldsymbol{X} + b_{*,j'})C\boldsymbol{p}_{*,j'}.$$
(31)

Thus, we must have that $L = L'$. As a result,

$$\{\mathrm{softmax}_{\bar{G}}((B\boldsymbol{p}_{j'})^\top \boldsymbol{X} + b_{j'}) : j' \in [L]\} = \{\mathrm{softmax}_{\bar{G}_*}((B\boldsymbol{p}_{*,j'})^\top \boldsymbol{X} + b_{*,j'}) : j' \in [L']\},$$

for almost every $\boldsymbol{X}$. Without loss of generality, we assume that

$$\mathrm{softmax}_{\bar{G}}((B\boldsymbol{p}_{j'})^\top \boldsymbol{X} + b_{j'}) = \mathrm{softmax}_{\bar{G}_*}((B\boldsymbol{p}_{*,j'})^\top \boldsymbol{X} + b_{*,j'}),$$

for any $j' \in [L]$, for almost every $\boldsymbol{X}$. Since the softmax function is invariant to translation, this result indicates that $b_{j'} = b_{*,j'} + r$ for some $r \in \mathbb{R}$ and for any $j' \in [L]$. Then, the equation (31) can be reduced to

$$\sum_{j=1}^L \exp(b_j) \exp((B\boldsymbol{p}_j)^\top \boldsymbol{X})C\boldsymbol{p}_j = \sum_{j=1}^L \exp(b_{*,j}) \exp((B\boldsymbol{p}_{*,j})^\top \boldsymbol{X})C\boldsymbol{p}_{*,j}, \quad (32)$$

for almost surely $\boldsymbol{X}$. Next, we will partition the index set $[L]$ into $m$ subsets $K_1, K_2, \ldots, K_m$ where $m \leq L$, such that $\exp(b_j) = \exp(b_{*,j'})$ for any $j, j' \in K_i$ and $i \in [m]$. It follows that $\exp(b_j) \neq \exp(b_{*,j'})$ when $j, j'$ do not belong to the same set $K_i$. Thus, we can rewrite equation (32) as

$$\sum_{i=1}^m \sum_{j \in K_i} \exp(b_j) \exp((B\boldsymbol{p}_j)^\top \boldsymbol{X})C\boldsymbol{p}_j$$
$$= \sum_{i=1}^m \sum_{j \in K_i} \exp(b_{*,j}) \exp((B\boldsymbol{p}_{*,j})^\top \boldsymbol{X})C\boldsymbol{p}_{*,j},$$

for almost surely $\boldsymbol{X}$. Given the above equation, for each $i \in [m]$, we obtain that

$$\{((B\boldsymbol{p}_j)^\top, \boldsymbol{p}_j) : j \in K_i\} = \{((B\boldsymbol{p}_{*,j})^\top, \boldsymbol{p}_{*,j}) : j \in K_i\},$$

for almost surely $\boldsymbol{X}$, which directly leads to

$$\{\boldsymbol{p}_j : j \in K_i\} = \{\boldsymbol{p}_{*,j} : j \in K_i\}$$

Without loss of generality, we assume that $\boldsymbol{p}_j = \boldsymbol{p}_{*,j}$ for all $j \in K_i$. Consequently, we get that

$$\sum_{i=1}^m \sum_{j \in K_i} \exp(b_j)\delta_{\boldsymbol{p}_j} = \sum_{i=1}^m \sum_{j \in K_i} \exp(b_{*,j})\delta_{\boldsymbol{p}_{*,j}},$$

or $\bar{G} \equiv \bar{G}_*$. The proof is completed.

### B.3 PROOF OF THEOREM 4.3

The proof strategy of Theorem 4.3 is also similar to that of Theorem 4.2. We first establish the parametric convergence rate $\mathcal{O}_P(\sqrt{\log(n)/n})$ of the estimated regression function $f_{\widetilde{G}_n}$ to the true regression function $f_{\widetilde{G}_*}$ in Section B.3.1. Then, in Section B.3.2, we establish the lower bound $\|f_{\widetilde{G}} - f_{\widetilde{G}_*}\|_{L^2(\mu)} \geq C' \mathcal{D}_2(G, \widetilde{G}_*)$ for any $\widetilde{G} \in \widetilde{\mathcal{G}}_{L'}(\Xi)$ for some universal constant $C'$.

### B.3.1 CONVERGENCE RATE OF DENSITY ESTIMATION

**Proposition B.2.** *Given the least square estimator $\widetilde{G}_n$ in equation (16), the convergence rate of the model estimation $f_{\widetilde{G}_n}(\cdot)$ to the true model $f_{\widetilde{G}_*}(\cdot)$ under the $L^2(\mu)$ norm is parametric on the sample size, that is,*

$$\|f_{\widetilde{G}_n} - f_{\widetilde{G}_*}\|_{L^2(\mu)} = \mathcal{O}_P(\sqrt{\log(n)/n}). \tag{33}$$

The proof argument of Proposition B.2 is similar to that of Proposition B.1; therefore, it is omitted.

### B.3.2 FROM DENSITY ESTIMATION TO EXPERT ESTIMATION

Given the convergence rate of regression function estimation in Proposition 4.3, our goal is to demonstrate the following inequality:

$$\inf_{\widetilde{G} \in \widetilde{\mathcal{G}}_{L'}(\Xi)} \|f_{\widetilde{G}} - f_{\widetilde{G}_*}\|_{L^2(\mu)}/\mathcal{D}_3(\widetilde{G}, \widetilde{G}_*) > 0.$$

Similar to the proof of Theorem 4.2, we divide the proof of the above inequality into local and global parts.

**Local part:**  We will demonstrate that

$$\lim_{\varepsilon \to 0} \inf_{\widetilde{G} \in \widetilde{\mathcal{G}}_{L'}(\Xi): \mathcal{D}_3(\widetilde{G}, \widetilde{G}_*) \leq \varepsilon} \|f_{\widetilde{G}} - f_{\widetilde{G}_*}\|_{L^2(\mu)}/\mathcal{D}_3(\widetilde{G}, \widetilde{G}_*) > 0$$

Assume by contrary that the above claim does not hold. Then, there exists a sequence of mixing measures $\widetilde{G}_n := \sum_{j'=1}^{L'} \exp(b_{n,j'})\delta_{(W_{n,1}\boldsymbol{p}_{n,j'}, W_{n,2}\boldsymbol{p}_{n,j'})}$ in $\widetilde{\mathcal{G}}_{L'}(\Xi)$ such that as $n \to \infty$, we have

$$\begin{cases} \mathcal{D}_{3n} := \mathcal{D}_3(\widetilde{G}_n, \widetilde{G}_*) \to 0, \\ \|f_{\widetilde{G}_n} - f_{\widetilde{G}_*}\|_{L^2(\mu)}/\mathcal{D}_{3n} \to 0. \end{cases}$$

To ease the ensuing presentation, we also denote $\mathcal{V}_j^n := \mathcal{V}_j(\widetilde{G}_n)$ as a Voronoi cell of $G_n$ generated by the $j$-th components of $\widetilde{G}_*$. Since our arguments are asymptotic, we may assume that those Voronoi cells do not depend on the sample size, i.e., $\mathcal{V}_j = \mathcal{V}_j^n$. Therefore, we can represent the Voronoi loss $\mathcal{D}_{3n}$ as follows:

$$\begin{aligned} \mathcal{D}_{3n} = &\sum_{j'=1}^{L} \Big| \sum_{i \in \mathcal{V}_{j'}} \exp(b_{n,i}) - \exp(b_{*,j'}) \Big| \\ &+ \sum_{j' \in [L]: |\mathcal{V}_{j'}|=1} \sum_{i \in \mathcal{V}_{j'}} \exp(b_{n,i})(\|\Delta W_{n,1}\boldsymbol{p}_{n,ij'}\| + \|\Delta W_{n,2}\boldsymbol{p}_{n,ij'}\|) \\ &+ \sum_{j' \in [L]: |\mathcal{V}_{j'}|>1} \sum_{i \in \mathcal{V}_{j'}} \exp(b_{n,i})(\|\Delta W_{n,1}\boldsymbol{p}_{n,ij'}\|^2 + \|\Delta W_{n,2}\boldsymbol{p}_{n,ij'}\|^2) \end{aligned}$$

where we define $\Delta W_{n,1}\boldsymbol{p}_{n,ij'} = W_{n,1}\boldsymbol{p}_{n,i} - W_{*,1}\boldsymbol{p}_{*,j'}$ and $\Delta W_{n,2}\boldsymbol{p}_{n,ij'} = W_{n,2}\boldsymbol{p}_{n,i} - W_{*,2}\boldsymbol{p}_{*,j'}$ for all $i \in \mathcal{V}_{j'}$.

Additionally, since $\mathcal{D}_{3n} \to 0$, we have $\sum_{i \in \mathcal{V}_j} \exp(b_{n,i}) \to \exp(b_{*,j})$, $W_{n,1}\boldsymbol{p}_{n,i} \to W_{*,1}\boldsymbol{p}_{*,j}$, and $W_{n,2}\boldsymbol{p}_{n,i} \to W_{*,2}\boldsymbol{p}_{*,j}$ for any $i \in \mathcal{V}_j, j \in [L]$. Now, we divide the proof of the local part into three steps as follows:

**Step 1 - Taylor expansion.**  In this step, we would like to decompose the quantity

$$\widetilde{Q}_n(\boldsymbol{X}) := \Big[ \sum_{j=1}^{N} \exp(\boldsymbol{X}^\top A_j^0 \boldsymbol{X} + a_j^0) + \sum_{j'=1}^{L} \exp((B\bar{\sigma}_1(W_{*,1}\boldsymbol{p}_{*,j'}))^\top \boldsymbol{X} + b_{*,j'}) \Big] \cdot [f_{\widetilde{G}_n}(\boldsymbol{X}) - f_{\widetilde{G}_*}(\boldsymbol{X})],$$

as follows:

$$
\begin{aligned}
\widetilde{Q}_n(\boldsymbol{X}) = \sum_{j=1}^{L} \sum_{i \in \mathcal{V}_j} \exp(b_{n,i}) \Big[ & \exp((B\bar{\sigma}_1(W_{n,1}\boldsymbol{p}_{n,i}))^\top \boldsymbol{X}) C\bar{\sigma}_2(W_{n,2}\boldsymbol{p}_{n,i}) \\
& - \exp((B\bar{\sigma}_1(W_{*,1}\boldsymbol{p}_{*,j}))^\top \boldsymbol{X}) C\bar{\sigma}_2(W_{*,2}\boldsymbol{p}_{*,j}) \Big] \\
- \sum_{j=1}^{L} \sum_{i \in \mathcal{V}_j} \exp(b_{n,i}) \Big[ & \exp((B\bar{\sigma}_1(W_{n,1}\boldsymbol{p}_{n,i}))^\top \boldsymbol{X}) - \exp((B\bar{\sigma}_1(W_{*,1}\boldsymbol{p}_{*,j}))^\top \boldsymbol{X}) \Big] f_{\widetilde{G}_n}(\boldsymbol{X}) \\
+ \sum_{j=1}^{L} \Big( \sum_{i \in \mathcal{V}_j} & \exp(b_{n,i}) - \exp(b_{*,j}) \Big) \exp((B\bar{\sigma}_1(W_{*,1}\boldsymbol{p}_{*,j}))^\top \boldsymbol{X}) \Big[ C\bar{\sigma}_2(W_{*,2}\boldsymbol{p}_{*,j}) - f_{\widetilde{G}_n}(\boldsymbol{X}) \Big] \\
:= \widetilde{A}_n(\boldsymbol{X}) - \widetilde{B}_n(\boldsymbol{X}) & + \widetilde{C}_n(\boldsymbol{X}).
\end{aligned} \tag{34}
$$

**Decomposition of $\widetilde{A}_n(\boldsymbol{X})$.** To ease the ensuing presentation, we denote $E(\boldsymbol{X}; W_1\boldsymbol{p}) := \exp((B\bar{\sigma}_1(W_1\boldsymbol{p}))^\top \boldsymbol{X})$ and $H(W_2\boldsymbol{p}) = C\bar{\sigma}_2(W_2\boldsymbol{p})$, and $F(\boldsymbol{X}; W_1\boldsymbol{p}, W_2\boldsymbol{p}) = E(\boldsymbol{X}; W_1\boldsymbol{p})H(W_2\boldsymbol{p})$. Since each Voronoi cell $\mathcal{V}_j$ possibly has more than one element, we continue to decompose $\bar{A}_n$ as follows:

$$
\begin{aligned}
\widetilde{A}_n(\boldsymbol{X}) = \sum_{j:|\mathcal{V}_j|=1} \sum_{i \in \mathcal{V}_j} & \exp(b_{n,i}) \Big[ F(\boldsymbol{X}; W_{n,1}\boldsymbol{p}_{n,i}, W_{n,2}\boldsymbol{p}_{n,i}) - F(\boldsymbol{X}; W_{*,1}\boldsymbol{p}_{*,j}, W_{*,2}\boldsymbol{p}_{*,j}) \Big] \\
+ \sum_{j:|\mathcal{V}_j|>1} \sum_{i \in \mathcal{V}_j} & \exp(b_{n,i}) \Big[ F(\boldsymbol{X}; W_{n,1}\boldsymbol{p}_{n,i}, W_{n,2}\boldsymbol{p}_{n,i}) - F(\boldsymbol{X}; W_{*,1}\boldsymbol{p}_{*,j}, W_{*,2}\boldsymbol{p}_{*,j}) \Big] \\
:= \widetilde{A}_{n,1}(\boldsymbol{X}) + \widetilde{A}_{n,2} & (\boldsymbol{X})
\end{aligned}
$$

By means of the first-order Taylor expansion, we have

$$
E(\boldsymbol{X}; W_{n,1}\boldsymbol{p}_{n,i}) = E(\boldsymbol{X}; W_{*,1}\boldsymbol{p}_{*,j}) + \sum_{|\alpha|=1} (\Delta W_{n,1}\boldsymbol{p}_{n,ij})^\alpha \frac{\partial^{|\alpha|}E}{\partial(W_1\boldsymbol{p})^\alpha}(\boldsymbol{X}; W_{*,1}\boldsymbol{p}_{*,j}) + R_{ij,1}(\boldsymbol{X}),
$$

$$
H(W_{n,2}\boldsymbol{p}_{n,i}) = H(W_{*,2}\boldsymbol{p}_{*,j}) + \sum_{|\alpha|=1} (\Delta W_{n,2}\boldsymbol{p}_{n,ij})^\alpha \frac{\partial^{|\alpha|}H}{\partial(W_2\boldsymbol{p})^\alpha}(W_{*,2}\boldsymbol{p}_{*,j}) + R_{ij,2},
$$

for any $i \in \mathcal{V}_j$ and $j$ such that $|\mathcal{V}_j| = 1$. Here, $R_{ij,1}(\boldsymbol{X})$ and $R_{ij,2}$ are Taylor remainders. Putting the above results together leads to

$$
\begin{aligned}
\widetilde{A}_{n,1}(\boldsymbol{X}) = \sum_{j:|\mathcal{V}_j|=1} \sum_{i \in \mathcal{V}_j} \frac{\exp(b_{n,i})}{\alpha!} \sum_{|\alpha|=1} \Big\{ & (\Delta W_{n,1}\boldsymbol{p}_{n,ij})^\alpha \frac{\partial^{|\alpha|}E}{\partial(W_1\boldsymbol{p})^\alpha}(\boldsymbol{X}; W_{*,1}\boldsymbol{p}_{*,j}) H(W_{*,2}\boldsymbol{p}_{*,j}) \\
+ (\Delta W_{n,2}\boldsymbol{p}_{n,ij})^\alpha \frac{\partial^{|\alpha|}H}{\partial(W_2\boldsymbol{p})^\alpha} & (W_{*,2}\boldsymbol{p}_{*,j}) E(\boldsymbol{X}; W_{*,1}\boldsymbol{p}_{*,j}) \Big\} + \bar{R}_{n,1}(\boldsymbol{X}) \\
= \sum_{j:|\mathcal{V}_j|=1} \sum_{|\alpha|=1} \Big\{ M_{n,j,\alpha}^{(1)} & \frac{\partial^{|\alpha|}E}{\partial(W_1\boldsymbol{p})^\alpha}(\boldsymbol{X}; W_{*,1}\boldsymbol{p}_{*,j}) H(W_{*,2}\boldsymbol{p}_{*,j}) \\
+ M_{n,j,\alpha}^{(2)} \frac{\partial^{|\alpha|}H}{\partial(W_2\boldsymbol{p})^\alpha} & (W_{*,2}\boldsymbol{p}_{*,j}) E(\boldsymbol{X}; W_{*,1}\boldsymbol{p}_{*,j}) \Big\} + \bar{R}_{n,1}(\boldsymbol{X})
\end{aligned}
$$

where $\bar{R}_{n,1}(\boldsymbol{X})$ satisfies $\bar{R}_{n,1}(\boldsymbol{X})/\mathcal{D}_{3n} \to 0$ when $n \to \infty$, which is due to the uniform Lipschitz property of the function $F$. Furthermore, the formulations of $M_{n,j,\alpha}^{(1)}$ and $M_{n,j,\alpha}^{(2)}$ are given by:

$$
M_{n,j,\alpha}^{(1)} = \sum_{i \in \mathcal{V}_j} \frac{\exp(b_{n,i})}{\alpha!} (\Delta W_{n,1}\boldsymbol{p}_{n,ij})^\alpha,
$$

$$
M_{n,j,\alpha}^{(2)} = \sum_{i \in \mathcal{V}_j} \frac{\exp(b_{n,i})}{\alpha!} (\Delta W_{n,2}\boldsymbol{p}_{n,ij})^\alpha,
$$

for any $|\alpha| = 1$.

Moving to the term $\widetilde{A}_{n,2}(\boldsymbol{X})$, by applying the second-order Taylor expansions to $E(\boldsymbol{X}; W_{n,1}\boldsymbol{p}_{n,i})$ around $E(\boldsymbol{X}; W_{*,1}\boldsymbol{p}_{*,j})$ and $H(W_{n,2}\boldsymbol{p}_{n,i})$ around $H(W_{*,2}\boldsymbol{p}_{*,j})$ for any $i \in \mathcal{V}_j$ and $j$ such that $|\mathcal{V}_j| > 1$, we obtain that

$$
\begin{aligned}
\widetilde{A}_{n,2}(\boldsymbol{X}) = \sum_{j:|\mathcal{V}_j|>1} \sum_{1 \le |\alpha| \le 2} &\left\{ M_{n,j,\alpha}^{(1)} \frac{\partial^{|\alpha|}E}{\partial(W_1\boldsymbol{p})^\alpha}(\boldsymbol{X}; W_{*,1}\boldsymbol{p}_{*,j}) H(W_{*,2}\boldsymbol{p}_{*,j}) \right. \\
&\left. + M_{n,j,\alpha}^{(2)} \frac{\partial^{|\alpha|}H}{\partial(W_2\boldsymbol{p})^\alpha}(W_{*,2}\boldsymbol{p}_{*,j}) E(\boldsymbol{X}; W_{*,1}\boldsymbol{p}_{*,j}) \right\} \\
&+ \sum_{|\alpha|=1,|\beta|=1} M_{n,j,\alpha,\beta} \frac{\partial^{|\alpha|}E}{\partial(W_1\boldsymbol{p})^\alpha}(\boldsymbol{X}; W_{*,1}\boldsymbol{p}_{*,j}) \frac{\partial^{|\beta|}H}{\partial(W_2\boldsymbol{p})^\beta}(W_{*,2}\boldsymbol{p}_{*,j}) + \bar{R}_{n,2}(\boldsymbol{X})
\end{aligned}
$$

where $\bar{R}_{n,2}(\boldsymbol{X})$ satisfies $\bar{R}_{n,2}(\boldsymbol{X})/\mathcal{D}_{3n} \to 0$ when $n \to \infty$. Furthermore, we define

$$
M_{n,j,\alpha}^{(1)} = \sum_{i \in \mathcal{V}_j} \frac{\exp(b_{n,i})}{\alpha!} (\Delta W_{n,1}\boldsymbol{p}_{n,ij})^\alpha,
$$

$$
M_{n,j,\alpha}^{(2)} = \sum_{i \in \mathcal{V}_j} \frac{\exp(b_{n,i})}{\alpha!} (\Delta W_{n,2}\boldsymbol{p}_{n,ij})^\alpha,
$$

for any $|\alpha| = 2$ and

$$
M_{n,j,\alpha,\beta} = \sum_{i \in \mathcal{V}_j} \frac{\exp(b_{n,i})}{\alpha!\beta!} (\Delta W_{n,1}\boldsymbol{p}_{n,ij})^\alpha (\Delta W_{n,2}\boldsymbol{p}_{n,ij})^\beta,
$$

for any $|\alpha| = |\beta| = 1$. Direct calculation leads to the following formulations of the partial derivatives of $E(\boldsymbol{X}; W_1\boldsymbol{p})$ and $H(W_2\boldsymbol{p})$:

$$
\frac{\partial E}{\partial(W_1\boldsymbol{p})^{(u)}}(\boldsymbol{X}; W_1\boldsymbol{p}) = \exp((B\bar{\sigma}_1(W_1\boldsymbol{p}))^\top \boldsymbol{X})(B\frac{\partial\bar{\sigma}_1}{\partial(W_1\boldsymbol{p})^{(u)}}(W_1\boldsymbol{p}))^\top \boldsymbol{X},
$$

$$
\begin{aligned}
\frac{\partial^2 E}{\partial(W_1\boldsymbol{p})^{(u)}\partial(W_1\boldsymbol{p})^{(v)}}(\boldsymbol{X}; W_1\boldsymbol{p}) = \exp((B\bar{\sigma}_1(W_1\boldsymbol{p}))^\top \boldsymbol{X}) &\left\{ (B\frac{\partial^2\bar{\sigma}_1}{\partial(W_1\boldsymbol{p})^{(u)}\partial(W_1\boldsymbol{p})^{(v)}}(W_1\boldsymbol{p}))^\top \boldsymbol{X} \right. \\
&\left. + \boldsymbol{X}^\top (B\frac{\partial\bar{\sigma}_1}{\partial(W_1\boldsymbol{p})^{(u)}}(W_1\boldsymbol{p}))(B\frac{\partial\bar{\sigma}_1}{\partial(W_1\boldsymbol{p})^{(v)}}(W_1\boldsymbol{p}))^\top \boldsymbol{X} \right\},
\end{aligned}
$$

$$
\frac{\partial H}{\partial(W_2\boldsymbol{p})^{(u)}}(W_2\boldsymbol{p}) = C\frac{\partial\bar{\sigma}_2}{\partial(W_2\boldsymbol{p})^{(u)}}(W_2\boldsymbol{p}),
$$

$$
\frac{\partial^2 H}{\partial(W_2\boldsymbol{p})^{(u)}\partial(W_2\boldsymbol{p})^{(v)}}(W_2\boldsymbol{p}) = C\frac{\partial^2\bar{\sigma}_2}{\partial(W_2\boldsymbol{p})^{(u)}\partial(W_2\boldsymbol{p})^{(v)}}(W_2\boldsymbol{p}).
$$

Given the above formulations, we can rewrite $\widetilde{A}_{n,1}(\boldsymbol{X})$ and $\widetilde{A}_{n,2}(\boldsymbol{X})$ as follows:

$$
\widetilde{A}_{n,1}(\boldsymbol{X}) = \sum_{j:|\mathcal{V}_j|=1} \exp((B\bar{\sigma}_1(\boldsymbol{p}_{*,j}))^\top \boldsymbol{X}) \left[ L_{1,n}(\boldsymbol{p}_{*,j}) + L_{2,n}(\boldsymbol{p}_{*,j})^\top B^\top \boldsymbol{X} \right) + \bar{R}_{n,1}(\boldsymbol{X}),
$$

$$
\begin{aligned}
\widetilde{A}_{n,2}(\boldsymbol{X}) = \sum_{j:|\mathcal{V}_j|>1} \exp((B\bar{\sigma}_1(\boldsymbol{p}_{*,j}))^\top \boldsymbol{X}) &\left[ \bar{L}_{1,n}(\boldsymbol{p}_{*,j}) + \bar{L}_{2,n}(\boldsymbol{p}_{*,j})^\top B^\top \boldsymbol{X} \right. \\
&\left. + (B^\top \boldsymbol{X})^\top \bar{L}_{3,n}(\boldsymbol{p}_{*,j}) B^\top \boldsymbol{X} \right] + \bar{R}_{n,2}(\boldsymbol{X}),
\end{aligned}
$$

where the formulations of the functions $L_{1,n}$, $L_{2,n}$, $\bar{L}_{1,n}$, $\bar{L}_{2,n}$, and $\bar{L}_{3,n}$ are given by:

$$L_{1,n}(\boldsymbol{p}) = \sum_{u=1}^{d} M_{n,j,1_u}^{(2)} C \frac{\partial \bar{\sigma}_2}{\partial (W_2 \boldsymbol{p})^{(u)}} (W_2 \boldsymbol{p}),$$

$$L_{2,n}(\boldsymbol{p}) = \sum_{u=1}^{d} M_{n,j,1_u}^{(1)} \frac{\partial \bar{\sigma}_1}{\partial (W_1 \boldsymbol{p})^{(u)}} (W_1 \boldsymbol{p}) C \bar{\sigma}_2 (W_2 \boldsymbol{p}),$$

$$\bar{L}_{1,n}(\boldsymbol{p}) = \sum_{1 \leq u,v \leq d} M_{n,j,1_{uv}}^{(2)} C \frac{\partial^2 \bar{\sigma}_2}{\partial (W_2 \boldsymbol{p})^{(u)} \partial (W_2 \boldsymbol{p})^{(v)}} (W_2 \boldsymbol{p}),$$

$$= \sum_{u=1}^{d} M_{n,j,1_{uu}}^{(2)} C \frac{\partial^2 \bar{\sigma}_2}{\partial (W_2 \boldsymbol{p})^{(u)} \partial (W_2 \boldsymbol{p})^{(u)}} (W_2 \boldsymbol{p}),$$

$$\bar{L}_{2,n}(\boldsymbol{p}) = \sum_{u=1}^{d} M_{n,j,1_u}^{(1)} \frac{\partial \bar{\sigma}_1}{\partial (W_1 \boldsymbol{p})^{(u)}} (W_1 \boldsymbol{p}) C \bar{\sigma}_2 (W_2 \boldsymbol{p})$$

$$+ \sum_{1 \leq u,v \leq d} \Big[ M_{n,j,1_v,1_u} C \frac{\partial \bar{\sigma}_2}{\partial (W_2 \boldsymbol{p})^{(u)}} (\boldsymbol{p}) \frac{\partial \bar{\sigma}_1}{\partial (W_1 \boldsymbol{p})^{(v)}} (W_1 \boldsymbol{p})$$

$$+ M_{n,j,1_{uv}}^{(1)} \frac{\partial^2 \bar{\sigma}_1}{\partial (W_1 \boldsymbol{p})^{(u)} \partial (W_1 \boldsymbol{p})^{(v)}} (W_1 \boldsymbol{p}) C \bar{\sigma}_2 (W_2 \boldsymbol{p}) \Big],$$

$$\bar{L}_{3,n}(\boldsymbol{p}) = \sum_{1 \leq u,v \leq d} M_{n,j,1_{uv}}^{(1)} \frac{\partial \bar{\sigma}_1}{\partial (W_1 \boldsymbol{p})^{(u)}} (W_1 \boldsymbol{p}) \Big( \frac{\partial \bar{\sigma}_1}{\partial (W_1 \boldsymbol{p})^{(v)}} (W_1 \boldsymbol{p}) \Big)^{\top} C \bar{\sigma}_2 (W_2 \boldsymbol{p}).$$

Here, we denote $1_u$ is the vector that its $u$-th element is 1 while its other elements are 0 for any $1 \leq u \leq d$. Furthermore, $1_{uv}$ is the matrix that its $(u,v)$-th element is 1 while its other elements are 0 for any $1 \leq u,v \leq d$. The second equation in the formulation of $\bar{L}_{1,n}(\boldsymbol{p})$ is due to the fact that the function $\bar{\sigma}_2$ is only applied element wise to $W_2 p$, which leads to $\dfrac{\partial^2 \bar{\sigma}_2}{\partial (W_2 \boldsymbol{p})^{(u)} \partial (W_2 \boldsymbol{p})^{(v)}} (W_2 \boldsymbol{p}) = 0$ for all $u \neq v$.

**Decomposition of $\bar{B}_n(\boldsymbol{X})$.** We can rewrite $\bar{B}_n(\boldsymbol{X})$ as follows:

$$\bar{B}_n(\boldsymbol{X}) = \sum_{j:|\mathcal{V}_j|=1} \sum_{i \in \mathcal{V}_j} \exp(b_{n,i}) \Big[ E(\boldsymbol{X}; W_{n,1} \boldsymbol{p}_{n,i}) - E(\boldsymbol{X}; W_{*,1} \boldsymbol{p}_{*,j}) \Big] f_{G_n}(\boldsymbol{X})$$

$$+ \sum_{j:|\mathcal{V}_j|>1} \sum_{i \in \mathcal{V}_j} \exp(b_{n,i}) \Big[ E(\boldsymbol{X}; W_{n,1} \boldsymbol{p}_{n,i}) - E(\boldsymbol{X}; W_{*,1} \boldsymbol{p}_{*,j}) \Big] f_{G_n}(\boldsymbol{X})$$

$$:= \widetilde{B}_{n,1}(\boldsymbol{X}) + \widetilde{B}_{n,2}(\boldsymbol{X}).$$

By applying the first-order and second-order Taylor expansions, we get

$$\widetilde{B}_{n,1}(\boldsymbol{X}) = \sum_{j:|\mathcal{V}_j|=1} \sum_{|\alpha|=1} M_{n,j,\alpha}^{(1)} \frac{\partial^{|\alpha|} E}{\partial (W_1 \boldsymbol{p})^{\alpha}} (\boldsymbol{X}; W_{*,1} \boldsymbol{p}_{*,j}) f_{\widetilde{G}_n}(\boldsymbol{X}) + R_{n,3}(\boldsymbol{X}),$$

$$\widetilde{B}_{n,2}(\boldsymbol{X}) = \sum_{j:|\mathcal{V}_j|=1} \sum_{1 \leq |\alpha| \leq 2} M_{n,j,\alpha}^{(1)} \frac{\partial^{|\alpha|} E}{\partial (W_1 \boldsymbol{p})^{\alpha}} (\boldsymbol{X}; W_{*,1} \boldsymbol{p}_{*,j}) f_{\widetilde{G}_n}(\boldsymbol{X}) + R_{n,4}(\boldsymbol{X})$$

where $R_{n,3}(\boldsymbol{X})$, $R_{n,4}(\boldsymbol{X})$ is a Taylor remainder such that $R_{n,3}(\boldsymbol{X})/\mathcal{D}_{3n} \to 0$, $R_{n,4}(\boldsymbol{X})/\mathcal{D}_{3n} \to 0$ when $n \to \infty$. Therefore, we can express the functions $\widetilde{B}_{n,1}(\boldsymbol{X})$ and $\widetilde{B}_{n,2}(\boldsymbol{X})$ as follows:

$$\widetilde{B}_{n,1}(\boldsymbol{X}) = \sum_{j:|\mathcal{V}_j|=1} \exp((B\sigma_1(\boldsymbol{p}_{*,j}))^{\top} \boldsymbol{X}) N_{1,n}(\boldsymbol{p}_{*,j})^{\top} B^{\top} \boldsymbol{X} f_{\widetilde{G}_n}(\boldsymbol{X}) + R_{n,3}(\boldsymbol{X}),$$

$$\widetilde{B}_{n,2}(\boldsymbol{X}) = \sum_{j:|\mathcal{V}_j|>1} \exp((B\sigma_1(\boldsymbol{p}_{*,j}))^{\top} \boldsymbol{X}) \Big[ \bar{N}_{1,n}(\boldsymbol{p}_{*,j})^{\top} B^{\top} \boldsymbol{X} + (B^{\top} \boldsymbol{X})^{\top} \bar{N}_{2,n}(\boldsymbol{p}_{*,j}) B^{\top} \boldsymbol{X} \Big] f_{\widetilde{G}_n}(\boldsymbol{X})$$

$$+ R_{n,4}(\boldsymbol{X}),$$

where the formulations of the functions $N_{1,n}$, $\bar{N}_{1,n}$, and $\bar{N}_{2,n}$ are given by:

$$N_{1,n}(\boldsymbol{p}) = \sum_{u=1}^{d} M_{n,j,1_u}^{(1)} \frac{\partial \bar{\sigma}_1}{\partial (W_1\boldsymbol{p})^{(u)}}(W_1\boldsymbol{p}),$$

$$\bar{N}_{1,n}(\boldsymbol{p}) = \sum_{u=1}^{d} M_{n,j,1_u}^{(1)} \frac{\partial \bar{\sigma}_1}{\partial (W_1\boldsymbol{p})^{(u)}}(W_1\boldsymbol{p})$$
$$+ \sum_{1 \le u,v \le d} M_{n,j,1_{uv}}^{(1)} \frac{\partial^2 \bar{\sigma}_1}{\partial (W_1\boldsymbol{p})^{(u)} \partial (W_1\boldsymbol{p})^{(v)}}(W_1\boldsymbol{p}),$$

$$\bar{N}_{2,n}(\boldsymbol{p}) = \sum_{1 \le u,v \le d} M_{n,j,1_{uv}}^{(1)} \frac{\partial \bar{\sigma}_1}{\partial (W_1\boldsymbol{p})^{(u)}}(W_1\boldsymbol{p}) \frac{\partial \bar{\sigma}_1}{\partial (W_1\boldsymbol{p})^{(v)}}(W_1\boldsymbol{p})^{\top}.$$

Plugging the above expressions into equation (34), we can represent $\widetilde{Q}_n(\boldsymbol{X})$ as follows:

$$\widetilde{Q}_n(\boldsymbol{X}) = \sum_{j:|\mathcal{V}_j|=1} \exp((B\bar{\sigma}_1(W_{*,1}\boldsymbol{p}_{*,j}))^{\top}\boldsymbol{X}) \big[ L_{1,n}(\boldsymbol{p}_{*,j}) + L_{2,n}(\boldsymbol{p}_{*,j})^{\top}B^{\top}\boldsymbol{X} \big)$$

$$+ \sum_{j:|\mathcal{V}_j|>1} \exp((B\bar{\sigma}_1(W_{*,1}\boldsymbol{p}_{*,j}))^{\top}\boldsymbol{X}) \big[ \bar{L}_{1,n}(\boldsymbol{p}_{*,j}) + \bar{L}_{2,n}(\boldsymbol{p}_{*,j})^{\top}B^{\top}\boldsymbol{X} + (B^{\top}\boldsymbol{X})^{\top}\bar{L}_{3,n}(\boldsymbol{p}_{*,j})B^{\top}\boldsymbol{X} \big]$$

$$- \sum_{j:|\mathcal{V}_j|=1} \exp((B\bar{\sigma}_1(W_{*,1}\boldsymbol{p}_{*,j}))^{\top}\boldsymbol{X}) N_{1,n}(\boldsymbol{p}_{*,j})^{\top}B^{\top}\boldsymbol{X} f_{\widetilde{G}_n}(\boldsymbol{X})$$

$$- \sum_{j:|\mathcal{V}_j|>1} \exp((B\bar{\sigma}_1(W_{*,1}\boldsymbol{p}_{*,j}))^{\top}\boldsymbol{X}) \big[ \bar{N}_{1,n}(\boldsymbol{p}_{*,j})^{\top}B^{\top}\boldsymbol{X} + (B^{\top}\boldsymbol{X})^{\top}\bar{N}_{2,n}(\boldsymbol{p}_{*,j})B^{\top}\boldsymbol{X} \big] f_{\widetilde{G}_n}(\boldsymbol{X})$$

$$- \sum_{j=1}^{L} M_{n,j,0_d} \exp((B\bar{\sigma}_1(W_{*,1}\boldsymbol{p}_{*,j}))^{\top}\boldsymbol{X}) f_{\widetilde{G}_n}(\boldsymbol{X})$$

$$+ \sum_{j=1}^{L} M_{n,j,0_d} \exp((B\bar{\sigma}_1(W_{*,1}\boldsymbol{p}_{*,j}))^{\top}\boldsymbol{X}) C\bar{\sigma}_2(W_{*,2}\boldsymbol{p}_{*,j})$$

$$+ \bar{R}_{n,1}(\boldsymbol{X}) + \bar{R}_{n,2}(\boldsymbol{X}) - R_{n,3}(\boldsymbol{X}) - R_{n,4}(\boldsymbol{X})$$

$$= \sum_{j:|\mathcal{V}_j|=1} \exp((B\bar{\sigma}_1(W_{*,1}\boldsymbol{p}_{*,j}))^{\top}\boldsymbol{X}) \big[ L'_{1,n}(\boldsymbol{p}_{*,j}) + L_{2,n}(\boldsymbol{p}_{*,j})^{\top}B^{\top}\boldsymbol{X} \big)$$

$$+ \sum_{j:|\mathcal{V}_j|>1} \exp((B\bar{\sigma}_1(W_{*,1}\boldsymbol{p}_{*,j}))^{\top}\boldsymbol{X}) \big[ \bar{L}'_{1,n}(\boldsymbol{p}_{*,j}) + \bar{L}_{2,n}(\boldsymbol{p}_{*,j})^{\top}B^{\top}\boldsymbol{X} + (B^{\top}\boldsymbol{X})^{\top}\bar{L}_{3,n}(\boldsymbol{p}_{*,j})B^{\top}\boldsymbol{X} \big]$$

$$- \sum_{j:|\mathcal{V}_j|=1} \exp((B\bar{\sigma}_1(W_{*,1}\boldsymbol{p}_{*,j}))^{\top}\boldsymbol{X}) \big[ M_{n,j,0_d} + N_{1,n}(\boldsymbol{p}_{*,j})^{\top}B^{\top}\boldsymbol{X} \big] f_{\widetilde{G}_n}(\boldsymbol{X})$$

$$- \sum_{j:|\mathcal{V}_j|>1} \exp((B\bar{\sigma}_1(W_{*,1}\boldsymbol{p}_{*,j}))^{\top}\boldsymbol{X}) \big[ M_{n,j,0_d} + \bar{N}_{1,n}(\boldsymbol{p}_{*,j})^{\top}B^{\top}\boldsymbol{X} + (B^{\top}\boldsymbol{X})^{\top}\bar{N}_{2,n}(\boldsymbol{p}_{*,j})B^{\top}\boldsymbol{X} \big] f_{\widetilde{G}_n}(\boldsymbol{X})$$

$$+ \bar{R}_{n,1}(\boldsymbol{X}) + \bar{R}_{n,2}(\boldsymbol{X}) - R_{n,3}(\boldsymbol{X}) - R_{n,4}(\boldsymbol{X}), \tag{35}$$

where $M_{n,j,0_d} = \sum_{i \in \mathcal{V}_j} \exp(b_{n,i}) - \exp(b_{*,j})$ for any $j \in [L]$, $L'_{1,n}(\boldsymbol{p}_{*,j}) = L_{1,n}(\boldsymbol{p}_{*,j}) + M_{n,j,0_d} C\bar{\sigma}_2(W_{*,2}\boldsymbol{p}_{*,j})$, and $\bar{L}'_{1,n}(\boldsymbol{p}_{*,j}) = \bar{L}_{1,n}(\boldsymbol{p}_{*,j}) + M_{n,j,0_d} C\bar{\sigma}_2(W_{*,2}\boldsymbol{p}_{*,j})$.

**Step 2 - Non-vanishing coefficients.** From equation (35), we can represent $\widetilde{Q}_n(\boldsymbol{X})/\mathcal{D}_{3n}$ as a linear combination of the following independent functions:

$$\exp((B\bar{\sigma}_1(W_{*,1}\boldsymbol{p}_{*,j}))^{\top}\boldsymbol{X}), \ (B^{\top}\boldsymbol{X})^{(u)} \exp((B\bar{\sigma}_1(W_{*,1}\boldsymbol{p}_{*,j}))^{\top}\boldsymbol{X}),$$

$$(B^{\top}\boldsymbol{X})^{(u)}(B^{\top}\boldsymbol{X})^{(v)} \exp((B\bar{\sigma}_1(W_{*,1}\boldsymbol{p}_{*,j}))^{\top}\boldsymbol{X}), \ \exp((B\bar{\sigma}_1(W_{*,1}\boldsymbol{p}_{*,j}))^{\top}\boldsymbol{X}) f_{\widetilde{G}_n}(\boldsymbol{X}),$$

$$(B^{\top}\boldsymbol{X})^{(u)} \exp((B\bar{\sigma}_1(W_{*,1}\boldsymbol{p}_{*,j}))^{\top}\boldsymbol{X}) f_{\widetilde{G}_n}(\boldsymbol{X}), \ (B^{\top}\boldsymbol{X})^{(u)}(B^{\top}\boldsymbol{X})^{(v)} \exp((B\bar{\sigma}_1(W_{*,1}\boldsymbol{p}_{*,j}))^{\top}\boldsymbol{X}) f_{\widetilde{G}_n}(\boldsymbol{X})$$

for any $1 \leq j \leq L$ and $1 \leq u, v \leq d$.

Assume that all the coefficients of these linear independent functions in the formulation of $\widetilde{Q}_n(\boldsymbol{X})/\mathcal{D}_{3n}$ go to 0 as $n \to \infty$. It follows that $L'_{1,n}(\boldsymbol{p}_{*,j})/\mathcal{D}_{3n}$, $L_{2,n}(\boldsymbol{p}_{*,j})^{(u)}/\mathcal{D}_{3n}$, $\bar{L}'_{1,n}(\boldsymbol{p}_{*,j})/\mathcal{D}_{3n}$, $\bar{L}_{2,n}(\boldsymbol{p}_{*,j})^{(u)}/\mathcal{D}_{3n}$, $\bar{L}_{3,n}(\boldsymbol{p}_{*,j})^{(uv)}/\mathcal{D}_{3n}$, $N_{1,n}(\boldsymbol{p}_{*,j})/\mathcal{D}_{3n}$, $\bar{N}_{1,n}((\boldsymbol{p}_{*,j})^{(u)}/\mathcal{D}_{3n}$, $\bar{N}_{2,n}(\boldsymbol{p}_{*,j})^{(uv)}/\mathcal{D}_{3n}$, and $M_{n,j,0_d}/\mathcal{D}_{3n}$ approach 0 as $n \to \infty$ for any $1 \leq u, v \leq d$ and $1 \leq j \leq L$.

Then, as $M_{n,j,0_d}/\mathcal{D}_{3n} \to 0$, it indicates that

$$\frac{|M_{n,j,0_d}|}{\mathcal{D}_{2n}} = \frac{|\sum_{i \in \mathcal{V}_j} \exp(b_{n,i}) - \exp(b_{*,j})|}{\mathcal{D}_{3n}} \to 0,$$

for any $1 \leq j \leq L$. By summing these limits up when varying the index $j$ from 1 to $L$, we obtain that

$$\frac{\sum_{j=1}^{L} |\sum_{i \in \mathcal{V}_j} \exp(b_{n,i}) - \exp(b_{*,j})|}{\mathcal{D}_{3n}} \to 0. \tag{36}$$

Now, we consider indices $j \in [L]$ such that its corresponding Voronoi cell has only one element, i.e. $|\mathcal{V}_j| = 1$. As $L_{2,n}(\boldsymbol{p}_{*,j})^{(u)}/\mathcal{D}_{3n} \to 0$ and the first order derivatives of $\bar{\sigma}_1$ are non-zero, it indicates that $M_{n,j,1_u}^{(1)}/\mathcal{D}_{3n} \to 0$. It indicates that

$$\frac{\sum_{u=1}^{d} |M_{n,j,1_u}^{(1)}|}{\mathcal{D}_{2n}} = \frac{\sum_{i \in \mathcal{V}_j} \exp(b_{n,i})\|\Delta W_{n,1}p_{n,ij}\|}{\mathcal{D}_{3n}} \to 0.$$

Similarly, $L_{1,n}(\boldsymbol{p}_{*,j})/\mathcal{D}_{3n} \to 0$ also leads to $\dfrac{\sum_{i \in \mathcal{V}_j} \exp(b_{n,i})\|\Delta W_{n,2}p_{n,ij}\|}{\mathcal{D}_{3n}} \to 0$. Putting the above results together, we find that

$$\frac{\sum_{j:|\mathcal{V}_j|=1} \sum_{i \in \mathcal{V}_j} \exp(b_{n,i})(\|\Delta W_{n,1}p_{n,ij}\| + \|\Delta W_{n,2}p_{n,ij}\|}{\mathcal{D}_{3n}} \to 0. \tag{37}$$

Moving to indices $j \in [L]$ such that $|\mathcal{V}_j| > 1$, as $\bar{L}_{3,n}(\boldsymbol{p}_{*,j})^{(uu)}/\mathcal{D}_{3n} \to 0$, we obtain that

$$\frac{\sum_{u=1}^{d} \bar{L}_{3,n}(\boldsymbol{p}_{*,j})^{(uu)}}{\mathcal{D}_{3n}} = \frac{\sum_{i \in \mathcal{V}_j} \exp(b_{n,i})\|\Delta W_{n,1}\boldsymbol{p}_{n,ij}\|^2}{\mathcal{D}_{3n}} \to 0.$$

Likewise, as $\bar{L}_{1,n}(\boldsymbol{p}_{*,j})^{(uu)}/\mathcal{D}_{3n} \to 0$ and the second order derivatives of $\bar{\sigma}_2$ are non-zero, we also obtain that $\dfrac{\sum_{i \in \mathcal{V}_j} \exp(b_{n,i})\|\Delta W_{n,2}\boldsymbol{p}_{n,ij}\|^2}{\mathcal{D}_{3n}} \to 0$. Therefore, we find that

$$\frac{\sum_{j:|\mathcal{V}_j|>1} \sum_{i \in \mathcal{V}_j} \exp(b_{n,i})(\|\Delta W_{n,1}p_{n,ij}\|^2 + \|\Delta W_{n,2}p_{n,ij}\|^2}{\mathcal{D}_{3n}} \to 0.$$

Collecting all the above results, we obtain that

$$1 = \frac{\mathcal{D}_{3n}}{\mathcal{D}_{3n}} \to 0$$

as $n \to \infty$, which is a contradiction.

As a consequence, not all of the coefficients of the linear independent functions in the formulations of $\widetilde{Q}_n(\boldsymbol{X})/\mathcal{D}_{3n}$ go to 0 as $n \to \infty$.

**Step 3 - Application of Fatou's lemma.** Let us denote $m_n$ as the maximum of the absolute values of $\bar{L}'_{1,n}(\boldsymbol{p}_{*,j})/\mathcal{D}_{3n}$, $L_{2,n}(\boldsymbol{p}_{*,j})^{(u)}/\mathcal{D}_{3n}$, $\bar{L}'_{1,n}(\boldsymbol{p}_{*,j})/\mathcal{D}_{3n}$, $\bar{L}_{2,n}(\boldsymbol{p}_{*,j})^{(u)}/\mathcal{D}_{3n}$, $\bar{L}_{3,n}(\boldsymbol{p}_{*,j})^{(uv)}/\mathcal{D}_{3n}$, $N_{1,n}(\boldsymbol{p}_{*,j})/\mathcal{D}_{3n}$, $\bar{N}_{1,n}((\boldsymbol{p}_{*,j})^{(u)}/\mathcal{D}_{3n}$, $\bar{N}_{2,n}(\boldsymbol{p}_{*,j})^{(uv)}/\mathcal{D}_{3n}$, and $M_{n,j,0_d}/\mathcal{D}_{3n}$ for all $1 \leq u, v \leq d$. From the result of Step 2, it follows that $1/m_n \not\to \infty$ as $n \to \infty$.

Since $\|f_{\widetilde{G}_n} - f_{\widetilde{G}_*}\|_{L^2(\mu)}/\mathcal{D}_{3n} \to 0$ as $n \to \infty$, we obtain $\|f_{\widetilde{G}_n} - f_{\widetilde{G}_*}\|_{L^2(\mu)}/(m_n\mathcal{D}_{3n}) \to 0$. By applying Fatou's lemma, we get that

$$0 = \lim_{n \to \infty} \frac{\|f_{\widetilde{G}_n} - f_{\widetilde{G}_*}\|_{L^2(\mu)}}{m_n\mathcal{D}_{3n}} \geq \int \liminf_{n \to \infty} \frac{\left|f_{\widetilde{G}_n}(\boldsymbol{X}) - f_{\widetilde{G}_*}(\boldsymbol{X})\right|}{m_n\mathcal{D}_{3n}} d\mu(\boldsymbol{X}) \geq 0.$$

Therefore, $\liminf_{n\to\infty} \dfrac{\left| f_{\widetilde{G}_n}(\boldsymbol{X}) - f_{\widetilde{G}_*}(\boldsymbol{X}) \right|}{m_n \mathcal{D}_{2n}} = 0$ for almost surely $\boldsymbol{X}$. As $n \to \infty$, we denote

$$\frac{L'_{1,n}(\boldsymbol{p}_{*,j})}{m_n \mathcal{D}_{3n}} \to \alpha_j, \quad \frac{L_{2,n}(\boldsymbol{p}_{*,j})}{m_n \mathcal{D}_{3n}} \to \beta_j,$$

$$\frac{\bar{L}'_{1,n}(\boldsymbol{p}_{*,j})}{m_n \mathcal{D}_{3n}} \to \bar{\alpha}_j, \quad \frac{\bar{L}_{2,n}(\boldsymbol{p}_{*,j})}{m_n \mathcal{D}_{3n}} \to \bar{\beta}_j, \quad \frac{\bar{L}_{3,n}(\boldsymbol{p}_{*,j})}{m_n \mathcal{D}_{3n}} \to \bar{\gamma}_j,$$

$$\frac{M_{n,j,0_d}}{\mathcal{D}_{3n}} \to \tilde{\alpha}_j, \quad \frac{N_{1,n}(\boldsymbol{p}_{*,j})}{m_n \mathcal{D}_{3n}} \to \tilde{\beta}_j,$$

$$\frac{\bar{N}_{1,n}(\boldsymbol{p}_{*,j})}{m_n \mathcal{D}_{3n}} \to \widehat{\beta}_j, \quad \frac{\bar{N}_{2,n}(\boldsymbol{p}_{*,j})}{m_n \mathcal{D}_{3n}} \to \widehat{\gamma}_j$$

for any $1 \leq j \leq L$. Here, from the definition of $m_n$, at least one coefficient among $\{\alpha_j, \beta_j, \tilde{\alpha}_j, \tilde{\beta}_j\}_{j:|\mathcal{V}_j|=1}$, $\{\bar{\alpha}_j, \bar{\beta}_j, \bar{\gamma}_j, \tilde{\alpha}_j, \widehat{\beta}_j, \widehat{\gamma}_j\}_{j:|\mathcal{V}_j|>1}$ is different from 0. Then, the equation $\liminf_{n\to\infty} \dfrac{\left| f_{\widetilde{G}_n}(\boldsymbol{X}) - f_{\widetilde{G}_*}(\boldsymbol{X}) \right|}{m_n \mathcal{D}_{3n}} = 0$ leads to

$$\sum_{j:|\mathcal{V}_j|=1} \exp((B\bar{\sigma}_1(W_{*,1}\boldsymbol{p}_{*,j}))^\top \boldsymbol{X})(\alpha_j + \beta_j^\top (B^\top \boldsymbol{X}))$$

$$+ \sum_{j:|\mathcal{V}_j|>1} \exp((B\bar{\sigma}_1(W_{*,1}\boldsymbol{p}_{*,j}))^\top \boldsymbol{X})\left[\bar{\alpha}_j + \bar{\beta}_j^\top (B^\top \boldsymbol{X}) + (B^\top \boldsymbol{X})^\top \bar{\gamma}_j (B^\top \boldsymbol{X})\right]$$

$$- \sum_{j:|\mathcal{V}_j|=1} \exp((B\bar{\sigma}_1(W_{*,1}\boldsymbol{p}_{*,j}))^\top \boldsymbol{X})(\tilde{\alpha}_j + \tilde{\beta}_j^\top (B^\top \boldsymbol{X}))f_{\widetilde{G}_*}(\boldsymbol{X})$$

$$- \sum_{j:|\mathcal{V}_j|>1} \exp((B\bar{\sigma}_1(W_{*,1}\boldsymbol{p}_{*,j}))^\top \boldsymbol{X})\left[\tilde{\alpha}_j + \widehat{\beta}_j^\top (B^\top \boldsymbol{X}) + (B^\top \boldsymbol{X})^\top \widehat{\gamma}_j B^\top \boldsymbol{X}\right]f_{\widetilde{G}_*}(\boldsymbol{X}) = 0$$

for almost surely $\boldsymbol{X}$. By denoting $\boldsymbol{Z} = B^\top \boldsymbol{X}$, this equation also holds for almost surely $\boldsymbol{Z}$. However, the new equation implies that all the coefficients $\{\alpha_j, \beta_j, \tilde{\alpha}_j, \tilde{\beta}_j\}_{j:|\mathcal{V}_j|=1}$, $\{\bar{\alpha}_j, \bar{\beta}_j, \bar{\gamma}_j, \tilde{\alpha}_j, \widehat{\beta}_j, \widehat{\gamma}_j\}_{j:|\mathcal{V}_j|>1}$ are 0, which is a contradiction.

As a consequence, we obtain

$$\lim_{\varepsilon \to 0} \inf_{\widetilde{G} \in \widetilde{\mathcal{G}}_{L'}(\Xi):\mathcal{D}_3(\widetilde{G},\widetilde{G}_*) \leq \varepsilon} \|f_{\widetilde{G}} - f_{\widetilde{G}_*}\|_{L^2(\mu)}/\mathcal{D}_3(\widetilde{G},\widetilde{G}_*) > 0.$$

**Global part:** From local part, there exists a positive constant $\varepsilon'$ such that

$$\inf_{\widetilde{G} \in \widetilde{\mathcal{G}}_{L'}(\Xi):\mathcal{D}_3(\widetilde{G},\widetilde{G}_*) \leq \varepsilon'} \|f_{\widetilde{G}} - f_{\widetilde{G}_*}\|_{L^2(\mu)}/\mathcal{D}_3(\widetilde{G},\widetilde{G}_*) > 0.$$

Therefore, it is sufficient to prove that

$$\inf_{\widetilde{G} \in \widetilde{\mathcal{G}}_{L'}(\Xi):\mathcal{D}_3(\widetilde{G},\widetilde{G}_*) > \varepsilon'} \|f_{\widetilde{G}} - f_{\widetilde{G}_*}\|_{L^2(\mu)}/\mathcal{D}_3(\widetilde{G},\widetilde{G}_*) > 0.$$

Assume by contrary, then we can find a sequence of mixing measures $\widetilde{G}'_n := \sum_{j'=1}^{L'} \exp(b_{n,j'})\delta_{(W_{n,1}\boldsymbol{p}_{n,j'},W_{n,2}\boldsymbol{p}_{n,j'})}$ in $\widetilde{\mathcal{G}}_{L'}(\Xi)$ such that as $n \to \infty$, we have

$$\begin{cases} \mathcal{D}_3(\widetilde{G}'_n,\widetilde{G}_*) > \varepsilon' \\ \|f_{\widetilde{G}'_n} - f_{\widetilde{G}_*}\|_{L^2(\mu)}/\mathcal{D}_3(\widetilde{G}'_n,\widetilde{G}_*) \to 0, \end{cases}$$

which indicates that $\|f_{\widetilde{G}'_n} - f_{\widetilde{G}_*}\|_{L^2(\mu)} \to 0$ as $n \to \infty$.

Since $\Xi$ is a compact set, there exists a mixing measure $\widetilde{G}'$ in $\widetilde{\mathcal{G}}_{L'}(\Xi)$ such that one of $\widetilde{G}'_n$'s subsequences converges to $G\widetilde{G}'$. Since $\mathcal{D}_3(G\widetilde{G}'_n,\widetilde{G}_*) > \varepsilon'$, we deduce that $\mathcal{D}_3(\widetilde{G}',\widetilde{G}_*) > \varepsilon'$.
By invoking the Fatou's lemma, we have that

$$0 = \lim_{n\to\infty} \|f_{\widetilde{G}'_n} - f_{\widetilde{G}_*}\|_{L^2(\mu)} \geq \int \liminf_{n\to\infty} \left| f_{\widetilde{G}'_n}(\boldsymbol{X}) - f_{\widetilde{G}_*}(\boldsymbol{X}) \right|^2 d\mu(\boldsymbol{X}).$$

Thus, we have $f_{\widetilde{G}'} = f_{\widetilde{G}_*}$ for $\mu-$almost surely $\boldsymbol{X}$. From the identifiability property, we deduce that $\widetilde{G}' \equiv \widetilde{G}_*$. It follows that $\mathcal{D}_3(\widetilde{G}', \widetilde{G}_*) = 0$, contradicting the fact that $\mathcal{D}_3(\widetilde{G}', \widetilde{G}_*) > \varepsilon' > 0$. Hence, the proof is completed.

**Identifiability property.** We now prove the identifiability of one layer neural network structures among prompts. In particular, we will show that if $f_{\widetilde{G}(\boldsymbol{X})} = f_{\widetilde{G}_*(\boldsymbol{X})}$ for almost every $\boldsymbol{X}$, then it follows that $\widetilde{G} \equiv \widetilde{G}_*$.

For any $\widetilde{G} \in \widetilde{\mathcal{G}}_{L'}(\Xi)$ and $\widetilde{G}_*$, let us denote

$$\text{softmax}_{\widetilde{G}}(u) = \frac{\exp(u)}{\sum_{k=1}^{N} \exp(\boldsymbol{X}^\top A_k^0 \boldsymbol{X} + a_k^0) + \sum_{j'=1}^{L'} \exp((B\bar{\sigma}_1(W_1 \boldsymbol{p}_{j'}))^\top \boldsymbol{X} + b_{j'})},$$

$$\text{softmax}_{\widetilde{G}_*}(u_*) = \frac{\exp(u_*)}{\sum_{k=1}^{N} \exp(\boldsymbol{X}^\top A_k^0 \boldsymbol{X} + a_k^0) + \sum_{j'=1}^{L} \exp((B\bar{\sigma}_1(W_{*,1} \boldsymbol{p}_{*,j}))^\top \boldsymbol{X} + b_{*,j'})},$$

where

$$u \in \{\boldsymbol{X}^\top A_j^0 \boldsymbol{X} + a_j^0; (B\bar{\sigma}_1(W_1 \boldsymbol{p}_{j'}))^\top \boldsymbol{X} + b_{j'} : j \in [N], j' \in [L']\},$$
$$u_* \in \{\boldsymbol{X}^\top A_j^0 \boldsymbol{X} + a_j^0; (B\bar{\sigma}_1(W_{*,1} \boldsymbol{p}_{*,j'}))^\top \boldsymbol{X} + b_{*,j'} : j \in [N], j' \in [L]\}.$$

Since $f_{\widetilde{G}}(\boldsymbol{X}) = f_{\widetilde{G}_*}(\boldsymbol{X})$ for almost every $\boldsymbol{X}$, we have

$$\sum_{j=1}^{N} \text{softmax}_{\widetilde{G}}(\boldsymbol{X}^\top A_j^0 \boldsymbol{X} + a_j^0)) h(\boldsymbol{X}, \eta_j^0) + \sum_{j'=1}^{L'} \text{softmax}_{\widetilde{G}}((B\bar{\sigma}_1(W_1 \boldsymbol{p}_{j'}))^\top \boldsymbol{X} + b_{j'}) C\bar{\sigma}_2(W_2 \boldsymbol{p}_{j'})$$

$$= \sum_{j=1}^{N} \text{softmax}_{\widetilde{G}_*}(\boldsymbol{X}^\top A_j^0 \boldsymbol{X} + a_j^0)) h(\boldsymbol{X}, \eta_j^0)$$

$$+ \sum_{j'=1}^{L} \text{softmax}_{\widetilde{G}_*}((B\bar{\sigma}_1(W_{*,1} \boldsymbol{p}_{*,j'}))^\top \boldsymbol{X} + b_{*,j'}) C\bar{\sigma}_2(W_{*,2} \boldsymbol{p}_{*,j'}). \tag{38}$$

Thus, we must have that $L = L'$. As a result, we obtain that
$$\{\text{softmax}_{\widetilde{G}}((B\bar{\sigma}_1(W_1 \boldsymbol{p}_{j'}))^\top \boldsymbol{X} + b_{j'}) : j' \in [L]\}$$
$$= \{\text{softmax}_{\widetilde{G}_*}((B\bar{\sigma}_1(W_{*,1} \boldsymbol{p}_{*,j'}))^\top \boldsymbol{X} + b_{*,j'}) : j' \in [L']\},$$

for almost every $\boldsymbol{X}$. We may assume that
$$\text{softmax}_{\widetilde{G}}((B\bar{\sigma}_1(W_1 \boldsymbol{p}_{j'}))^\top \boldsymbol{X} + b_{j'}) = \text{softmax}_{\widetilde{G}_*}((B\bar{\sigma}_1(W_{*,1} \boldsymbol{p}_{*,j'}))^\top \boldsymbol{X} + b_{*,j'}),$$

for any $j' \in [L]$, for almost every $\boldsymbol{X}$. Since the softmax function is invariant to translation, this result indicates that $b_{j'} = b_{*,j'} + r$ for some $r \in \mathbb{R}$ and for any $j' \in [L]$. Then, the equation (38) can be reduced to

$$\sum_{j=1}^{L} \exp(b_j) \exp((B\bar{\sigma}_1(W_1 \boldsymbol{p}_j))^\top \boldsymbol{X}) C\bar{\sigma}_2(W_2 \boldsymbol{p}_j)$$

$$= \sum_{j=1}^{L} \exp(b_{*,j}) \exp((B\bar{\sigma}_1(W_{*,1} \boldsymbol{p}_{*,j}))^\top \boldsymbol{X}) C\bar{\sigma}_2(W_{*,2} \boldsymbol{p}_{*,j}), \tag{39}$$

for almost every $\boldsymbol{X}$. Next, we will partition the index set $[L]$ into $m$ subsets $K_1, K_2, \ldots, K_m$ where $m \leq L$, such that $\exp(b_j) = \exp(b_{*,j'})$ for any $j, j' \in K_i$ and $i \in [m]$. It follows that $\exp(b_j) \neq \exp(b_{*,j'})$ when $j, j'$ do not belong to the same set $K_i$. Thus, we can rewrite equation (39) as

$$\sum_{i=1}^{m} \sum_{j \in K_i} \exp(b_j) \exp((B\bar{\sigma}_1(W_1 \boldsymbol{p}_j))^\top \boldsymbol{X}) C\bar{\sigma}_2(W_2 \boldsymbol{p}_j)$$

$$= \sum_{i=1}^{m} \sum_{j \in K_i} \exp(b_{*,j}) \exp((B\bar{\sigma}_1(W_{*,1} \boldsymbol{p}_{*,j}))^\top \boldsymbol{X}) C\bar{\sigma}_2(W_{*,2} \boldsymbol{p}_{*,j}),$$

for almost surely $\boldsymbol{X}$. Given the above equation, for each $i \in [m]$, we obtain that

$$\{((B\bar{\sigma}_1(W_1\boldsymbol{p}_j))^\top, W_2\boldsymbol{p}_j) : j \in K_i\} = \{((B\bar{\sigma}_1(W_{*,1}\boldsymbol{p}_{*,j}))^\top, W_{*,2}\boldsymbol{p}_{*,j}) : j \in K_i\},$$

which directly leads to

$$\{W_1\boldsymbol{p}_j : j \in K_i\} = \{W_{*,1}\boldsymbol{p}_{*,j} : j \in K_i\} \quad \text{and} \quad \{W_2\boldsymbol{p}_j : j \in K_i\} = \{W_{*,2}\boldsymbol{p}_{*,j} : j \in K_i\}.$$

Without loss of generality, we assume that $W_1\boldsymbol{p}_j = W_{*,1}\boldsymbol{p}_{*,j}$ and $W_2\boldsymbol{p}_j = W_{*,2}\boldsymbol{p}_{*,j}$ for all $j \in K_i$. Consequently, we get that

$$\sum_{i=1}^m \sum_{j \in K_i} \exp(b_j)\delta_{(W_1\boldsymbol{p}_j, W_2\boldsymbol{p}_j)} = \sum_{i=1}^m \sum_{j \in K_i} \exp(b_{*,j})\delta_{(W_{*,1}\boldsymbol{p}_{*,j}, W_{*,2}\boldsymbol{p}_{*,j})},$$

which implies that $\widetilde{G} \equiv \widetilde{G}_*$. As a consequence, the proof is completed.

# C    ADDITIONAL PROOFS

In this appendix, we provide proof for the convergence rate of regression function estimation.

## C.1    PROOF OF PROPOSITION B.1

To start with, it is necessary to define the notations that will be used throughout this proof. First of all, let us denote by $\mathcal{F}_{L'}(\Omega)$ the set of regression functions w.r.t mixing measures in $\bar{\mathcal{G}}_{L'}(\Omega)$, that is,

$$\mathcal{F}_{L'}(\Omega) := \{f_{\bar{G}}(\boldsymbol{X}) : G \in \bar{\mathcal{G}}_{L'}(\Omega)\}.$$

Next, for each $\delta > 0$, we define the $L^2(\mu)$ ball centered around the regression function $f_{\bar{G}_*}(\boldsymbol{X})$ and intersected with the set $\mathcal{F}_{L'}(\Omega)$ as

$$\mathcal{F}_{L'}(\Omega, \delta) := \left\{f \in \mathcal{F}_{L'}(\Theta) : \|f - f_{\bar{G}_*}\|_{L^2(\mu)} \leq \delta\right\}.$$

Furthermore, van de Geer (2000) suggest capturing the size of the above set by using the following quantity:

$$\mathcal{J}_B(\delta, \mathcal{F}_{L'}(\Omega, \delta)) := \int_{\delta^2/2^{13}}^\delta H_B^{1/2}(t, \mathcal{F}_{L'}(\Omega, t), \|\cdot\|_{L^2(\mu)})\, dt \vee \delta, \tag{40}$$

in which $H_B(t, \mathcal{F}_{L'}(\Omega, t), \|\cdot\|_{L^2(\mu)})$ denotes the bracketing entropy van de Geer (2000) of $\mathcal{F}_{L'}(\Omega, t)$ under the $L^2(\mu)$-norm and $t \vee \delta := \max\{t, \delta\}$.

Subsequently, let us introduce a key result of this proof in Lemma C.1, which is achieved by applying similar arguments as those in Theorem 7.4 and Theorem 9.2 in van de Geer (2000).

**Lemma C.1.** *Take $\Psi(\delta) \geq \mathcal{J}_B(\delta, \mathcal{F}_{L'}(\Omega, \delta))$ that satisfies $\Psi(\delta)/\delta^2$ is a non-increasing function of $\delta$. Then, for some universal constant $c$ and for some sequence $(\delta_n)$ such that $\sqrt{n}\delta_n^2 \geq c\Psi(\delta_n)$, we achieve that*

$$\mathbb{P}\Big(\|f_{\bar{G}_n} - f_{\bar{G}_*}\|_{L^2(\mu)} > \delta\Big) \leq c\exp\left(-\frac{n\delta^2}{c^2}\right),$$

*for all $\delta \geq \delta_n$.*

**General picture.** We begin with deriving the bracketing entropy inequality

$$H_B(\varepsilon, \mathcal{F}_{L'}(\Omega), \|\cdot\|_{L^2(\mu)}) \lesssim \log(1/\varepsilon), \tag{41}$$

for any $0 < \varepsilon \leq 1/2$. Then, it follows that

$$\mathcal{J}_B(\delta, \mathcal{F}_{L'}(\Omega, \delta)) = \int_{\delta^2/2^{13}}^\delta H_B^{1/2}(t, \mathcal{F}_{L'}(\Omega, t), \|\cdot\|_{L^2(\mu)})\, dt \vee \delta \lesssim \int_{\delta^2/2^{13}}^\delta \log(1/t)dt \vee \delta. \tag{42}$$

Let $\Psi(\delta) = \delta \cdot [\log(1/\delta)]^{1/2}$, then $\Psi(\delta)/\delta^2$ is a non-increasing function of $\delta$. Additionally, equation (42) indicates that $\Psi(\delta) \geq \mathcal{J}_B(\delta, \mathcal{F}_{L'}(\Omega, \delta))$. Moreover, by choosing $\delta_n = \sqrt{\log(n)/n}$, we

have that $\sqrt{n}\delta_n^2 \geq c\Psi(\delta_n)$ for some universal constant $c$. Then, according to Lemma C.1, we reach the conclusion of Theorem B.1.

As a result, it suffices to establish the inequality in equation (41).

**Proof of equation (41).** Let $f_{\bar{G}}$ be an arbitrary regression function in $\mathcal{F}_{L'}(\Omega)$. As the prompts $\boldsymbol{p}_{j'}$ are both bounded, we obtain that $|f_{\bar{G}}(\boldsymbol{X})| \leq M$ for all $\boldsymbol{X}$ where $M > 0$ is some universal constant.

Next, let $\tau \leq \varepsilon$ and $\{\pi_1, \ldots, \pi_{\bar{N}}\}$ be the $\tau$-cover under the $L^\infty$ norm of the set $\mathcal{F}_{L'}(\Omega)$ in which $\bar{N} := N(\tau, \mathcal{F}_{L'}(\Omega), \|\cdot\|_{L^2(\mu)})$ is the $\tau$-covering number of the metric space $(\mathcal{F}_k(\Omega), \|\cdot\|_{L^\infty(\mu)})$. Then, we construct the brackets of the form $[L_i(\boldsymbol{X}), U_i(\boldsymbol{X})]$ for all $i \in [\bar{N}]$ as follows:

$$L_i(\boldsymbol{X}) := \max\{\pi_i(\boldsymbol{X}) - \tau, 0\},$$
$$U_i(\boldsymbol{X}) := \max\{\pi_i(\boldsymbol{X}) + \tau, M\}.$$

It can be verified that $\mathcal{F}_{L'}(\Omega) \subset \cup_{i=1}^{\bar{N}}[L_i(\boldsymbol{X}), U_i(\boldsymbol{X})]$. Furthermore, we also get that

$$\|U_i - L_i\|_{L^2(\mu)} = \left(\int (U_i(\boldsymbol{X}) - L_i(\boldsymbol{X}))^2 \mathrm{d}\mu(\boldsymbol{X})\right)^{1/2} \leq \left(\int 4\tau^2 \mathrm{d}\mu(\boldsymbol{X})\right)^{1/2} = 2\tau,$$

From the definition of the bracketing entropy, we have that

$$H_B(2\tau, \mathcal{F}_{L'}(\Omega), \|\cdot\|_{L^2(\mu)}) \leq \log \bar{N} = \log N(\tau, \mathcal{F}_{L'}(\Omega), \|\cdot\|_{L^\infty}). \tag{43}$$

Thus, it is sufficient to establish an upper bound for the covering number $\bar{N}$. For that purpose, we denote $\Delta = \{(b, \boldsymbol{p}) \in \mathbb{R} \times \mathbb{R}^d : (b, \boldsymbol{p}) \in \Theta\}$. Since $\Omega$ is a compact set, $\Delta$ is also compact. Thus, there exist $\tau$-covers for $\Delta$, denoted by $\Delta_\tau$, respectively. Then, we find that

$$|\Delta_\tau| \leq \mathcal{O}(\tau^{-(d+1)L'}).$$

For each mixing measure $\bar{G} = \sum_{i=1}^{L'} \exp(b_i)\delta_{\boldsymbol{p}_i} \in \bar{\mathcal{G}}_{L'}(\Omega)$, we consider a corresponding mixing measure $\check{G}$ defined as

$$\check{G} := \sum_{i=1}^{L'} \exp(\check{b}_i)\delta_{\check{\boldsymbol{p}}_i},$$

where $(\check{b}_i, \check{\boldsymbol{p}}_i) \in \Delta_\tau$ is the closest to $(b_i, \boldsymbol{p}_i)$ in that set. Let us denote

$$D(\boldsymbol{X}) := \sum_{i'=1}^{N} \exp(\boldsymbol{X}^\top A_{i'}^0 \boldsymbol{X} + a_{i'}^0) + \sum_{j'=1}^{L'} \exp((B\boldsymbol{p}_{j'})^\top \boldsymbol{X} + b_{j'}),$$

$$\check{D}(\boldsymbol{X}) := \sum_{i'=1}^{N} \exp(\boldsymbol{X}^\top A_{i'}^0 \boldsymbol{X} + a_{i'}^0) + \sum_{j'=1}^{L'} \exp((B\check{\boldsymbol{p}}_{j'})^\top \boldsymbol{X} + \check{b}_{j'}).$$

Subsequently, we aim to show that $\|f_{\bar{G}} - f_{\check{G}}\|_\infty \lesssim \tau$. In particular, we have

$$\|f_{\bar{G}} - f_{\check{G}}\|_\infty \leq \sup_{\boldsymbol{X} \in \mathcal{X}} \left\|\sum_{j=1}^{L'} \frac{\exp((B\boldsymbol{p}_j)^\top \boldsymbol{X} + b_j)}{D(\boldsymbol{X})} \cdot C\boldsymbol{p}_j - \sum_{j=1}^{L'} \frac{\exp((B\check{\boldsymbol{p}}_j)^\top \boldsymbol{X} + \check{b}_j)}{\check{D}(\boldsymbol{X})} \cdot C\check{\boldsymbol{p}}_j\right\|$$

$$+ \sup_{\boldsymbol{X} \in \mathcal{X}} \left\|\sum_{j=1}^{N} \exp(\boldsymbol{X}^\top A_j^0 \boldsymbol{X} + a_j^0)h(\boldsymbol{X}, \eta_j^0)\left(\frac{1}{D(\boldsymbol{X})} - \frac{1}{\check{D}(\boldsymbol{X})}\right)\right\|$$

$$\leq \sup_{\boldsymbol{X} \in \mathcal{X}} \left\|\sum_{j=1}^{L'} \frac{\exp((B\boldsymbol{p}_j)^\top \boldsymbol{X} + b_j)}{D(\boldsymbol{X})} \cdot C(\boldsymbol{p}_j - \check{\boldsymbol{p}}_j)\right\|$$

$$+ \sup_{\boldsymbol{X} \in \mathcal{X}} \left\|\sum_{j=1}^{L'} \left[\frac{\exp((B\boldsymbol{p}_j)^\top \boldsymbol{X} + b_j)}{D(\boldsymbol{X})} - \frac{\exp((B\check{\boldsymbol{p}}_j)^\top \boldsymbol{X} + \check{b}_j)}{\check{D}(\boldsymbol{X})}\right] \cdot C\check{\boldsymbol{p}}_j\right\|$$

$$+ \sup_{\boldsymbol{X} \in \mathcal{X}} \left\|\sum_{j=1}^{N} \exp(\boldsymbol{X}^\top A_j^0 \boldsymbol{X} + a_j^0)h(\boldsymbol{X}, \eta_j^0)\left(\frac{1}{D(\boldsymbol{X})} - \frac{1}{\check{D}(\boldsymbol{X})}\right)\right\|$$

$$:= T_1 + T_2 + T_3.$$

Then, it is sufficient to demonstrate that $T_1 \lesssim \tau$ and $T_2 \lesssim \tau$, respectively. First of all, we get that

$$T_1 = \sup_{\boldsymbol{X} \in \mathcal{X}} \left\| \sum_{j=1}^{L'} \frac{\exp((B\boldsymbol{p}_j)^\top \boldsymbol{X} + b_j)}{D(\boldsymbol{X})} \cdot C(\boldsymbol{p}_j - \check{\boldsymbol{p}}_j) \right\|$$

$$\leq \sup_{\boldsymbol{X} \in \mathcal{X}} \left\| \sum_{j=1}^{L'} C(\boldsymbol{p}_j - \check{\boldsymbol{p}}_j) \right\| \leq C \cdot \sum_{j=1}^{L'} \|\boldsymbol{p}_j - \check{\boldsymbol{p}}_j\| \leq CL'\tau \lesssim \tau.$$

Here, the first inequality occurs as the softmax weight is bounded by 1. Next, we have

$$T_2 = \sup_{\boldsymbol{X} \in \mathcal{X}} \left\| \sum_{j=1}^{L'} \left[ \frac{\exp((B\boldsymbol{p}_j)^\top \boldsymbol{X} + b_j)}{D(\boldsymbol{X})} - \frac{\exp((B\check{\boldsymbol{p}}_j)^\top \boldsymbol{X} + \check{b}_j)}{\check{D}(\boldsymbol{X})} \right] \cdot C\check{\boldsymbol{p}}_j \right\|$$

$$\leq \sum_{j=1}^{L'} \sup_{\boldsymbol{X} \in \mathcal{X}} \left\| \left[ \frac{\exp((B\boldsymbol{p}_j)^\top \boldsymbol{X} + b_j)}{D(\boldsymbol{X})} - \frac{\exp((B\check{\boldsymbol{p}}_j)^\top \boldsymbol{X} + \check{b}_j)}{\check{D}(\boldsymbol{X})} \right] \cdot C\check{\boldsymbol{p}}_j \right\|$$

$$\leq \sum_{j=1}^{L'} \sup_{\boldsymbol{X} \in \mathcal{X}} \left\| \exp((B\boldsymbol{p}_j)^\top \boldsymbol{X} + b_j) \left( \frac{1}{D(\boldsymbol{X})} - \frac{1}{\check{D}(\boldsymbol{X})} \right) \right\|$$

$$+ \sum_{j=1}^{L'} \sup_{\boldsymbol{X} \in \mathcal{X}} \left\| \frac{1}{\check{D}(\boldsymbol{X})} \left[ \exp((B\boldsymbol{p}_j)^\top \boldsymbol{X} + b_j) - \exp((B\check{\boldsymbol{p}}_j)^\top \boldsymbol{X} + \check{b}_j) \right] \right\| \tag{44}$$

Note that since both the input space $\mathcal{X}$ and the parameter space $\Omega$ are bounded, the product $D(\boldsymbol{X})\check{D}(\boldsymbol{X})$ is bounded. Thus, we have

$$\frac{1}{D(\boldsymbol{X})} - \frac{1}{\check{D}(\boldsymbol{X})} \lesssim |D(\boldsymbol{X}) - \check{D}(\boldsymbol{X})|$$

$$= \left| \sum_{j'=1}^{L'} \left[ \exp((B\boldsymbol{p}_{j'})^\top \boldsymbol{X} + b_{j'}) - \exp((B\check{\boldsymbol{p}}_{j'})^\top \boldsymbol{X} + \check{b}_{j'}) \right] \right|$$

$$\lesssim \sum_{j'=1}^{L'} \left[ \|\boldsymbol{p}_{j'} - \check{\boldsymbol{p}}_{j'}\| \cdot \|\boldsymbol{X}\| + |b_j - \check{b}_{j'}| \right]$$

$$\leq L'(B+1)\tau \lesssim \tau,$$

where $B$ is the bounded constant of $\|\boldsymbol{X}\|$. As a result, we deduce that

$$\exp((B\boldsymbol{p}_j)^\top \boldsymbol{X} + b_j) \left( \frac{1}{D(\boldsymbol{X})} - \frac{1}{\check{D}(\boldsymbol{X})} \right) \lesssim \frac{1}{D(\boldsymbol{X})} - \frac{1}{\check{D}(\boldsymbol{X})} \lesssim \tau,$$

$$\frac{1}{\check{D}(\boldsymbol{X})} \left[ \exp((B\boldsymbol{p}_j)^\top \boldsymbol{X} + b_j) - \exp((B\check{\boldsymbol{p}}_j)^\top \boldsymbol{X} + \check{b}_j) \right] \lesssim \left[ \|\boldsymbol{p}_j - \check{\boldsymbol{p}}_j\| \cdot \|\boldsymbol{X}\| + |b_j - \check{b}_j| \right] \lesssim \tau.$$

Therefore, it follows that $T_2 \lesssim \tau$. Additionally, we also have

$$T_3 \leq \sum_{j=1}^{N} \sup_{\boldsymbol{X} \in \mathcal{X}} \left( \frac{1}{D(\boldsymbol{X})} - \frac{1}{\check{D}(\boldsymbol{X})} \right) \cdot \left\| \exp(\boldsymbol{X}^\top A_j^0 \boldsymbol{X} + a_j^0) h(\boldsymbol{X}, \eta_j^0) \right\|$$

$$\lesssim \tau \cdot \sum_{j=1}^{N} \sup_{\boldsymbol{X} \in \mathcal{X}} \cdot \left\| \exp(\boldsymbol{X}^\top A_j^0 \boldsymbol{X} + a_j^0) h(\boldsymbol{X}, \eta_j^0) \right\| \lesssim \tau.$$

As a result, we achieve that

$$\|f_{\bar{G}} - f_{\check{G}}\|_\infty \leq T_1 + T_2 + T_3 \lesssim \tau.$$

By definition of the covering number, we deduce that

$$N(\tau, \mathcal{F}_{L'}(\Theta), \|\cdot\|_{L^\infty}) \leq |\Delta_\tau| \leq \mathcal{O}(n^{-(d+1)L'}). \tag{45}$$

Combine equations equation 43 and equation 45, we achieve that

$$H_B(2\tau, \mathcal{F}_{L'}(\Theta), \|\cdot\|_{L^2(\mu)}) \lesssim \log(1/\tau).$$

Let $\tau = \varepsilon/2$, then we obtain that

$$H_B(\varepsilon, \mathcal{F}_{L'}(\Theta), \|\cdot\|_{L^2(\mu)}) \lesssim \log(1/\varepsilon).$$

Hence, the proof is completed.

## D RELATED WORK

**Parameter-Efficient Fine-Tuning.** Full fine-tuning is a common approach for adapting pre-trained foundation models to specific downstream tasks. However, this method requires updating all model parameters, which leads to high computational costs and the need to store a separate fine-tuned model for each task. As a more efficient alternative, parameter-efficient fine-tuning (PEFT) has emerged to address these limitations (Xin et al., 2024; Li & Liang, 2021; Hu et al., 2021). PEFT updates only a small subset of parameters, offering the potential to achieve performance comparable to, or even exceeding, that of full fine-tuning. For instance, LoRA (Hu et al., 2021) approximates weight updates through low-rank matrices that are added to the original model weights, while Bitfit (Zaken et al., 2021) modifies only the bias terms, freezing all other parameters. Adapters (Houlsby et al., 2019) introduce lightweight modules into each Transformer layer, and SSF (Lian et al., 2022) employs scaling and shifting of deep features.

**Prompt-based techniques.** Unlike the previously discussed methods of fine-tuning backbones, prompt-tuning (Lester et al., 2021) and prefix-tuning (Li & Liang, 2021) introduce learnable prompt tokens into the input space. These tokens are optimized while the backbone model remains frozen, offering substantial computational efficiency. Despite its apparent simplicity, prompting has demonstrated notable performance improvements without the need for complex module-specific designs (Liu et al., 2021). VPT (Jia et al., 2022) extends this idea to vision tasks by introducing tunable prompt tokens that are prepended to the original tokens in the first or multiple layers. Additionally, (Levine et al., 2022) introduces input-dependent prompt tuning, which generates prompt tokens using a generator. SPT (Zhu & Tan, 2023) proposes a mechanism that automatically determines which layers should receive new soft prompts and which should propagate prompts from preceding layers.

**Analysis of prompt-based techniques.** Recent research has increasingly focused on understanding the theoretical foundations that drive the success of prompt-based methods, aiming to uncover the underlying mechanisms responsible for their effectiveness. For instance, He et al. (2021) investigates the relationship between prefix-tuning and adapters, while Le et al. (2024) examines prefix-tuning within the framework of mixture of experts models. Additionally, Petrov et al. (2023) explores the limitations of prompting, demonstrating that it cannot change the relative attention patterns and can only bias the outputs of an attention layer in a fixed direction. Unlike these prior works, our study delves into the theoretical principles behind key implementation techniques, particularly reparameterization, that enable prefix-tuning to achieve competitive performance.

**Mixture of Experts.** Building on the foundational concept of mixture models (Jacobs et al., 1991; Jordan & Jacobs, 1994), prior works by Eigen et al. (2014); Shazeer et al. (2017) established the MoE layer as a key component for efficiently scaling model capacity. MoE models have since gained widespread attention for their adaptability across various domains, including large language models (Du et al., 2022; Zhou et al., 2023), computer vision (Riquelme et al., 2021; Puigcerver et al., 2023), and multi-task learning (Ma et al., 2018). Recent studies have investigated the convergence rates for expert estimation in MoE models, focusing on different assumptions and configurations of gating and expert functions. Ho et al. (2022), assuming data from an input-free gating Gaussian MoE, demonstrated that expert estimation rates for maximum likelihood estimation depend on the algebraic independence of the expert functions. Similarly, employing softmax gating, Nguyen et al. (2023; 2024a) found that expert estimation rates are influenced by the solvability of polynomial systems arising from the interaction between gating and expert parameters. More recently, Nguyen et al. (2024c;b) utilized least square estimation to propose an identifiable condition for expert functions, particularly for feedforward networks with nonlinear activations. They showed that under these conditions, estimation rates are significantly faster compared to models using polynomial experts.

Table 3: Evaluation metrics for each dataset.

| Datasets | Task | Metrics |
|---|---|---|
| FGVC | Image classification | Accuracy |
| VTAB-1K | Image classification | Accuracy |
| E2E | Table-to-text generation | BLEU, NIST, METEOR, ROUGE-L, CIDEr |
| WebNLG | Table-to-text generation | BLEU, METEOR, TER |
| XSUM | Summarization | ROUGE-1, ROUGE-2, ROUGE-L |

Table 4: Specifications of datasets evaluated for visual tasks. Following Jia et al. (2022), we randomly sampled the train and val sets since there are no public splits available.

| Dataset | Description | # Classes | Train | Val | Test |
|---|---|---|---|---|---|
| *Fine-grained visual recognition tasks (FGVC)* | | | | | |
| CUB-200-2011 (Wah et al., 2011) | Fine-grained bird species recognition | 200 | 5,394 | 600 | 5,794 |
| NABirds (Van Horn et al., 2015) | Fine-grained bird species recognition | 55 | 21,536 | 2,393 | 24,633 |
| Oxford Flowers (Nilsback & Zisserman, 2008) | Fine-grained flower species recognition | 102 | 1,020 | 1,020 | 6,149 |
| Stanford Dogs (Khosla et al., 2011) | Fine-grained dog species recognition | 120 | 10,800 | 1,200 | 8,580 |
| Stanford Cars (Gebru et al., 2017) | Fine-grained car recognition | 196 | 7,329 | 815 | 8,041 |
| *Visual Task Adaptation Benchmark (VTAB-1K)* | | | | | |
| CIFAR-100 (Krizhevsky et al., 2009) | | 100 | | | 10,000 |
| Caltech101 (Fei-Fei et al., 2006) | | 102 | | | 6,084 |
| DTD (Cimpoi et al., 2014) | | 47 | | | 1,880 |
| Flowers102 (Nilsback & Zisserman, 2008) | Natural | 102 | 800/1000 | 200 | 6,149 |
| Pets (Parkhi et al., 2012) | | 37 | | | 3,669 |
| SVHN (Netzer et al., 2011) | | 10 | | | 26,032 |
| Sun397 (Xiao et al., 2010) | | 397 | | | 21,750 |
| Patch Camelyon (Veeling et al., 2018) | | 2 | | | 32,768 |
| EuroSAT (Helber et al., 2019) | | 10 | | | 5,400 |
| Resisc45 (Cheng et al., 2017) | Specialized | 45 | 800/1000 | 200 | 6,300 |
| Retinopathy (Graham, 2015) | | 5 | | | 42,670 |
| Clevr/count (Johnson et al., 2017) | | 8 | | | 15,000 |
| Clevr/distance (Johnson et al., 2017) | | 6 | | | 15,000 |
| DMLab (Beattie et al., 2016) | | 6 | | | 22,735 |
| KITTI/distance (Geiger et al., 2013) | | 4 | | | 711 |
| dSprites/loc (Matthey et al., 2017) | Structured | 16 | 800/1000 | 200 | 73,728 |
| dSprites/ori (Matthey et al., 2017) | | 16 | | | 73,728 |
| SmallNORB/azi (LeCun et al., 2004) | | 18 | | | 12,150 |
| SmallNORB/ele (LeCun et al., 2004) | | 9 | | | 12,150 |

# E  ADDITIONAL EXPERIMENTAL DETAILS

## E.1  DATASETS DESCRIPTION

Table 4 summarizes the details of the evaluated datasets for visual tasks. Each VTAB-1K task contains 1,000 training examples. We follow the protocol from VPT (Jia et al., 2022) to perform the split of the train, validation, and test sets.

For language tasks, we employ E2E (Novikova et al., 2017) and WebNLG (Gardent et al., 2017) for table-to-text generation. The E2E dataset comprises approximately 50,000 examples across eight distinct fields, featuring multiple test references for each source table, with an average output length of 22.9 tokens. The WebNLG dataset contains 22,000 examples, where the input consists of sequences of (subject, property, object) triples, with an average output length of 22.5 tokens. For summarization, we utilize the XSUM dataset (Narayan et al., 2018), which is an abstractive summarization dataset for news articles. This dataset contains 225,000 examples, with an average article length of 431 words and an average summary length of 23.3 words.

Table 5: Per-task fine-tuning results for VTAB-1k benchmarks. We report the average accuracy over five independent runs. Best results among all methods except Finetune are **bolded**.

| Method | Natural | | | | | | | Specialized | | | | Structured | | | | | | | |
|---|---|---|---|---|---|---|---|---|---|---|---|---|---|---|---|---|---|---|---|
| | CIFAR-100 | Caltech101 | DTD | Flowers102 | Pets | SVHN | Sun397 | Patch Camelyon | EuroSAT | Resisc45 | Retinopathy | Clevr/count | Clevr/distance | DMLab | KITTI/distance | dSprites/loc | dSprites/ori | SmallNORB/azi | SmallNORB/ele |
| Finetune | 68.9 | 87.7 | 64.3 | 97.2 | 86.9 | 87.4 | 38.8 | 79.7 | 95.7 | 84.2 | 73.9 | 56.3 | 58.6 | 41.7 | 65.5 | 57.5 | 46.7 | 25.7 | 29.1 |
| Deep-share$_\text{SHALLOW}$ | **76.8** | 88.9 | 62.4 | **97.7** | **86.2** | 68.0 | **50.5** | 78.6 | 90.7 | 75.7 | 73.7 | 39.2 | 55.2 | 35.4 | 55.5 | 47.7 | 35.8 | 15.0 | 24.4 |
| No-share$_\text{SHALLOW}$ | 63.5 | 87.3 | 62.3 | 96.7 | 85.8 | 36.0 | 51.4 | 78.7 | 90.5 | 71.1 | 72.9 | 36.8 | 43.8 | 34.6 | 54.0 | 13.4 | 22.6 | 10.5 | 21.5 |
| Deep-share$_\text{DEEP}$ | 75.5 | **90.7** | **65.4** | 96.6 | 86.0 | **78.5** | 46.7 | **79.5** | **95.1** | **80.6** | **74.0** | **69.9** | **58.2** | **40.9** | **69.5** | **72.4** | **46.8** | **23.9** | **34.4** |
| No-share$_\text{DEEP}$ | 70.0 | 88.5 | 62.2 | 96.7 | 85.3 | 43.5 | 45.8 | 78.0 | 93.4 | 75.7 | 73.9 | 41.5 | 55.0 | 34.1 | 60.0 | 39.6 | 31.9 | 15.4 | 24.0 |

## E.2 IMPLEMENTATION DETAILS

In visual tasks, we preprocess the data by normalizing it with ImageNet's mean and standard deviation, applying a random resize and crop to $224 \times 224$ pixels, and implementing a random horizontal flip for FGVC datasets. For the VTAB-1k suite, we resize images directly to $224 \times 224$ pixels. Following Jia et al. (2022), we perform a grid search to determine optimal hyperparameters, specifically learning rates from the set $[50, 25, 10, 5, 2.5, 1, 0.5, 0.25, 0.1, 0.05]$ and weight decay values from $[0.01, 0.001, 0.0001, 0.0]$, evaluated on the validation set for each task. For prompt length, we select $N_p$ to ensure the number of new prefix experts within each attention head corresponds to the optimal prompt length established by Jia et al. (2022). The SGD optimizer is utilized for 100 epochs, incorporating a linear warm-up during the initial 10 epochs, followed by a cosine learning rate schedule. We report the average test set accuracy across five independent runs, maintaining consistent batch size settings of 64 and 128. All experiments were implemented in PyTorch (Paszke et al., 2017) and executed on NVIDIA A100-40GB GPUs.

In our experiments with language datasets, we adopt the hyperparameter configuration proposed by Li & Liang (2021), which includes the number of epochs, batch size, and prefix length. For the learning rate, we conduct a grid search across the following values: $[1e-1, 5e-2, 1e-2, 5e-3, 1e-3, 5e-4, 1e-4, 5e-5, 1e-5]$. During training, we utilize the AdamW optimizer with a linear learning rate scheduler. For decoding in table-to-text datasets, we implement beam search with a beam size of 5. For summarization, we employ a beam size of 6 and apply length normalization with a factor of 0.8.

## F ADDITIONAL EXPERIMENTS

### F.1 PER-TASK RESULTS ON VTAB-1K

Table 5 summarizes the results for each task on VTAB-1K. Across most datasets, either Deep-share$_\text{DEEP}$ or Deep-share$_\text{SHALLOW}$ consistently achieves the highest performance, often comparable to full fine-tuning. While prefix-tuning slightly underperforms full fine-tuning on some datasets, its average accuracy remains competitive. These results underscore the effectiveness of reparameterization in enabling prefix-tuning to perform on par with full fine-tuning. Additionally, Deep-share configurations significantly outperform No-share settings on most datasets. For instance, on SVHN, Deep-share$_\text{SHALLOW}$ outperforms No-share$_\text{SHALLOW}$ by 32%, and on Clevr/count, Deep-share$_\text{DEEP}$ exceeds No-share$_\text{DEEP}$ by 28.4%. These findings emphasize the critical role of reparameterization, highlighting the benefits of shared structures over non-shared configurations.

### F.2 PER-TASK RESULTS ON FGVC

Table 6 presents the detailed results for each task in the FGVC dataset, as visualized in Figure 2. Across all FGVC tasks, both the Simple-share and Deep-share methods consistently outperform the No-share baseline. For example, on the Stanford Cars dataset, Deep-share$_\text{Deep}$ and Simple-share$_\text{Deep}$ exceed the No-share baseline by 16.8% and 20.3%, respectively. Additionally, these methods lead to significantly higher average accuracy, surpassing the No-share baseline by 5.96%

Table 6: Per-task fine-tuning results for FGVC benchmarks. We report the average accuracy over five independent runs. Best results among all methods are **bolded**.

| Method | CUB-200-2011 | NABirds | Oxford Flowers | Stanford Dogs | Stanford Cars | Mean Acc |
|---|---|---|---|---|---|---|
| Deep-share$_{\text{SHALLOW}}$ | 87.2 | 81.5 | 98.6 | 91.1 | 63.4 | 84.36 |
| Simple-share$_{\text{SHALLOW}}$ | 86.6 | 79.3 | 98.4 | 90.8 | 55.4 | 82.10 |
| No-share$_{\text{SHALLOW}}$ | 85.1 | 77.8 | 97.9 | 86.4 | 54.7 | 80.38 |
| Deep-share$_{\text{DEEP}}$ | 87.8 | **84.5** | 98.2 | **91.6** | 79.3 | 88.28 |
| Simple-share$_{\text{DEEP}}$ | **88.7** | 84.3 | **98.8** | 90.6 | **82.8** | **89.04** |
| No-share$_{\text{DEEP}}$ | 85.9 | 79.0 | 97.9 | 86.3 | 62.5 | 82.32 |

Table 7: Comparison of fine-tuning results between common techniques on FGVC and VTAB-1K.

| Method | FGVC | VTAB-1K | | |
|---|---|---|---|---|
| | | Natural | Specialized | Structural |
| Finetune | 88.54 | 75.88 | 83.36 | 47.64 |
| Partial-1 | 82.63 | 69.44 | 78.53 | 34.17 |
| Adapter | 85.66 | 70.39 | 77.11 | 33.43 |
| VPT-Shallow | 84.62 | 76.81 | 79.66 | 46.98 |
| VPT-Deep | 89.11 | 78.48 | 82.43 | 54.98 |
| No-share$_{\text{SHALLOW}}$ | 80.38 | 69.00 | 77.20 | 29.65 |
| No-share$_{\text{DEEP}}$ | 82.32 | 70.29 | 80.20 | 37.69 |
| Deep-share$_{\text{SHALLOW}}$ | 84.36 | 75.79 | 79.48 | 38.53 |
| Deep-share$_{\text{DEEP}}$ | 88.28 | 77.06 | 82.28 | 52.00 |

and 6.72%, respectively. This substantial improvement underscores the empirical effectiveness of leveraging shared structures to enhance prefix-tuning performance. Notably, Simple-share$_{\text{Deep}}$ achieves the highest average accuracy among all methods, even surpassing full fine-tuning and Deep-share. However, the theoretical comparison between Simple-share and Deep-share remains an open question and is left for future investigation.

As shown in Figure 2 and Table 6, the performance of Simple-share and Deep-share varies depending on the task and implementation variant (i.e., `SHALLOW` and `DEEP`), with Simple-share sometimes surpassing Deep-share and vice versa. Simple-share consistently shows strong performance relative to No-share and remains competitive with Deep-share. Moreover, Simple-share offers a more parameter-efficient approach, as it does not require the additional MLP used in Deep-share for reparameterization, thereby simplifying the shared structure's implementation. In contrast, Deep-share's use of MLP for reparameterization introduces greater flexibility, which may offer benefits in specific applications. Nonetheless, a more thorough theoretical and empirical comparison between Deep-share and Simple-share remains an open question, which we propose for future research.

### F.3 COMPARISON WITH OTHER FINE-TUNING TECHNIQUES

Table 7 and Table 8 present a comparative analysis of prefix-tuning against common fine-tuning techniques. In the vision domain, prefix-tuning demonstrates competitive performance, achieving results comparable to full fine-tuning and surpassing several alternative methods, though it slightly trails behind VPT. No-share, however, shows significantly weaker performance compared to VPT, underscoring the importance of reparameterization in enhancing prefix-tuning's effectiveness. Similarly, in the language domain, prefix-tuning delivers strong results, with reparameterization once again playing a crucial role in its success relative to other fine-tuning approaches.

### F.4 VISUALIZATION OF PROMPT EFFECTS

To evaluate the effectiveness of the prompt techniques, we conduct a visual analysis to examine the areas of the input data that these techniques guide the model to focus on. Specifically, we use

Table 8: Comparison of fine-tuning results between common techniques on E2E and WebNLG.

| Method | E2E | | | | | WebNLG | | | | | | | | |
| | BLEU | NIST | MET | R-L | CIDEr | BLEU | | | MET | | | TER ↓ | | |
| | | | | | | S | U | A | S | U | A | S | U | A |
|---|---|---|---|---|---|---|---|---|---|---|---|---|---|---|
| Finetune | 68.2 | 8.62 | 46.2 | 71.0 | 2.47 | 64.2 | 27.7 | 46.5 | 0.45 | 0.30 | 0.38 | 0.33 | 0.76 | 0.53 |
| Partial-2 | 68.1 | 8.59 | 46.0 | 70.8 | 2.41 | 53.6 | 18.9 | 36.0 | 0.38 | 0.23 | 0.31 | 0.49 | 0.99 | 0.72 |
| Adapter | 66.3 | 8.41 | 45.0 | 69.8 | 2.40 | 54.5 | 45.1 | 50.2 | 0.39 | 0.36 | 0.38 | 0.40 | 0.46 | 0.43 |
| No-share | 68.0 | 8.61 | 45.8 | 71.0 | 2.41 | 61.1 | 42.8 | 53.5 | 0.43 | 0.35 | 0.40 | 0.36 | 0.49 | 0.42 |
| Deep-share | 69.9 | 8.78 | 46.3 | 71.5 | 2.45 | 63.9 | 44.3 | 54.5 | 0.45 | 0.36 | 0.41 | 0.34 | 0.52 | 0.42 |

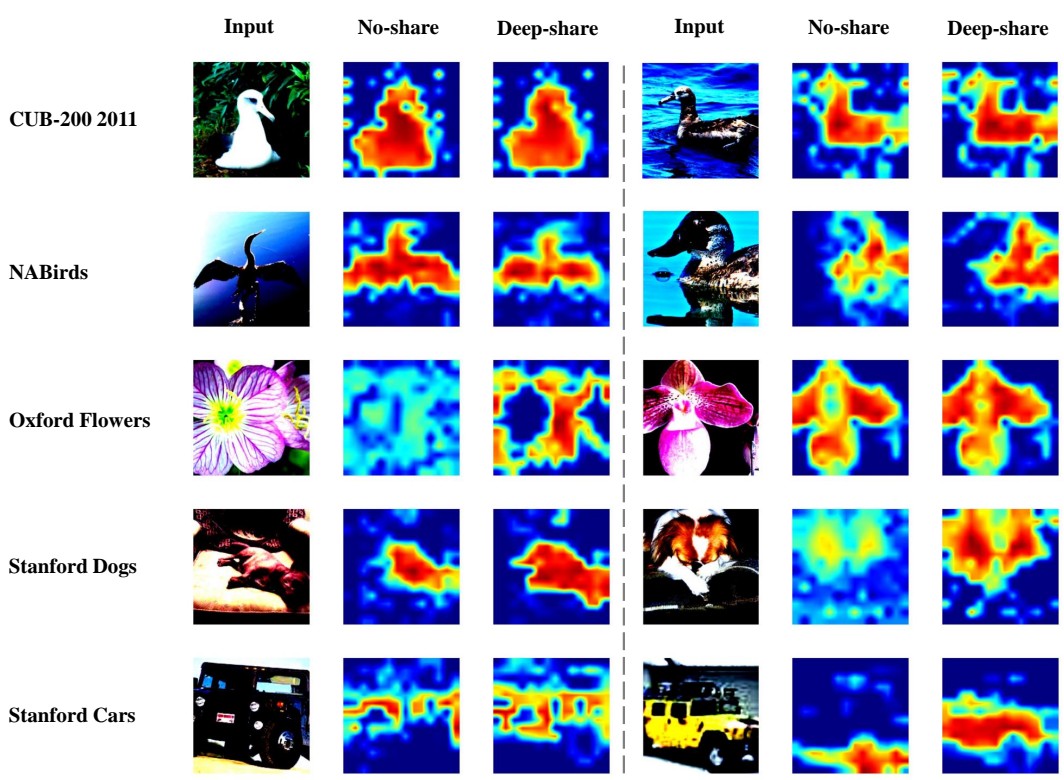

Figure 3: GradCAM visualization of Deep-share and No-share on FGVC tasks. Red regions indicate high-class activation scores. From left to right: input image after standard data augmentation, GradCAM output for No-share, and GradCAM output for Deep-share.

GradCAM (Selvaraju et al., 2017) to generate attention maps by computing the gradients of a target concept with respect to the model's final layer, highlighting the salient input regions most influential in predicting the target concept. Figure 3 illustrates examples from FGVC benchmarks, comparing heatmaps generated for Deep-share and No-share.

The visual differences between the two approaches reveal that Deep-share more effectively guides the model to attend to relevant image regions. For instance, in dog images, while No-share struggles to identify the entire structure of the dog, Deep-share successfully localizes and highlights key structural features. This ability to localize salient patterns demonstrates the improved capacity of Deep-share to capture meaningful visual information. These findings suggest that the reparame-

terization employed by Deep-share not only enhances the effectiveness of prefix-tuning but also improves the model's interpretability by providing more coherent and accurate visual explanations.

