# OpenReview forum: "Revisiting Prefix-tuning: Statistical Benefits of Reparameterization among Prompts"
_ICLR.cc/2025/Conference — ICLR 2025 Poster_

### Official Review · Reviewer_DYsW · 2024-10-28

**Soundness:** 3
**Presentation:** 3
**Contribution:** 3
**Rating:** 8
**Confidence:** 3

**Summary:**

This paper explores prefix-tuning—a parameter-efficient method for adapting large pre-trained models to downstream tasks. Specifically, the study investigates the effectiveness of reparameterization in prefix-tuning and offers theoretical foundations for its success. Prefix-tuning, in contrast to traditional fine-tuning, involves modifying only a small subset of model parameters, thereby reducing computational cost and storage requirements. The authors demonstrate that reparameterization is not merely a technical optimization but encodes a shared structure between key and value vectors, enhancing sample efficiency. By empirically validating these theoretical insights across tasks in both visual and language domains, the authors provide a new perspective on prompt-based tuning.

**Strengths:**

1. Provides a novel theoretical perspective on reparameterization, bridging prefix-tuning and mixture-of-experts frameworks.
2. Demonstrates empirical improvements across multiple tasks and architectures, supporting the theoretical claims with robust data.
3. Enhances the understanding of prompt-based tuning mechanisms, potentially broadening their applications.

**Weaknesses:**

1. The study is limited to specific configurations, leaving questions on broader applicability to different model types and prompt architectures.
2. While theoretically sound, the real-world implications, particularly computational demands of the reparameterization method, could be explored in more depth.

**Questions:**

1. How does reparameterization improve sample efficiency in prefix-tuning, and what are the core theoretical principles underlying this claim?
2. What differentiates the shared structure induced by reparameterization in prefix-tuning from other parameter-sharing mechanisms, such as those used in multi-task learning?
3. How does the proposed shared structure model compare with traditional mixture-of-experts (MoE) frameworks in terms of theoretical benefits and efficiency?
4. The authors discuss potential memory overhead associated with the reparameterization method. What optimizations could address this issue in practice?

---

> ### Author Response · Authors · 2024-11-18
> **Response to Reviewer DYsW (Part 1)**
>
> Thank you for your constructive feedback, insightful comments, and **the positive score of 8.** Below, we provide a detailed, point-by-point response to these comments.
>
> **Q1: The study is limited to specific configurations, leaving questions on broader applicability to different model types and prompt architectures.**
>
> Thank you for your comment. The purpose of our paper is to **provide theoretical evidence on the benefits of reparameterization in prefix-tuning.** To the best of our knowledge, even in the specific setting of prefix-tuning, that problem has not been rigorously studied. Finally, we would like to remark that while the current theories are for prefix-tuning, the key theoretical insights on the benefits of reparameterization can potentially still be applicable to other popular parameter-efficient fine-tuning techniques (PEFTs), such as LoRA, etc. Since it is beyond the scope of the current work, we leave the generalization of the theories to other PEFTs for future work.
>
> **Q2: Potential memory overhead associated with the reparameterization method. What optimizations could address this issue in practice?**
>
>
> Thank you for your question. Reparameterization has been widely employed in prefix-tuning to enhance its performance, and **the primary aim of our work is to provide theoretical evidence supporting the benefits of reparameterization in this context.** However, while reparameterization offers significant advantages, it also introduces potential limitations. One notable concern is the additional parameters introduced, particularly within the MLP, which can increase memory overhead during training. It is important to note, however, that **this overhead does not impact inference efficiency, as only the optimized prefix vectors are retained after training.** Nonetheless, this increased memory usage during training could pose challenges, particularly in resource-constrained environments. To mitigate this issue, several optimization strategies could be explored. For instance, techniques such as LoRA can be applied to the MLP used in reparameterization, or parameter-sharing strategies can be implemented across attention layers to reduce redundancy. These optimizations, while promising, are beyond the scope of the current work. We leave the exploration of such techniques as future work, aiming to further enhance the scalability and efficiency of reparameterization.
>
>
> **Q3: How does reparameterization improve sample efficiency in prefix-tuning, and what are the core theoretical principles underlying this claim?**
>
> Thanks for your questions. We will address your concerns respectively as follows:
>
> **1. The core theoretical principles:** In the theoretical analysis in Section 4, we consider the regression framework where the regression function takes the form of a prefix mixture of experts (MoE) model with $N$ pre-trained experts and $L$ unknown experts given by Eq.(11). Our goal is to determine the convergence rates of least square estimation of the unknown prompt parameters in this model under two main scenarios when there are non-shared (without reparameterization) and shared (with reparameterization) structures among the prompts in Section 4.1 and Section 4.2, respectively. Then, we will claim that the reparameterization improves the sample efficiency of the prefix-tuning if the convergence rates of prompt estimation under the latter scenario are faster (need fewer data points to estimate the prompts with the same approximation error) than those under the former scenario.
>
> **2. How the reparameterization improves the sample efficiency of prefix tuning:** In Section 4.1 where there is no reparameterization among the prompts, we show in Theorem 4.1 that the convergence rates of the prompt estimation are slower than any polynomial rates and could be as slow as $O_P(1/\log^{\tau}(n))$ for some constant $\tau>0$, where $n$ denotes the sample size. On the other hand, when there is a reparameterization among the prompts (Section 4.2), we demonstrate in Theorem 4.2 (under the simple linear setting) and Theorem 4.3 (under the one-layer neural network setting) that the convergence rates of the prompt estimation are either $O_P(n^{1-/2})$ or $O_P(n^{-1/4})$ (up to some logarithmic terms), which are significantly faster than those where there is no reparameterization. In other words, when employing the reparameterization strategy among the prompts, we need fewer data points to estimate the prompts with the same approximation error. Hence, we claim that the reparameterization strategy helps improve the sample efficiency of the prefix-tuning method.

---

> ### Author Response · Authors · 2024-11-18
> **Response to Reviewer DYsW (Part 2)**
>
> **Q4: What differentiates the shared structure induced by reparameterization in prefix-tuning from other parameter-sharing mechanisms, such as those used in multi-task learning?**
>
> Thanks for your question. For the simplicity of our response, we will only compare reparameterization in prefix-tuning versus parameter-sharing mechanism in multi-task learning. The shared structures in prefix tuning are among prompts in the pre-trained Transformer architecture, which leads to better optimization landscape and faster convergence rates of parameter estimation. For multi-task learning, the shared structures are among different tasks, namely, the prompts in each task are assumed to shared global prompt while having their own task-specific structures (e.g., rank-one matrix as proposed in [R.1]). While the two shared approaches are different, we conjecture that the theoretical insights from the benefits of shared structure in prefix-tuning may still be applicable to shed light into the benefits of shared structure in multi-task learning. For example, the estimation of global prompt may enjoy fast convergence rates thank to the shared structure among different tasks, which may lead to sharp optimization landscape around the global prompt. We leave a comprehensive theoretical analysis for shared structure in multi-task learning for the future work.
>
> [R.1] Zhen Wang, Rameswar Panda, Leonid Karlinsky, Rogerio Feris, Huan Sun, Yoon Kim. Multitask Prompt Tuning Enables Parameter-Efficient Transfer Learning. ICLR, 2023.
>
>
> **Q5: How does the proposed shared structure model compare with traditional mixture-of-experts (MoE) frameworks in terms of theoretical benefits and efficiency?**
>
> Thanks for your question. From the results of our theoretical study in Section 4, we can claim that the proposed shared structure model (prompt reparameterization) helps improve the sample efficiency of the prefix-tuning compared to the traditional MoE framework (without prompt reparameterization).
>
> In particular, in Section 4.1 where there is no reparameterization among the prompts, we show in Theorem 4.1 that the convergence rates of the prompt estimation are slower than any polynomial rates and could be as slow as $O_P(1/\log^{\tau}(n))$ for some constant $\tau>0$, where $n$ denotes the sample size. On the other hand, when there is a reparameterization among the prompts (Section 4.2), we demonstrate in Theorem 4.2 (under the simple linear setting) and Theorem 4.3 (under the one-layer neural network setting) that the convergence rates of the prompt estimation are either $O_P(n^{1-/2})$ or $O_P(n^{-1/4})$ (up to some logarithmic terms), which are significantly faster than those where there is no reparameterization. In other words, when employing the reparameterization strategy among the prompts, we need fewer data points to estimate the prompts with the same approximation error. Hence, we claim that the reparameterization strategy helps improve the sample efficiency of the prefix-tuning method.

---

### Official Review · Reviewer_jyvW · 2024-10-29

**Soundness:** 3
**Presentation:** 3
**Contribution:** 3
**Rating:** 6
**Confidence:** 3

**Summary:**

This study primarily explores the theoretical foundations and effectiveness of prompt-based techniques, such as prefix-tuning and prompt-tuning. The authors point out that the key to achieving performance parity with full fine-tuning lies in the reparameterization strategy, demonstrating that this strategy implicitly encodes a shared structure between prefix key and value vectors. Through extensive experiments in both visual and language domains, the research empirically confirms that the effectiveness of prefix-tuning across various tasks benefits from this shared structure. Additionally, the authors find similar structural advantages in prompt-tuning as well.

**Strengths:**

1. Many existing studies have only applied prompt techniques in different domains without providing theoretical proof of their effectiveness. In contrast, this research explores the theoretical foundations and effectiveness of prompt techniques from the perspective of reparameterization and other angles.
2. The authors' theoretical analysis shows that the shared structure significantly improves the efficiency of parameter estimation, resulting in faster convergence compared to non-shared alternatives.
3. The authors also observed that when there is a lack of shared structure among prompt parameters, the performance of prompt learning is negatively impacted. This finding underscores the importance of shared structure in optimizing the learning process and provides theoretical support for future research on how to effectively design and utilize shared structures to enhance the effectiveness of prompt learning.

**Weaknesses:**

1. The lack of additional visualization results means that, despite the authors' analysis of the effectiveness of prompt techniques, there is a deficiency in more intuitive visual outcomes. As a reader, I am particularly interested in understanding which specific areas of the images the prompts cause the model to focus on.
2. The paper primarily discusses the benefits of reparameterization, but the discussion on its potential negative impacts or limitations may be insufficient.

**Questions:**

1. It is recommended that the authors include more intuitive visual analyses, such as which areas the prompt techniques guide the model to focus on, to help readers better understand the effectiveness of the prompt techniques.
2. To improve the reproducibility of the experiments, it is recommended that the authors provide a detailed account of the experimental setup, including hyperparameters and other relevant settings.

---

> ### Author Response · Authors · 2024-11-17
> **Response to Reviewer jyvW**
>
> Thank you for your constructive feedback and insightful comments, as well as for giving **good (3)** grade to the **soundness**, **presentation**, and **contribution** of our paper. We hope to address your concerns with the responses below and eventually convince you to raise your final score.
>
> **Q1: Include more intuitive visual analyses, such as which areas the prompt techniques guide the model to focus on.**
>
> Thank you for your valuable suggestions. We have incorporated a visualization analysis of the areas influenced by the prompt techniques in the current manuscript (in blue color) under Appendix F.4.
>
> Specifically, the visual differences between Deep-share and No-share reveal that Deep-share more effectively guides the model to attend to relevant image regions. For instance, in dog images from the Stanford Dogs dataset, while No-share struggles to identify the entire structure of the dog, Deep-share successfully localizes and highlights key structural features. This ability to localize salient patterns demonstrates the improved capacity of Deep-share to capture meaningful visual information. These findings suggest that the reparameterization employed by Deep-share not only enhances the effectiveness of prefix-tuning but also improves the model's interpretability by providing more coherent and accurate visual explanations.
>
>
>
> **Q2: Discussion on potential negative impacts or limitations of reparameterization.**
>
> Thank you for your question. Reparameterization has been widely employed in prefix-tuning. However, several prior studies have utilized this technique without theoretical justifications for its effectiveness. In this work, we address this gap by exploring reparameterization from both theoretical and empirical perspectives, providing a more comprehensive understanding of its impact. Additionally, we demonstrate its crucial role in enabling prefix-tuning to achieve competitive performance across various tasks.
>
> While reparameterization offers promising benefits, it is not without potential limitations. One key limitation is the introduction of additional parameters, particularly within the MLP, which can lead to increased memory overhead during training. Although this overhead does not affect inference speed—since only the prefix vectors are retained after training —this added memory burden may become problematic during training, especially for resource-constrained environments. We suggest that future work could focus on optimizing the implementation to mitigate such memory overhead, thereby enhancing the scalability and efficiency of the reparameterization technique.
>
>
> **Q3: Detailed account of the experimental setup, including hyperparameters and other relevant settings.**
>
> Thank you for your valuable suggestions. We have already incorporated the detailed description of the experimental setup in Section 5.1 and Appendix E of the paper. This includes detailed information on the **datasets and evaluation metrics** (Section 5.1 and Appendix E), **baseline models**, **pre-trained backbones** (Section 5.1), as well as the **implementation details, including hyperparameters and tuning strategies**.

---

> > ### Author Response · Authors · 2024-11-23
> > **Looking forward to your response**
> >
> > Dear Reviewer jyvW,
> >
> > We would like to thank you very much for insightful review, and we hope that our response addresses your previous concerns regarding our paper. However, as the discussion period is expected to end in the next few days, please feel free to let us know if you have any further comments on our work. We would be willing to address any additional concerns from you. Otherwise, we hope that you will consider increasing your rating.
> >
> > Thank you again for spending time on the paper, we really appreciate it!
> >
> > Best regards,
> >
> > Authors

---

### Official Review · Reviewer_d1Nj · 2024-11-04

**Soundness:** 3
**Presentation:** 2
**Contribution:** 3
**Rating:** 6
**Confidence:** 3

**Summary:**

The paper investigates the reparameterization strategy in prompt tuning based on the mixture-of-experts model. By providing theoretical analysis on prompt learning and prefix tuning, the paper proves that the shared structure, e.g., reparameterization among prompts, helps considerably improve the sample efficiency of prompt learning. The experiments on both visual and language tasks demonstrate the theoretical analyses.

**Strengths:**

1. The paper provides detailed theoretical analyses of the reparameterization strategy for prompt learning.

2. The experiments on several tasks demonstrate the theoretical analyses.

**Weaknesses:**

1. Although the paper provides detailed theoretical proof of the effectiveness of the reparameterization strategy in prefix tuning, my main concern is about the technical contribution of the paper. Since both the reparameterization strategy and the mixture-of-experts are already investigated in previous works, the technical contribution of the paper seems a bit weak.

2. The theoretical analyses are provided based on the mixture-of-experts models. However, I didn't find the necessary connections between the MoE model and the reparameterization strategy on prefix tuning. Why do the proofs need to be analyzed on the MoE model? Do the proofs still be held without the MoE model? And is there any empirical results to demonstrate the proofs without the MoE architecture? Relying on the pretrained MoE model will also lead to increased costs.

**Questions:**

1. I found the notation $N$ in the paper confusing in the paragraph under equation 5. It is used for both the number of the input embeddings and the number of the experts. Does it means the numbers of the embeddings and experts should be the same? Otherwise, it is better to replace on of them with another notation.

2. What's the basic architecture of the MoE model in the paper? Is it also conducted by prefix tuning with the reparameterization strategy? How are the experts pretrained in the experiments?

3. In Figure 2, sometimes the simple-share method is better than the deep-share method. Any discussions on the selection of the methods?

---

> ### Author Response · Authors · 2024-11-15
> **Response to Reviewer d1Nj (Part 1)**
>
> Thank you for your constructive feedback and insightful comments. We hope to address your concerns with the responses below and eventually convince you to raise your final score.
>
> __Q1: The technical contribution of the paper.__
>
> Thank you for your question. Although reparameterization has been explored and implemented in several existing studies, its theoretical foundations remain largely unexplored. As noted by **Reviewer jyvW**, _"Many existing studies have only applied prompt techniques in different domains without providing theoretical proof of their effectiveness. In contrast, this research explores the theoretical foundations and effectiveness of prompt techniques from the perspective of reparameterization and other angles"_.
>
> To the best of our knowledge, we are among the first to **rigorously investigate reparameterization, both empirically and theoretically.** One key contribution is a comprehensive set of experiments on visual and language tasks, demonstrating that reparameterization is crucial for enabling prefix-tuning to achieve competitive performance. Beyond empirical validation, we argue that reparameterization is more than a mere engineering technique; it is supported by substantial theoretical principles. Specifically, we show that the shared structure encoded through reparameterization plays a fundamental role in its success. This claim is substantiated by additional experiments and theoretical analysis, **establishing a foundational basis for the use of reparameterization in prompt-based techniques.**
>
>
> __Q2: The necessary connections between the MoE model and the reparameterization strategy on prefix tuning. Why do the proofs need to be analyzed on the MoE model? Do the proofs still be held without the MoE model? And is there any empirical results to demonstrate the proofs without the MoE architecture? Relying on the pretrained MoE model will also lead to increased costs.__
>
> Thank you for your questions.
>
> (1) *"The necessary connections between the MoE model and the reparameterization strategy on prefix tuning":* In Section 2.2, we build upon a recent finding [1] showing that the output of each attention head can be equivalently formulated as the output of multiple MoE models. **Since this is an equivalent connection, it indicates that proving the benefits of reparameterization in prefix tuning is equivalent to establishing the benefits of reparameterization in MoE model. Therefore, our approach uses MoE as a framework to develop theoretical justification for the reparameterization strategy, not to introduce a new technique derived from MoEs.** The MoE model serves as a conceptual tool, enabling us to illustrate and analyze prefix-tuning and the reparameterization strategy.
>
>
> (2) *"Why do the proofs need to be analyzed on the MoE model? Do the proofs still be held without the MoE model?":* As explained above, the proofs rely on this MoE-based perspective to provide a structured understanding of the reparameterization strategy. Since prefix-tuning and MoE are equivalent, the proofs for the MoE model can still be used to prove the prefix-tuning setting, where the unknown experts in the MoE model become prompts in prefix-tuning.
>
>
> (3) *"Relying on the pretrained MoE model will also lead to increased costs":* As explained above, we use MoE models as a tool to represent pre-trained models, without altering the architecture of the pre-trained model in the prefix-tuning process. In prefix-tuning, the parameters of the MoE pre-trained model remain unchanged during the tuning process, thereby no additional costs are incurred when working with pre-trained MoE models..
>
>
> __Q3: The notation $N$ in the paragraph under equation 5.__
>
> Thank you for your valuable suggestion. In equation (5), we present the original architecture of Mixture of Experts, which serves as a foundation before introducing its relationship to prefix-tuning and reparameterization in the following sections. Following your suggestion, we have replaced $N$ with a distinct symbol, which is $N'$, to clearly differentiate between these two quantities. The notation in equation (5) has been updated in the current manuscript. Thank you again for your valuable feedback.

---

> ### Author Response · Authors · 2024-11-15
> **Response to Reviewer d1Nj (Part 2)**
>
> __Q4: What's the basic architecture of the MoE model in the paper? Is it also conducted by prefix tuning with the reparameterization strategy? How are the experts pre-trained in the experiments?__
>
> Thank you for your question.
>
> As described in Section 2.2, the basic architecture of an MoE model is given as:
> $$\hat{\boldsymbol{y}}= \sum_{i=1}^{N'} \frac{\exp\left(s_i(\boldsymbol{X})\right)}{\sum_{j=1}^N\exp\left(s_j(\boldsymbol{X})\right)} \cdot f_i(\boldsymbol{X})$$
> where $s_i(\boldsymbol{X})$ denotes the score function for the $i$-th expert $f_i(\boldsymbol{X})$.
>
> In the context of our work, each attention head within the MSA layer can be interpreted as a specialized architecture composed of multiple MoE models. Specifically, the output of each vector $\tilde{\boldsymbol{h}}_{l, i}$ produced by an attention head is formulated as the result of an MoE model, where the expert functions and score functions are as follows for $j \in [N], j' \in [L]$:
>
> $$
> f_j(\boldsymbol{X}) := {W_l^V}^\top E_{j} \boldsymbol{X}, \ f_{N + j'}(\boldsymbol{X}) := {W_l^V}^\top \boldsymbol{p}^V_{j'}
> $$
>
> $$
> s_{i,j}(\boldsymbol{X}) := \frac{\boldsymbol{X}^\top E_{i}^{\top} W_l^Q  {W_l^K}^\top E_{j} \boldsymbol{X}}{\sqrt{d_{v}}}, \
> s_{i, N + j'}(\boldsymbol{X}) := \frac{\boldsymbol{X}^\top E_{i}^{\top} W_l^Q  {W_l^K}^\top \boldsymbol{p}^K_{j'}}{\sqrt{d_v}}.
> $$
>
> This MoE architecture is utilized primarily as a framework to explain the reparameterization strategy in our work.
>
> Regarding the pre-trained experts used in the experiments, they are derived from the pre-trained attention weights, which remain unchanged during the fine-tuning process. In line with the principles of prefix-tuning, the pre-trained model parameters remain frozen, and only the additional parameters associated with prefix-tuning are updated. This ensures that the MoE model, as represented for the pre-trained architecture, remains unchanged while enabling fine-tuning through the added prefix.
>
>
> __Q5: In Figure 2, sometimes the simple-share method is better than the deep-share method. Any discussions on the selection of the methods?__
>
> Thank you for your valuable suggestions. Our primary objective in Section 5.2 is to illustrate the statistical advantages of incorporating shared structures within prefix-tuning, specifically through a comparative analysis of Deep-share and Simple-share against No-share configurations.
>
> As shown in Figure 2 and Table 6, the performance of Simple-share and Deep-share varies depending on the task and implementation variant (i.e., $\texttt{SHALLOW}$ and $\texttt{DEEP}$), with Simple-share sometimes surpassing Deep-share and vice versa. Simple-share consistently shows strong performance relative to No-share and remains competitive with Deep-share. Moreover, Simple-share offers a more parameter-efficient approach, as it does not require the additional MLP used in Deep-share for reparameterization, thereby simplifying the shared structure's implementation. In contrast, Deep-share's use of MLP for reparameterization introduces greater flexibility, which may offer benefits in specific applications. Nonetheless, a more thorough theoretical and empirical comparison between Deep-share and Simple-share remains an open question, which we propose for future research.
>
> We have incorporated the above discussion into the current manuscript (in blue color) in Appendix F.2.
>
>
> __References__
>
> [1] Mixture of experts meets prompt-based continual learning, NeurIPS 2024.

---

> > ### Author Response · Authors · 2024-11-25
> > **Looking forward to your response**
> >
> > Dear Reviewer d1Nj,
> >
> > We would like to thank you very much for insightful review, and we hope that our response addresses your previous concerns regarding our paper. However, as the discussion period is expected to end in the next few days, please feel free to let us know if you have any further comments on our work. We would be willing to address any additional concerns from you. Otherwise, we hope that you will consider increasing your rating.
> >
> > Thank you again for spending time on the paper, we really appreciate it!
> >
> > Best regards,
> >
> > Authors

---

> ### Comment · Reviewer_d1Nj · 2024-11-25
>
> Thanks for your responses. Most of my concerns have been addressed and I would like to increase my score to 6.

---

> > ### Author Response · Authors · 2024-11-25
> > **Thank You**
> >
> > Dear Reviewer d1Nj,
> >
> > We're glad our rebuttal addresses most of your concerns and appreciate that you increase your rating to 6.
> >
> > We will incorporate your suggestions into the revision of our paper as discussed. Please feel free to let us know if you have any further concerns.
> >
> > Best,
> >
> > The Authors

---

### Official Review · Reviewer_KNxA · 2024-11-04

**Soundness:** 3
**Presentation:** 3
**Contribution:** 2
**Rating:** 6
**Confidence:** 2

**Summary:**

This paper studies the role of reparameterization in the context of prefix-tuning within foundation models. The authors provide a theoretical study that demonstrates that reparameterization induces a shared structure between prefix key and value vectors in order to attain strong performance. They also include experimentation of language and vision-related models. Furthermore, they also tie the findings to mixture-of-expert approaches and prompt-tuning approaches.

**Strengths:**

* The paper is clearly written and includes useful foundational knowledge to help set-up the theoretical framework.
* The work studies both the case of when prompts contain shared structures and when they do not, providing extensibility to the framework
* The theoretical analysis builds usefully from a linear setting to a one-layer setting.
* The work contains a broad set of experiments across multiple vision and language oriented datasets. In addition, the experiments study prefix tuning under different kinds of fine-tuning.

**Weaknesses:**

1. The beginning of Section 4 highlights the simplification of the theoretical analysis ("we focus only on the first head, namely, l = 1 in equation (6), and the first row of the attention in this head, namely, i = 1 in
equation (6)."). The work would be strengthened with greater discussion of how this simplification might provide limitation in the extensibility of the analysis
2. The work would be strengthened by discussion of how included results on somewhat small-scale set-ups (trained on ImageNet-21k or older versions of GPT-2) might translate to larger scale models and tasks.

**Questions:**

[Numbered per Weaknesses above]
1. The work would be strengthened by more thorough discussion of potential limitations of the problem simplification within the theoretical analysis section.
2. How does results presented in Section 5.2 translate to larger scale systems?

---

> ### Author Response · Authors · 2024-11-15
> **Response to Reviewer KNxA**
>
> Thank you for your constructive feedback and insightful comments, as well as for giving **good (3)** grade to the **soundness** and **presentation** of our paper. We hope to address your concerns with the responses below and eventually convince you to raise your final score.
>
> __Q1: The simplification of the theoretical analysis in Section 4.__
>
> Thank you for your question. As outlined in Section 2.2, each output vector produced by an attention head can be formulated as the result of an MoE model. From equation (6), we can see that **each MoE model follows a similar architectural structure.** Therefore, focusing our theoretical analysis on the first head and the first row within this head is **not a restrictive simplification. Instead, enhances interpretability while preserving the generalizability of our findings, enabling us to draw meaningful insights without compromising rigor.**
>
> To illustrate in detail, consistent with the notation in Section 2.2, we express the score function as follows:
> $$
> s_{i,j}(\boldsymbol{X}) := \frac{\boldsymbol{X}^\top E_{i}^{\top} W_l^Q  {W_l^K}^\top E_{j} \boldsymbol{X}}{\sqrt{d_{v}}}
> := \boldsymbol{X}^\top A_j^0 \boldsymbol{X}, \
> s_{i, N + j'}(\boldsymbol{X}) := \frac{\boldsymbol{X}^\top E_{i}^{\top} W_l^Q  {W_l^K}^\top \boldsymbol{p}^K_{j'}}{\sqrt{d_v}} := (B \boldsymbol{p}^K_{ j'})^\top \boldsymbol{X},
> $$
> where $A_j^0 := \frac{E_{i}^{\top} W_l^Q {W_l^K}^\top E_{j}}{\sqrt{d_{v}}}$ and $B := \frac{E_{i}^{\top} W_l^Q  {W_l^K}^\top}{\sqrt{d_v}}$ are pre-trained matrices. The expert functions can be represented as:
> $$
> f_j(\boldsymbol{X}) := {W_l^V}^\top E_{j} \boldsymbol{X} := h(\boldsymbol{X},\eta^0_j) \ , \quad f_{N + j'}(\boldsymbol{X}) := {W_l^V}^\top \boldsymbol{p}^V_{j'} := C\boldsymbol{p}^V_{j'},
> $$
> where $C := {W_l^V}^\top$ is the pre-trained matrix. As shown, all MoE models in equation (6) follow the same structural form, consistent with equation (11). Thus, we focus on the first head and the first row within this head for clarity while preserving generalizability.
>
> We have revised the beginning of Section 4 in the current manuscript (in blue color) as follows:
>
> The interpretation of prefix-tuning via mixtures of experts in equation (6) provides a natural way to understand prompt learning in prefix-tuning via the convergence analysis of prompt estimation in these MoE models. **Moreover, as shown in equation (6), each MoE model in each attention head follows a similar structure. Thus, to simplify the presentation of our analysis**, we focus only on the first head, namely, $l = 1$ in equation (6), and the first row of the attention in this head, namely, $i = 1$ in equation (6). In particular, we consider a regression framework for MoE models as follows.
>
>
> __Q2: How included results might translate to larger scale models and tasks?__
>
> Thank you for your question. In Section 3, we demonstrate that reparameterization induces a shared structure between the prefix key and prefix value vectors, which we argue is fundamental to the effectiveness of prefix-tuning. To substantiate this claim, we present experimental results in Section 5.2 across both vision and language tasks, aimed at **illustrating the advantages of shared structure over non-shared configurations.** Specifically, we show that removing reparameterization (i.e., the "No-share" configuration) leads to a notable performance drop compared to setups that employ reparameterization. Furthermore, even configurations with minimal shared structure, such as the "Simple-share" setup, exhibit substantially better performance than the "No-share" case. These findings highlight the benefits of shared structure over non-shared cases and underscore the critical role of reparameterization in enhancing the effectiveness of prefix-tuning by enabling shared structure.
>
>
> Additionally, we emphasize **that reparameterization techniques are widely applied in prefix-tuning across diverse model scales, from small to large [1, 2, 3]. Our primary objective is to elucidate the mechanisms driving the effectiveness of reparameterization—particularly the advantages conferred by shared structural elements**. To this end, we focus on model setups and tasks commonly used in prior studies [1, 4], ensuring the relevance and comparability of our results. Moreover, our theoretical analysis in Section 4 makes no assumptions about specific pre-trained weights, which strengthens the scalability and generalizability of our findings to larger pre-trained datasets and more complex model architectures.
>
> __References__
>
> [1] Prefix-tuning: Optimizing continuous prompts for generation, ACL 2021.
>
> [2] P-tuning v2: Prompt  tuning can be comparable to fine-tuning universally across scales and tasks, ACL 2022.
>
> [3] Parameter-efficient fine-tuning for large  models: A comprehensive survey, arXiv 2024.
>
> [4] Visual prompt tuning, ECCV 2022.

---

> > ### Author Response · Authors · 2024-11-25
> > **Looking forward to your response**
> >
> > Dear Reviewer KNxA,
> >
> > We would like to thank you very much for insightful review, and we hope that our response addresses your previous concerns regarding our paper. However, as the discussion period is expected to end in the next few days, please feel free to let us know if you have any further comments on our work. We would be willing to address any additional concerns from you. Otherwise, we hope that you will consider increasing your rating.
> >
> > Thank you again for spending time on the paper, we really appreciate it!
> >
> > Best regards,
> >
> > Authors

---

> > > ### Comment · Reviewer_KNxA · 2024-11-25
> > >
> > > Thanks for your response to my review and for addressing my concerns. I will increase my score to 6.

---

> > > > ### Author Response · Authors · 2024-11-25
> > > > **Thank You**
> > > >
> > > > Dear Reviewer KNxA,
> > > >
> > > > We're glad our rebuttal addresses your concerns and appreciate that you increase your rating to 6.
> > > >
> > > > We will continue revising our paper based on constructive feedback from the reviewer and other reviewers. Please feel free to let us know if you have any further concerns.
> > > >
> > > > Best,
> > > >
> > > > Authors

---

### Author Response · Authors · 2024-11-18
**General Response**

Dear Area Chairs and Reviewers,

We would like to express our gratitude for your valuable feedback and suggestions, which have significantly contributed to the enhancement of our manuscript. We are encouraged by the endorsements that:

- __Contributions:__
    - __Theoretical foundations of reparameterization in prefix-tuning:__ While many existing studies have applied reparameterization in prefix-tuning without theoretical validation, our paper establishes the theoretical foundations underlying its efficacy, particularly through the shared structure (reviewer jyvW).
    - __Exploration of shared and non-shared structures:__ We demonstrate that shared structures significantly enhance parameter estimation efficiency and facilitate faster convergence (reviewers KNxA, jyvW).
    - __Novel theoretical perspective:__ Our work offers a fresh theoretical perspective on reparameterization, bridging the prefix-tuning and mixture-of-experts frameworks (reviewer DYsW).
    - __Deeper understanding of prompt-based tuning mechanisms:__ We contribute to a deeper understanding of prompt-based tuning mechanisms, emphasizing their potential for broader applications and inspiring future research on designing shared structures to improve prompt learning (reviewers DYsW, jyvW).
- __Soundness:__
    - __Comprehensive theoretical analysis:__ Our analysis is detailed (reviewer d1Nj).
    - __Robust experimental evaluation:__ Our experiments validate the theoretical analyses across multiple tasks and architectures (reviewers KNxA, d1Nj, DYsW).
- __Presentation:__ The manuscript is well-written, effectively integrating the theoretical framework with foundational knowledge (reviewer KNxA).

We will address the concerns of the reviewers separately in our responses to their reviews and include according changes in the revision of our paper.

Best regards,

Authors

---

### Author Response · Authors · 2024-11-22
**Summary of Revisions**

Dear Area Chairs and Reviewers,

We sincerely thank the reviewers for their time and effort in reviewing our submission and for providing thoughtful and constructive feedback. Below, we summarize the changes and additional results incorporated based on the reviews:

1. We have revised the discussion at the beginning of Section 4 to enhance clarity. The analysis presented is not a restrictive simplification but rather an approach that improves interpretability while preserving the generalizability of our findings. This allows us to extract meaningful insights without sacrificing rigor.
2. To improve clarity, we have updated the notation in Equation (5), changing $N$ to $N'$.
3. We have included a detailed comparison between Deep-share and Simple-share in Appendix F.2.
4. To enhance readers' understanding of the effectiveness of prompt techniques, we have added intuitive visual analyses in Appendix F.4, illustrating how prompt techniques effectively guide the model to focus on specific areas.

Thank you again for your constructive feedback, which has significantly contributed to improving our work.

Best regards,

Authors

---

### Meta-Review · Area_Chair_t4j3 · 2024-12-18

**Metareview:**

Due to the simplicity, prompt/prefix learning has often been treated as a trick. This paper builds a theoretical foundation for prompt learning and reveals that the reparameterization in prompt/prefix learning is connected to mixture-of-experts, which explains why it works well. The paper received four reviews with 3x borderline accept and 1x accept ratings. The reviews are mostly positive and all reviewers appreciated the theoretical insights. In terms of weaknesses, the reviewers questioned the technical novelty and asked for more clarifications on some concepts, which were addressed in the rebuttal. The AC thinks the insights would be of interest to the prompt learning community and possibly beyond, and therefore recommends that the paper be accepted.

**Additional Comments On Reviewer Discussion:**

The reviewers requested more clarifications on some concepts and the rebuttal has well addressed them. One reviewer questioned the technical contribution and the authors explained that the theoretical analysis provided in this work provides significant insights, which the AC agrees.

---

### Decision · Program_Chairs · 2025-01-22

Accept (Poster)